# Memory-Constrained Algorithms for Convex Optimization

**Moïse Blanchard**
Operations Research Center
MIT
Cambridge, MA 02139
moiseb@mit.edu

**Junhui Zhang**
Operations Research Center
MIT
Cambridge, MA 02139
junhuiz@mit.edu

**Patrick Jaillet**
Department of Electrical Engineering and Computer Science
MIT
Cambridge, MA 02139
jaillet@mit.edu

## Abstract

We propose a family of recursive cutting-plane algorithms to solve feasibility problems with constrained memory, which can also be used for first-order convex optimization. Precisely, in order to find a point within a ball of radius $\epsilon$ with a separation oracle in dimension $d$—or to minimize 1-Lipschitz convex functions to accuracy $\epsilon$ over the unit ball—our algorithms use $\mathcal{O}(\frac{d^2}{p} \ln \frac{1}{\epsilon})$ bits of memory, and make $\mathcal{O}((C\frac{d}{p} \ln \frac{1}{\epsilon})^p)$ oracle calls, for some universal constant $C \geq 1$. The family is parametrized by $p \in [d]$ and provides an oracle-complexity/memory trade-off in the sub-polynomial regime $\ln \frac{1}{\epsilon} \gg \ln d$. While several works gave lower-bound trade-offs (impossibility results) [31, 5]—we explicit here their dependence with $\ln \frac{1}{\epsilon}$, showing that these also hold in any sub-polynomial regime—to the best of our knowledge this is the first class of algorithms that provides a positive trade-off between gradient descent and cutting-plane methods in any regime with $\epsilon \leq 1/\sqrt{d}$. The algorithms divide the $d$ variables into $p$ blocks and optimize over blocks sequentially, with approximate separation vectors constructed using a variant of Vaidya's method. In the regime $\epsilon \leq d^{-\Omega(d)}$, our algorithm with $p = d$ achieves the information-theoretic optimal memory usage and improves the oracle-complexity of gradient descent.

## 1 Introduction

Optimization algorithms are ubiquitous in machine learning, from solving simple regressions to training neural networks. Their essential roles have motivated numerous studies on their efficiencies, which are usually analyzed through the lens of oracle-complexity: given an oracle (such as function value, or subgradient oracle), how many calls to the oracle are needed for an algorithm to output an approximate optimal solution? [34]. However, ever-growing problem sizes have shown an inadequacy in considering only the oracle-complexity, and have motivated the study of the trade-off between oracle-complexity and other resources such as memory [52, 31, 5] and communication[25, 40, 42, 45, 33, 53, 51, 50].

In this work, we study the oracle-complexity/memory trade-off for first-order non-smooth convex optimization, and the closely related feasibility problem, with a focus on developing memory efficient

37th Conference on Neural Information Processing Systems (NeurIPS 2023).

(deterministic) algorithms. Since [52] formally posed as open problem the question of characterizing this trade-off, there have been exciting results showing what is impossible: for convex optimization in $\mathbb{R}^d$, [31] shows that any randomized algorithm with $d^{1.25-\delta}$ bits of memory needs at least $\tilde{\Omega}(d^{1+4\delta/3})$ queries, and this has later been improved for deterministic algorithms to $d^{1-\delta}$ bits of memory or $\tilde{\Omega}(d^{1+\delta/3})$ queries by [5]; in addition [5] shows that for the feasibility problem with a separation oracle, any algorithm which uses $d^{2-\delta}$ bits of memory needs at least $\tilde{\Omega}(d^{1+\delta})$ queries.

Despite these recent results on the lower bounds, all known first-order convex optimization algorithms that output an $\epsilon$-suboptimal point fall into two categories: those that have quadratic memory in the dimension $d$ but can potentially achieve the optimal $\mathcal{O}(d \ln \frac{1}{\epsilon})$ query complexity, as represented by the center-of-mass method, and those that have $\mathcal{O}(\frac{1}{\epsilon^2})$ query complexity but only need the optimal $\mathcal{O}(d \ln \frac{1}{\epsilon})$ bits of memory, as represented by the classical gradient descent [52]. In addition, the above-mentioned memory bounds apply only between queries, and in particular the center-of-mass method [52] is allowed to use infinite memory during computations.

We propose a family of memory-constrained algorithms for the stronger feasibility problem in which one aims to find a point within a set $Q$ containing a ball of radius $\epsilon$, with access to a separation oracle. In particular, this can be used for convex optimization since the subgradient information provides a separation vector. Our algorithms use $\mathcal{O}(\frac{d^2}{p} \ln \frac{1}{\epsilon})$ bits of memory (including during computations) and $\mathcal{O}((C\frac{d}{p} \ln \frac{1}{\epsilon})^p)$ queries for some universal constant $C \geq 1$, and a parameter $p \in [d]$ that can be chosen by the user. Intuitively, in the context of convex optimization, the algorithms are based on the idea that for any function $f(\boldsymbol{x}, \boldsymbol{y})$ convex in the pair $(\boldsymbol{x}, \boldsymbol{y})$, the partial minimum $\min_{\boldsymbol{y}} f(\boldsymbol{x}, \boldsymbol{y})$ as a function of $\boldsymbol{x}$ is still convex and, using a variant of Vaidya's method proposed in [27], our algorithm can approximate subgradients for that function $\min_{\boldsymbol{y}} f(\boldsymbol{x}, \boldsymbol{y})$, thereby turning an optimization problem with variables $(\boldsymbol{x}, \boldsymbol{y})$ to one with just $\boldsymbol{x}$. This idea, applied recursively with the variables divided into $p$ blocks, gives our family of algorithms and the above-mentioned memory and query complexity. The main algorithmic contribution is in how we design the recursive dimension reduction procedure: a technical step of the design and analysis is to ensure that the necessary precision for recursive computations can be achieved using low memory. Last, our algorithms account for memory usage throughout computations, as opposed to simply between calls to the gradient oracle, which was the traditional approach in the literature.

When $p = 1$, our algorithm is a memory-constrained version of Vaidya's method [48, 27], and improves over the center-of-mass [52] method by a factor of $\ln \frac{1}{\epsilon}$ in terms of memory while having optimal oracle-complexity. The improvements provided by our algorithms are more significant in regimes when $\epsilon$ is very small in the dimension $d$: increasing the parameter $p$ can further reduce the memory usage of Vaidya's method ($p = 1$) by a factor $\ln \frac{1}{\epsilon} / \ln d$, while still improving over the oracle-complexity of gradient descent. In particular, in a regime $\ln \frac{1}{\epsilon} = \text{poly}(\ln d)$, these memory improvements are only in terms of $\ln d$ factors. However, in sub-polynomial regimes with potentially $\ln \frac{1}{\epsilon} = d^c$ for some constant $c > 0$, these provide polynomial improvements to the memory of standard cutting-plane methods.

As a summary, this paper makes the following contributions.

- Our class of algorithms provides a trade-off between memory-usage and oracle-complexity whenever $\ln \frac{1}{\epsilon} \gg \ln d$. Further, taking $p = 1$ improves the memory-usage from center-of-mass [52] by a factor $\ln \frac{1}{\epsilon}$, while preserving the optimal oracle-complexity.

- For $\ln \frac{1}{\epsilon} \geq \Omega(d \ln d)$, our algorithm with $p = d$ is the first known algorithm that outperforms gradient descent in terms of the oracle-complexity, but still maintains the optimal $\mathcal{O}(d \ln \frac{1}{\epsilon})$ memory usage.

- We show how to obtain a $\ln \frac{1}{\epsilon}$ dependence in the known lower-bound trade-offs [31, 5], confirming that the oracle-complexity/memory trade-off is necessary for any regime $\epsilon \lesssim \frac{1}{\sqrt{d}}$.

## 2 Setup and Preliminaries

In this section, we precise the formal setup for our results. We follow the framework introduced in [52], to define the memory constraint on algorithms with access to an oracle $\mathcal{O} : \mathcal{S} \to \mathcal{R}$ which

takes as input a query $q \in \mathcal{S}$ and outputs a response $\mathcal{O}(q) \in \mathcal{R}$. Here, the algorithm is constrained to update an internal $M$-bit memory between queries to the oracle.

**Definition 2.1** ($M$-bit memory-constrained algorithm [52, 31, 5]). *Let $\mathcal{O} : \mathcal{S} \to \mathcal{R}$ be an oracle. An $M$-bit memory-constrained algorithm is specified by a query function $\psi_{query} : \{0,1\}^M \to \mathcal{S}$ and an update function $\psi_{update} : \{0,1\}^M \times \mathcal{S} \times \mathcal{R} \to \{0,1\}^M$. The algorithm starts with the memory state $\mathsf{Memory}_0 = 0^M$ and iteratively makes queries to the oracle. At iteration $t$, it makes the query $q_t = \psi_{query}(\mathsf{Memory}_{t-1})$ to the oracle, receives the response $r_t = \mathcal{O}(q_t)$ then updates its memory $\mathsf{Memory}_t = \psi_{update}(\mathsf{Memory}_{t-1}, q_t, r_t)$.*

The algorithm can stop at any iteration and the last query is its final output. Importantly, this model does not enforce constraints on the memory usage during the computation of $\psi_{update}$ and $\psi_{query}$. This is ensured in the stronger notion of a memory-constrained algorithm with computations. These are precisely algorithms that have constrained memory including for computations, with the only specificity that they need a decoder function $\phi$ to make queries to the oracle from their bit memory, and a discretization function $\psi$ to write a discretized response into the algorithm's memory.

**Definition 2.2** ($M$-bit memory-constrained algorithm with computations). *Let $\mathcal{O} : \mathcal{S} \to \mathcal{R}$ be an oracle. We suppose that we are given a decoding function $\phi : \{0,1\}^\star \to \mathcal{S}$ and a discretization function $\psi : \mathcal{R} \times \mathbb{N} \to \{0,1\}^\star$ such that $\psi(r,n) \in \{0,1\}^n$ for all $r \in \mathcal{R}$. An $M$-bit memory-constrained algorithm with computations is only allowed to use an $M$-bit memory in $\{0,1\}^M$ even during computations. The algorithm has three special memory placements $Q, N, R$. Say the contents of $Q$ and $N$ are $q$ and $n$ respectively. To make a query, $R$ must contain at least $n$ bits. The algorithm submits $q$ to the encoder which then submits the query $\phi(q)$ to the oracle. If $r = \mathcal{O}(\phi(q))$ is the oracle response, the discretization function then writes $\psi(r,n)$ in the placement $R$.*

**Feasibility problem.** In this problem, the goal is to find a point $\boldsymbol{x} \in Q$, where $Q \subset \mathcal{C}_d := [-1,1]^d$ is a convex set. We choose the cube $[-1,1]^d$ as prior bound for convenience in our later algorithms, but the choice of norm for this prior ball can be arbitrary and does not affect our results. The algorithm has access to a *separation oracle* $O_S : \mathcal{C}_d \to \{\mathsf{Success}\} \cup \mathbb{R}^d$, such that for a query $\boldsymbol{x} \in \mathbb{R}^d$ either returns $\mathsf{Success}$ if $\boldsymbol{x} \in Q$, or a separating hyperplane $\boldsymbol{g} \in \mathbb{R}^d$, i.e., such that $\boldsymbol{g}^\top \boldsymbol{x} < \boldsymbol{g}^\top \boldsymbol{x}'$ for any $\boldsymbol{x}' \in Q$. We suppose that the separating hyperplanes are normalized, $\|\boldsymbol{g}\|_2 = 1$. An algorithm solves the feasibility problem with accuracy $\epsilon$ if the algorithm is successful for any feasibility problem such that $Q$ contains an $\epsilon$-ball $B_d(\boldsymbol{x}^\star, \epsilon)$ for $\boldsymbol{x}^\star \in \mathcal{C}_d$.

As an important remark, this formulation asks that the separation oracle is consistent over time: when queried at the exact same point $\boldsymbol{x}$, the oracle always returns the same separation vector. In this context, we can use the natural decoding function $\phi$ which takes as input $d$ sequences of bits and outputs the vector with coordinates given by the sequences interpreted in base 2. Similarly, the natural discretization function $\psi$ takes as input the separation hyperplane $\boldsymbol{g}$ and outputs a discretized version up to the desired accuracy. From now, we can omit these implementation details and consider that the algorithm can query the oracle for discretized queries $\boldsymbol{x}$, up to specified rounding errors.

**Remark 2.1.** *An algorithm for the feasibility problem with accuracy $\epsilon/(2\sqrt{d})$ can be used for first-order convex optimization. Suppose one aims to minimize a 1-Lipschitz convex function $f$ over the unit ball, and output an $\epsilon$-suboptimal solution, i.e., find a point $\boldsymbol{x}$ such that $f(\boldsymbol{x}) \le \min_{\boldsymbol{y} \in B_d(0,1)} f(\boldsymbol{y}) + \epsilon$. A separation oracle for $Q = \{\boldsymbol{x} : f(\boldsymbol{x}) \le \min_{\boldsymbol{y} \in B_d(0,1)} f(\boldsymbol{y}) + \epsilon\}$ is given at a query $\boldsymbol{x}$ by the subgradient information from the first-order oracle: $-\frac{\partial f(\boldsymbol{x})}{\|\partial f(\boldsymbol{x})\|}$. Its computation can also be carried out memory-efficiently up to rounding errors since if $\|\partial f(\boldsymbol{x})\| \le \epsilon/(2\sqrt{d})$, the algorithm can return $\boldsymbol{x}$ and already has the guarantee that $\boldsymbol{x}$ is an $\epsilon$-suboptimal solution ($\mathcal{C}_d$ has diameter $2\sqrt{d}$). Notice that because $f$ is 1-Lipschitz, $Q$ contains a ball of radius $\epsilon/(2\sqrt{d})$ (the factor $1/(2\sqrt{d})$ is due to potential boundary issues). Hence, it suffices to run the algorithm for the feasibility problem while keeping in memory the queried point with best function value.*

## 2.1 Known trade-offs between oracle-complexity and memory

**Known lower-bound trade-offs.** All known lower bounds apply to the more general class of memory-constrained algorithms without computational constraints given in Definition 2.1. [34] first showed that $\mathcal{O}(d \ln \frac{1}{\epsilon})$ queries are needed for solving convex optimization to ensure that one finds an $\epsilon$-suboptimal solution. Further, $\mathcal{O}(d \ln \frac{1}{\epsilon})$ bits of memory are needed even just to output a solution in

the unit ball with $\epsilon$ accuracy [52]. These historical lower bounds apply in particular to the feasibility problem and are represented in the pictures of Fig. 1 as the dashed pink region.

More recently, [31] showed that achieving both optimal oracle-complexity and optimal memory is impossible for convex optimization. They show that a possibly randomized algorithm with $d^{1.25-\delta}$ bits of memory makes at least $\tilde{\Omega}(d^{1+4\delta/3})$ queries. This result was extended for deterministic algorithms in [5] which shows that a deterministic algorithm with $d^{1-\delta}$ bits of memory makes $\tilde{\Omega}(d^{1+\delta/3})$ queries. For the feasibility problem, they give an improved trade-off: any deterministic algorithm with $d^{2-\delta}$ bits of memory makes $\tilde{\Omega}(d^{1+\delta})$ queries. These trade-offs are represented in the left picture of Fig. 1 as the pink, red, and purple solid region, respectively. Using a clever and more careful analysis,[11] showed that similar lower bounds can be carried out for deterministic algorithms as well.

**Known upper-bound trade-offs.** Prior to this work, to the best of our knowledge only two algorithms were known in the oracle-complexity/memory landscape. First, cutting-plane algorithms achieve the optimal oracle-complexity $\mathcal{O}(d\ln\frac{1}{\epsilon})$ but use quadratic memory. The memory-constrained (MC) center-of-mass method analyzed in [52] uses in particular $\mathcal{O}(d^2\ln^2\frac{1}{\epsilon})$ memory. Instead, if one uses Vaidya's method which only needs to store $\mathcal{O}(d)$ cuts instead of $\mathcal{O}(d\ln\frac{1}{\epsilon})$, we show that one can achieve $\mathcal{O}(d^2\ln\frac{1}{\epsilon})$ memory. These algorithms only use the separation oracle and hence apply to both convex optimization and the feasibility problem. On the other hand, the memory-constrained gradient descent for convex optimization [52] uses the optimal $\mathcal{O}(d\ln\frac{1}{\epsilon})$ memory but makes $\mathcal{O}(\frac{1}{\epsilon^2})$ iterations. While the analysis in [52] is only carried for convex optimization, we can give a modified proof showing that gradient descent can also be used for the feasibility problem.

## 2.2 Other related works

Vaidya's method [48, 38, 1, 2] and the variant [27] that we use in our algorithms, belong to the family of cutting-plane methods. Perhaps the simplest example of an algorithm in this family is the center-of-mass method, which achieves the optimal $\mathcal{O}(d\ln\frac{1}{\epsilon})$ oracle-complexity but is computationally intractable, and the only known random walk-based implementation [4] has computational complexity $\mathcal{O}(d^7\ln\frac{1}{\epsilon})$. Another example is the ellipsoid method, which has suboptimal $\mathcal{O}(d^2\ln\frac{1}{\epsilon})$ query complexity, but has an improved computational complexity $\mathcal{O}(d^4\ln\frac{1}{\epsilon})$. [8] pointed out that Vaidya's method achieves the best of both worlds by sharing the $\mathcal{O}(d\ln\frac{1}{\epsilon})$ optimal query complexity of the center-of-mass, and achieving a computational complexity of $\mathcal{O}(d^{1+\omega}\ln\frac{1}{\epsilon})$[1]. In a major breakthrough, this computational complexity was improved to $\mathcal{O}(d^3\ln^3\frac{1}{\epsilon})$ in [27], then to $\mathcal{O}(d^3\ln\frac{1}{\epsilon})$ in [22]. We refer to [8, 27, 22] for more detailed comparisons of these algorithms.

Another popular convex optimization algorithm that requires quadratic memory is the Broyden– Fletcher– Goldfarb– Shanno (BFGS) algorithm [43, 7, 20, 21], which stores an approximated inverse Hessian matrix as gradient preconditioner. Several works aimed to reduce the memory usage of BFGS; in particular, the limited memory BFGS (L-BFGS) stores a few vectors instead of the entire approximated inverse Hessian matrix [37, 30]. However, it is still an open question whether even the original BFGS converges for non-smooth convex objectives [29].

Lying at the other extreme of the oracle-complexity/memory trade-off is gradient descent, which achieves the optimal memory usage but requires significantly more queries than center-of-mass or Vaidya's method in the regime $\epsilon \lesssim \frac{1}{\sqrt{d}}$. There is a rich literature of works aiming to speed up gradient descent, such as the optimized gradient method [17, 16], Nesterov's Acceleration [35], the triple momentum method [41], geometric descent [9], quadratic averaging [18], the information-theoretic exact method [46], or Big-Step-Little-Step method [23]. Interested readers can find a comprehensive survey on acceleration methods in [12]. However, these acceleration methods usually require additional smoothness or strong convexity assumptions (or both) on the objective function, due to the known $\Omega(\frac{1}{\epsilon^2})$ query lower bound in the large-scale regime $\epsilon \gtrsim \frac{1}{\sqrt{d}}$ for any first order method where the query points lie in the span of the subgradients of previous query points [36].

Besides accelerating gradient descent, researchers have investigated more efficient ways to leverage subgradients obtained in previous iterations. Of interest are bundle methods [3, 24, 28], that have

---

[1] $\omega < 2.373$ is the exponent of matrix multiplication

found a wide range of applications [47, 26]. In their simplest form, they minimize the sum of the maximum of linear lower bounds constructed using past oracle queries, and a regularization term penalizing the distance from the current iteration variable. Although the theoretical convergence rate of the bundle method is the same as that of gradient descent, in practice, bundle methods can benefit from previous information and substantially outperform gradient descent [3].

Our works are focused on high-accuracy regimes, when the accuracy $\epsilon$ is sub-polynomial. We note that for their lower-bound result on randomized algorithms, [11] also required sub-polynomial accuracies, which raises the question whether this is a general phenomenon for the study of memory-constrained algorithms in convex optimization. This also relates our work to the study of low-dimensional problems—or even constant dimension—which has been investigated in the literature [49, 10].

Last, the increasing size of optimization problems has also motivated the development of communication-efficient optimization algorithms in distributed settings such as [25, 40, 42, 45, 33, 53, 51, 50]. Moreover, recent works have explored the trade-off between sample complexity and memory/communication complexity for learning problems under the streaming model, with notable contributions including [6, 13, 14, 39, 44, 32].

## 3 Main results

We first check that the memory-constrained gradient descent method solves feasibility problems. This was known for convex optimization [52] and the same algorithm with a modified proof gives the following result. For completeness, the proof is given in Appendix D.

**Proposition 3.1.** *The memory-constrained gradient descent algorithm solves the feasibility problem with accuracy $\epsilon \leq \frac{1}{\sqrt{d}}$ using $\mathcal{O}(d \ln \frac{1}{\epsilon})$ bits of memory and $\mathcal{O}(\frac{1}{\epsilon^2})$ separation oracle calls.*

Our main contribution is a class of algorithms based on Vaidya's cutting-plane method that provide a query-complexity / memory tradeoff for optimization in $\mathbb{R}^d$. More precisely, we show the following, where $\omega < 2.373$ is the exponent of matrix multiplication, such that multiplying two $n \times n$ matrices runs in $\mathcal{O}(n^\omega)$ time.

**Theorem 3.2.** *For any $1 \leq p \leq d$, there is a deterministic first-order algorithm that solves the feasibility problem in dimension $d$ for accuracy $\epsilon \leq \frac{1}{\sqrt{d}}$, using $\mathcal{O}(\frac{d^2}{p} \ln \frac{1}{\epsilon})$ bits of memory (including during computations), with $\mathcal{O}((C\frac{d}{p} \ln \frac{1}{\epsilon})^p)$ calls to the separation oracle, and computational complexity $\mathcal{O}((C(\frac{d}{p})^{1+\omega} \ln \frac{1}{\epsilon})^p)$, where $C \geq 1$ is a universal constant.*

For simplicity, in Section 4, we describe algorithms that achieve this trade-off without computation concerns (Definition 2.1), which already provide the main elements of our method. The proof of oracle-complexity and memory usage is given in Appendix A. In Appendix B, we consider computational constraints and give corresponding algorithms using the cutting-plane method of [27].

To better understand the implications of Theorem 3.2, it is useful to compare the provided class of algorithms to the two algorithms known in the oracle-complexity/memory tradeoff landscape: the memory-constrained center-of-mass method and the memory-constrained gradient descent [52].

For $p = 1$, our resulting procedure, which is essentially a memory-constrained Vaidya's algorithm, has optimal oracle-complexity $\mathcal{O}(d \ln \frac{1}{\epsilon})$ and uses $\mathcal{O}(d^2 \ln \frac{1}{\epsilon})$ bits of memory. This improves by a $\ln \frac{1}{\epsilon}$ factor the memory usage of the center-of-mass-based algorithm provided in [52], which used $\mathcal{O}(d^2 \ln^2 \frac{1}{\epsilon})$ memory and had the same optimal oracle-complexity.

Next, we recall that the memory-constrained gradient descent method used the optimal number $\mathcal{O}(d \ln \frac{1}{\epsilon})$ bits of memory (including for computations), and a sub-optimal $\mathcal{O}(\frac{1}{\epsilon^2})$ oracle-complexity. While the memory of our algorithms decreases with $p$, their oracle-complexity is exponential in $p$. This significantly restricts the values of $p$ for which the oracle-complexity is improved over that of gradient descent. The range of application of Theorem 3.2 is given in the next result, where $\vee$ and $\wedge$ represent maximum and minimum respectively.

**Corollary 3.1.** *The algorithms given in Theorem 3.2 effectively provide a tradeoff for $p \leq \mathcal{O}(\frac{\ln \frac{1}{\epsilon}}{\ln d} \wedge d)$. Precisely, this provides a tradeoff between*

- *using $\mathcal{O}(d^2 \ln \frac{1}{\epsilon})$ memory with optimal $\mathcal{O}(d \ln \frac{1}{\epsilon})$ oracle-complexity, and*

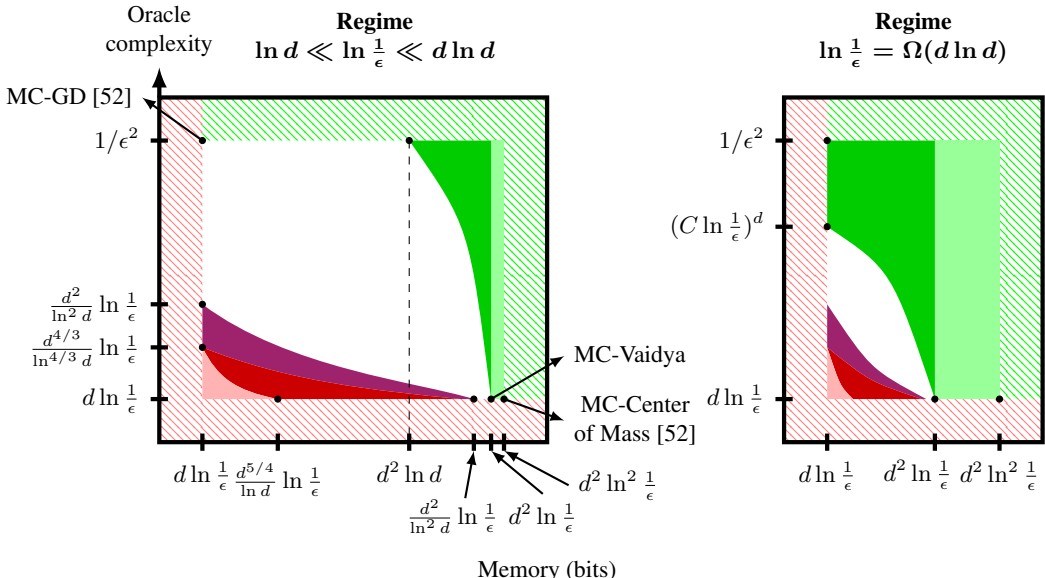

Figure 1: Trade-offs between available memory and first-order oracle-complexity for the feasibility problem over the unit ball. MC=Memory-constrained. GD=Gradient Descent. The left picture corresponds to the regime $\epsilon \gg d^{-\Omega(d)}$ and $\epsilon \le 1/\text{poly}(d)$ and the right picture represents the regime $\epsilon \le d^{-\mathcal{O}(d)}$. For both figures, the dashed pink "L" (resp. green inverted "L") region corresponds to historical lower (resp. upper) bounds for randomized algorithms. The solid pink (resp. red) lower bound tradeoff is due to [31] (resp. [5]) for randomized algorithms (resp. deterministic algorithms). The purple region is a lower bound tradeoff for the feasibility problem for accuracy $\epsilon$ and deterministic algorithms [5]. All these lower-bound trade-offs are represented with their $\ln \frac{1}{\epsilon}$ dependence (Theorem 3.3). We use memory-constrained Vaidya's method to gain a factor $\ln \frac{1}{\epsilon}$ in memory compared to memory-constrained center-of-mass [52], which gives the light green region, and a class of algorithms represented in dark green, that allows trading query-complexity for an extra $\ln \frac{1}{\epsilon} / \ln d$ factor saved in memory (Theorem 3.2). The dark green dashed region in the left figure emphasizes that the area covered by our class of algorithms depends highly on the regime for the accuracy $\epsilon$: the resulting improvement in memory is more significant as $\epsilon$ is smaller. In the regime when $\epsilon \le d^{-\mathcal{O}(d)}$ (right figure), our class of algorithms improves over the oracle-complexity of gradient descent while keeping the optimal memory $\mathcal{O}(d \ln \frac{1}{\epsilon})$.

- *using $\mathcal{O}(d^2 \ln d \vee d \ln \frac{1}{\epsilon})$ memory with $\mathcal{O}(\frac{1}{\epsilon^2} \wedge (C \ln \frac{1}{\epsilon})^d)$ oracle-complexity.*

Importantly, for $\epsilon \le \frac{1}{d^{\Omega(d)}}$, taking $p = d$ yields an algorithm that uses the optimal memory $\mathcal{O}(d \ln \frac{1}{\epsilon})$ and has an improved query complexity over gradient descent. In this regime of small (virtually constant) dimension, for the same memory usage, gradient descent has a query complexity that is polynomial in $\epsilon$, $\mathcal{O}(\frac{1}{\epsilon^2})$, while our algorithm has poly-logarithmic dependence in $\epsilon$, $\mathcal{O}_d(\ln^d \frac{1}{\epsilon})$, where $\mathcal{O}_d$ hides an exponential constant in $d$. It remains open whether this $\ln^d \frac{1}{\epsilon}$ dependence in the oracle-complexity is necessary. To the best of our knowledge, this is the first example of an algorithm that improves over gradient descent while keeping its optimal memory usage in any regime where $\epsilon \le \frac{1}{\sqrt{d}}$. While this improvement holds only in the exponential regime $\epsilon \le \frac{1}{d^{\mathcal{O}(d)}}$, Theorem 3.2 still provides a non-trivial trade-off whenever $\ln \frac{1}{\epsilon} \gg \ln d$, and improves over the known memory-constrained center-of-mass in the standard regime $\epsilon \le \frac{1}{\sqrt{d}}$ [52]. Fig. 1 depicts the trade-offs in the two regimes mentioned earlier.

Last, we note that the lower-bound trade-offs presented in [31, 5] do not show a dependence in the accuracy $\epsilon$. Especially in the regime when $\ln \frac{1}{\epsilon} \gg \ln d$, this yields sub-optimal lower bounds (in fact even in the regime $\epsilon = 1/\text{poly}(d)$, our more careful analysis improves the lower bound on the memory by a $\ln d$ factor). We show with simple arguments that one can extend their results to include a $\ln \frac{1}{\epsilon}$ factor for both memory and query complexity. Fig. 1 presented these improved lower bounds.

**Theorem 3.3.** *For $\epsilon \leq 1/poly(d)$ and any $\delta \in [0, 1]$ (the notation $\tilde{\Omega}$ hides $\ln^{\mathcal{O}(1)} d$ factors),*

1. *any (randomized) algorithm guaranteed to minimize 1-Lipschitz convex functions over the unit ball with accuracy $\epsilon$ uses $d^{5/4-\delta} \ln \frac{1}{\epsilon}$ bits of memory or makes $\tilde{\Omega}(d^{1+4\delta/3} \ln \frac{1}{\epsilon})$ queries,*

2. *any deterministic algorithm guaranteed to minimize 1-Lipschitz convex functions over the unit ball with accuracy $\epsilon$ uses $d^{2-\delta} \ln \frac{1}{\epsilon}$ bits of memory or makes $\tilde{\Omega}(d^{1+\delta/3} \ln \frac{1}{\epsilon})$ queries,*

3. *any deterministic algorithm guaranteed to solve the feasibility problem over the unit ball with accuracy $\epsilon$ uses $d^{2-\delta} \ln \frac{1}{\epsilon}$ bits of memory or makes $\tilde{\Omega}(d^{1+\delta} \ln \frac{1}{\epsilon})$ queries.*

The proof is given in Appendix C and the arguments therein could readily be used to exhibit the $\ln \frac{1}{\epsilon}$ dependence of potential future works improving over these lower bounds trade-offs.

**Sketch of proof.** At a high level, [31, 5] use a barrier term $\|\boldsymbol{Ax}\|_\infty$ where $\boldsymbol{A}$ has $\Theta(d)$ rows: if an algorithm does not have enough memory, $\boldsymbol{A}$ cannot be fully stored which in turn incurs a sub-optimal oracle-complexity. To achieve a $\ln \frac{1}{\epsilon}$ improvement in memory (Appendix C.1), we modify the sampling of rows of $\boldsymbol{A}$, from uniform on vertices of the hypercube to uniform in an $\epsilon$-net. The proof can then be adapted accordingly. Last, one can improve the oracle-complexity by a $\ln \frac{1}{\epsilon} / \ln d$ factor (Appendix C.2) using a standard rescaling argument [34].

# 4 Memory-constrained feasibility problem without computation

In this section, we present a class of algorithms that are memory-constrained according to Definition 2.1 and achieve the desired memory and oracle-complexity bounds. We emphasize that the memory constraint is only applied between calls to the oracle and as a result, the algorithm is allowed infinite computation memory and computation power between calls to the oracle.

We start by defining discretization functions that will be used in our algorithms. For $\xi > 0$ and $x \in [-1, 1]$, we pose $\mathsf{Discretize}_1(x, \xi) = sign(x) \cdot \xi \lfloor |x|/\xi \rfloor$. Next, we define the discretization $\mathsf{Discretize}_d$ for general dimensions $d \geq 1$. For any $\boldsymbol{x} \in C$ and $\xi > 0$,

$$\mathsf{Discretize}_d(\boldsymbol{x}, \xi) = \left( \mathsf{Discretize}_1 \left( x_1, \xi/\sqrt{d} \right), \ldots, \mathsf{Discretize}_1 \left( x_d, \xi/\sqrt{d} \right) \right).$$

## 4.1 Memory-constrained Vaidya's method

Our algorithm recursively uses Vaidya's cutting-plane method [48] and subsequent works expanding on this method. We briefly describe the method. Given a polyhedron $\mathcal{P} = \{\boldsymbol{x} : \boldsymbol{Ax} \geq \boldsymbol{b}\}$, we define $s_i(\boldsymbol{x}) = \boldsymbol{a}_i^\top \boldsymbol{x} - b_i$ and $\boldsymbol{S}_x = diag(s_i(x), i \in [d])$. We will also use the shorthand $\boldsymbol{A}_x = \boldsymbol{S}_x^{-1} \boldsymbol{A}$. The volumetric barrier is defined as

$$V_{\boldsymbol{A},\boldsymbol{b}}(\boldsymbol{x}) = \frac{1}{2} \ln \det(\boldsymbol{A}_x^\top \boldsymbol{A}_x).$$

At each step, Vaidya's method queries the volumetric center of the polyhedron, which is the point minimizing the volumetric barrier. For convenience, we denote by $\mathsf{VolumetricCenter}$ this function, i.e., for any $\boldsymbol{A} \in \mathbb{R}^{m \times d}$ and $\boldsymbol{b} \in \mathbb{R}^d$ defining a non-empty polyhedron $\mathcal{P} = \{\boldsymbol{x} : \boldsymbol{Ax} \geq \boldsymbol{b}\}$,

$$\mathsf{VolumetricCenter}(\boldsymbol{A}, \boldsymbol{b}) = \arg \min_{\boldsymbol{x}:\boldsymbol{Ax}>\boldsymbol{b}} V_{\boldsymbol{A},\boldsymbol{b}}(\boldsymbol{x}).$$

When the polyhedron is unbounded, we can for instance take $\mathsf{VolumetricCenter}(\boldsymbol{A}, \boldsymbol{b}) = \boldsymbol{0}$. Vaidya's method makes use of leverage scores for each constraint $i$ of the polyhedron, defined as $\sigma_i = (\boldsymbol{A}_x \boldsymbol{H}^{-1} \boldsymbol{A}_x^\top)_{i,i}$, where $\boldsymbol{H} = \boldsymbol{A}_x^\top \boldsymbol{A}_x$. We are now ready to define the update procedure for the polyhedron considered by Vaidya's volumetric method. We denote by $\mathcal{P}_t$ the polyhedron stored in memory after making $t$ queries. The method keeps in memory the constraints defining the current polyhedron and the iteration index $k$ when these constraints were added, which will be necessary for our next procedures. Hence, the polyhedron will be stored in the form $\mathcal{P}_t = \{(k_i, \boldsymbol{a}_i, b_i), i \in [m]\}$, and the associated constraints are given via $\{\boldsymbol{x} : \boldsymbol{Ax} \geq \boldsymbol{b}\}$ where $\boldsymbol{A}^\top = [\boldsymbol{a}_1, \ldots, \boldsymbol{a}_m]$ and $\boldsymbol{b}^\top = [b_1, \ldots, b_m]$. By abuse of notation, we will write $\mathsf{VolumetricCenter}(\mathcal{P})$ for the volumetric center of the polyhedron $\mathsf{VolumetricCenter}(\boldsymbol{A}, \boldsymbol{b})$ where $\boldsymbol{A}$ and $\boldsymbol{b}$ define the constraints stored in $\mathcal{P}$.

Initially, the polyhedron is simply $\mathcal{C}_d$, these constraints are given $-1$ index for convenience, and they will not play a role in the next steps. At each iteration, if the constraint $i \in [m]$ with minimum leverage score $\sigma_i$ falls below a given threshold $\sigma_{min}$, it is removed from the polyhedron. Otherwise, we query the volumetric center of the current polyhedron and add the separation hyperplane as a constraint to the polyhedron. We bound the number of iterations of the procedure by

$$T(\delta, d) = \left\lceil c \cdot d \left( 1.4 \ln \frac{1}{\delta} + 2 \ln d + 2 \ln(1 + 1/\sigma_{min}) \right) \right\rceil,$$

where $\sigma_{min}$ and $c$ are parameters that will be fixed shortly. Instead of making a call directly to the oracle $O_S$, we instead suppose that one has access to an oracle $O : \mathcal{I}_d \to \mathbb{R}^d$ where $\mathcal{I}_d = (\mathbb{Z} \times \mathbb{R}^{d+1})^\star$ has exactly the shape of the memory storing the information from the polyhedron. This form of oracle is used in our recursive calls to Vaidya's method. For example, such an oracle can simply be $O : \mathcal{P} \in \mathcal{I}_d \mapsto O_S(\mathsf{VolumetricCenter}(\mathcal{P}))$. Last, in our recursive method, we will not assume that oracle responses are normalized. As a result, we specify that if the norm of the response is too small, we can stop the algorithm. We assume however that the oracle already returns discretized vectors, which will be ensured in the following procedures. The cutting-plane algorithm is formally described in Algorithm 1. With an appropriate choice of parameters, this procedure finds an approximate solution of feasibility problems. We base the constants from [2].

---

**Input:** $O : \mathcal{I}_d \to \mathbb{R}^d$, $\delta, \xi \in (0,1)$
1 Let $T_{max} = T(\delta, d)$ and initialize $\mathcal{P}_0 := \{(-1, \boldsymbol{e}_i, -1), (-1, -\boldsymbol{e}_i, -1), i \in [d]\}$
2 **for** $t = 0, \ldots, T_{max}$ **do**
3      **if** $\{\boldsymbol{x} : \boldsymbol{A}\boldsymbol{x} \geq \boldsymbol{b}\} = \emptyset$ **then return** $\mathcal{P}_t$;
4      **if** $\min_{i \in [m]} \sigma_i < \sigma_{min}$ **then**
5         $\mathcal{P}_{t+1} = \mathcal{P}_t \setminus \{(k_j, \boldsymbol{a}_j, b_j)\}$ where $j \in \arg\min_{i \in [m]} \sigma_i$
6      **else if** $\boldsymbol{\omega} := \mathsf{VolumetricCenter}(\mathcal{P}_t) \notin \mathcal{C}_d$ **then**
7         $\mathcal{P}_{t+1} = \mathcal{P}_t \cup \{(-1, -sign(\omega_j)\boldsymbol{e}_j, -1)\}$ where $j \in [d]$ has $|\omega_j| > 1$
8      **else**
9         $\boldsymbol{g} = O(\mathcal{P}_t)$ and $b = \xi \left\lceil \frac{\boldsymbol{g}^\top \boldsymbol{\omega}}{\xi} \right\rceil$, where $\boldsymbol{\omega} = \mathsf{VolumetricCenter}(\mathcal{P}_t)$
10        $\mathcal{P}_{t+1} = \mathcal{P}_t \cup \{(t, \boldsymbol{g}, b)\}$
11        **if** $\|\boldsymbol{g}\| \leq \delta$ **then return** $\mathcal{P}_{t+1}$ ;
12 **end**
13 **return** $\mathcal{P}_{T_{max}+1}$.

**Algorithm 1:** Memory-constrained Vaidya's volumetric method

---

**Lemma 4.1.** *Fix $\sigma_{min} = 0.04$ and $c = \frac{1}{0.0014} \approx 715$. Let $\delta, \xi \in (0,1)$ and $O : \mathcal{I}_d \to \mathbb{R}^d$. Write $\mathcal{P} = \{(k_i, \boldsymbol{a}_i, b_i), i \in [m]\}$ as the output of Algorithm 1 run with $O$, $\delta$ and $\xi$. Then,*

$$\min_{\substack{\lambda_i \geq 0,\ i \in [m], \\ \sum_{i \in [m]} \lambda_i = 1}} \max_{\boldsymbol{y} \in \mathcal{C}_d} \sum_{i=1}^m \lambda_i (\boldsymbol{a}_i^\top \boldsymbol{y} - b_i) = \max_{\boldsymbol{x} \in \mathcal{C}_d} \min_{i \in [m]} (\boldsymbol{a}_i^\top \boldsymbol{x} - b_i) \leq \delta.$$

From now, we use the parameters $\sigma_{min} = 0.04$ and $c = 1/0.0014$ as in Lemma 4.1. Since the memory of both Vaidya's method and center-of-mass consists primarily of the constraints, we recall an important feature of Vaidya's method that the number of constraints at any time is $\mathcal{O}(d)$.

**Lemma 4.2** ([48, 1, 2]). *At any time while running Algorithm 1, the number of constraints of the current polyhedron is at most $\frac{d}{\sigma_{min}} + 1$.*

## 4.2 A recursive algorithm

We write $\mathcal{C}_{m+n} = \mathcal{C}_m \times \mathcal{C}_n$ and aim to apply Vaidya's method to the first $m$ coordinates. To do so, we need to approximate a separation oracle on these $m$ coordinates only, which corresponds to giving separation hyperplanes with small values for the last $n$ coordinates. This can be achieved using the following auxiliary linear program. For $\mathcal{P} \in \mathcal{I}_n$, we define

$$\min_{\substack{\lambda_i \geq 0,\ i \in [m], \\ \sum_{i \in [m]} \lambda_i = 1}} \max_{\boldsymbol{y} \in \mathcal{C}_n} \sum_{i=1}^m \lambda_i (\boldsymbol{a}_i^\top \boldsymbol{y} - b_i), \quad m = |\mathcal{P}| \qquad (\mathcal{P}_{aux}(\mathcal{P}))$$

where as before, $\boldsymbol{A}$ and $\boldsymbol{b}$ define the constraints stored in $\mathcal{P}$. The procedure to obtain an approximate separation oracle on the first $n$ coordinates $\mathcal{C}_n$ is given in Algorithm 2 and using Lemma 4.1 we can show that this procedure provides approximate separation vectors for the first $n$ coordinates.

**Input:** $\delta, \xi, O_x : \mathcal{I}_n \to \mathbb{R}^m$ and $O_y : \mathcal{I}_n \to \mathbb{R}^n$
1 Run Algorithm 1 with $\delta, \xi$ and $O_y$ to obtain polyhedron $\mathcal{P}^\star$
2 Solve $\mathcal{P}_{aux}(\mathcal{P}^\star)$ to get a solution $\boldsymbol{\lambda}^\star$
3 Store $\boldsymbol{k}^\star = (k_i, i \in [m])$ where $m = |\mathcal{P}^\star|$, and $\boldsymbol{\lambda}^\star \leftarrow \mathsf{Discretize}(\boldsymbol{\lambda}^\star, \xi)$
4 Initialize $\mathcal{P}_0 := \{(-1, \boldsymbol{e}_i, -1), (-1 - \boldsymbol{e}_i, -1), \ i \in [d]\}$ and $\boldsymbol{u} = \boldsymbol{0} \in \mathbb{R}^m$
5 **for** $t = 0, 1, \dots, \max_i k_i$ **do**
6      **if** $t = k_i^\star$ *for some* $i \in [m]$ **then**
7          $\boldsymbol{g}_x = O_x(\mathcal{P}_t)$
8          $\boldsymbol{u} \leftarrow \mathsf{Discretize}_m(\boldsymbol{u} + \lambda_i^\star \boldsymbol{g}_x, \xi)$
9      Update $\mathcal{P}_t$ to get $\mathcal{P}_{t+1}$ as in Algorithm 1
10 **end**
11 **return** $\boldsymbol{u}$

**Algorithm 2:** $\mathsf{ApproxSeparationVector}_{\delta,\xi}(O_x, O_y)$

The next step involves using this approximation recursively. We write $d = \sum_{i=1}^p k_i$, and interpret $\mathcal{C}_d$ as $\mathcal{C}_{k_1} \times \cdots \times \mathcal{C}_{k_p}$. In particular, for $\boldsymbol{x} \in \mathcal{C}_d$, we write $\boldsymbol{x} = (\boldsymbol{x}_1, \dots, \boldsymbol{x}_p)$ where $\boldsymbol{x}_i \in \mathcal{C}_{k_i}$ for $i \in [p]$. Applying Algorithm 2 recursively, we can obtain an approximate separation oracle for the first $i$ coordinates $\mathcal{C}_{k_1} \times \cdots \times \mathcal{C}_{k_i}$. However, storing such separation vectors would be too memory-expensive, e.g., for $i = p$, that would correspond to storing the separation hyperplanes from the oracle $O_S$ directly. Instead, given $j \in [i]$, Algorithm 3 recursively computes the $\boldsymbol{x}_j$ component of an approximate separation oracle for the first $i$ variables $(\boldsymbol{x}_1, \dots, \boldsymbol{x}_i)$, via the procedure $\mathsf{ApproxOracle}(i, j)$.

**Input:** $\delta, \xi, 1 \le j \le i \le p, \mathcal{P}^{(r)} \in \mathcal{I}_{k_r}$ for $r \in [i], O_S : \mathcal{C}_d \to \mathbb{R}^d$
1 **if** $i = p$ **then**
2      $\boldsymbol{x}_r = \mathsf{VolumetricCenter}(\boldsymbol{A}_r, \boldsymbol{b}_r)$ where $(\boldsymbol{A}_r, \boldsymbol{b}_r)$ defines the constraints stored in $\mathcal{P}^{(r)}$ for
     $r \in [p]$
3      $(\boldsymbol{g}_1, \dots, \boldsymbol{g}_p) = O_S(\boldsymbol{x}_1, \dots, \boldsymbol{x}_p)$
4      **return** $\mathsf{Discretize}_{k_j}(\boldsymbol{g}_j, \xi)$
5 **end**
6 Define $O_x : \mathcal{I}_{k_{i+1}} \to \mathbb{R}^{k_j}$ as $\mathsf{ApproxOracle}_{\delta,\xi,\mathcal{O}_f}(i+1, j, \mathcal{P}^{(1)}, , \dots, \mathcal{P}^{(i)}, \cdot)$
7 Define $O_y : \mathcal{I}_{k_{i+1}} \to \mathbb{R}^{k_{i+1}}$ as $\mathsf{ApproxOracle}_{\delta,\xi,\mathcal{O}_f}(i+1, i+1, \mathcal{P}^{(1)}, \dots, \mathcal{P}^{(i)}, \cdot)$
8 **return** $\mathsf{ApproxSeparationVector}_{\delta,\xi}(O_x, O_y)$

**Algorithm 3:** $\mathsf{ApproxOracle}_{\delta,\xi,O_S}(i, j, \mathcal{P}^{(1)}, \dots, \mathcal{P}^{(i)})$

We can then use $\mathsf{ApproxOracle}_{\delta,\xi,O_S}(1, 1, \cdot)$ to solve the original problem with the memory-constrained Vaidya's method. In Appendix A, we show that taking $\delta = \frac{\epsilon}{4d}$ and $\xi = \frac{\sigma_{min}\epsilon}{32d^{5/2}}$ achieves the desired oracle-complexity and memory usage. The final algorithm is given in Algorithm 4.

**Input:** $\delta, \xi$, and $\mathcal{O}_S : \mathcal{C}_d \to \mathbb{R}^d$ a separation oracle
**Check:** Throughout the algorithm, if $O_S$ returned $\mathsf{Success}$ to a query $\boldsymbol{x}$, **return** $\boldsymbol{x}$
1 Run Algorithm 1 with parameters $\delta$ and $\xi$ and oracle $\mathsf{ApproxOracle}_{\delta,\xi,O_S}(1, 1, \cdot)$

**Algorithm 4:** Memory-constrained algorithm for convex optimization

**A geometric illustration of the recursive step.** In Figure 2, we give a 2-dimensional feasibility problem with target $\boldsymbol{p}^* = (p_1^*, p_2^*)$ and two blocks (i.e. $p = 2$) as an illustration of our recursive approach (Algorithm 2) to construct an approximate separating hyperplane for a "reduced" problem.

Suppose at a step of the Algorithm 4, the current value of the $x_1$ coordinate is $c$. We aim to find an approximate separating hyperplane between $x_1 = p_1^*$ and $x_1 = c$. Algorithm 2 first runs Algorithm 1 (i.e. the memory-constrained Vaidya) to find two separating hyperplanes (the two blue hyperplanes). Lemma 4.1 then guarantees the existence of a convex combination of the 2 blue hyperplanes – the

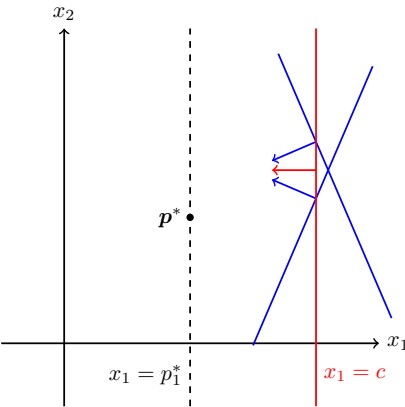

Figure 2: Intuition for the recursive procedure in Algorithm 4. Using the separation hyperplanes (blue) found by Algorithm 1, i.e., the memory-constrained Vaidya, it constructs an approximate separation hyperplane (red) between $x_1 = c$ and the target $x_1 = p_1^*$.

red hyperplane– which is approximately parallel to the $x_2$-axis and thus can serve as an approximate separating hyperplane between $x_1 = p_1^*$ and $x_1 = c$.

**Sketch of proof.** At the high level, the algorithm recursively runs Vaidya's method Algorithm 1 for each level of computation $i \in [p]$. Since each run of Algorithm 4 requires $\mathcal{O}(\frac{d}{p} \ln \frac{1}{\epsilon})$ queries, the total number of calls to the oracle, which is exponential in the number of levels, is $\mathcal{O}(\mathcal{O}(\frac{d}{p} \ln \frac{1}{\epsilon})^p)$. As for the memory usage, the algorithm mainly needs to keep in memory the constraints defining the polyhedrons at each level $i \in [p]$. From Lemma 4.2, each polyhedron only requires $\mathcal{O}(\frac{d}{p})$ constraints that each require $\mathcal{O}(\frac{d}{p} \ln \frac{1}{\epsilon})$ bits of memory. Hence, the total memory needed is $\mathcal{O}(\frac{d^2}{p} \ln \frac{1}{\epsilon})$. The main difficulty lies in showing that the algorithm is successful. To do so, we need to show that the precision in the successive approximated separation oracles Algorithm 2 is sufficient. To avoid an exponential dependence of the approximation error in $p$—which would be prohibitive for the memory usage of our method—each run of Vaidya's method Algorithm 1 is run for more iterations than the precision of the separation vectors would classically allow. To give intuition, if the separation oracle came from a convex optimization subgradient oracle for a function $f$, the iterates at a level $i$ do not converge to the true "minimizer" of $\min_{\boldsymbol{x}_i} f^{(i)}(\boldsymbol{x}_1, \ldots, \boldsymbol{x}_i)$, where $f^{(i)}(\cdot) = \min_{\boldsymbol{x}_{i+1}, \ldots, \boldsymbol{x}_p} f(\cdot, \boldsymbol{x}_{i+1}, \ldots, \boldsymbol{x}_p)$, but instead converge to a close enough point while still providing meaningful approximate subgradients at the higher level $i - 1$ (in Algorithm 2).

## 5 Discussion and Conclusion

To the best of our knowledge, this work is the first to provide some positive trade-off between oracle-complexity and memory-usage for convex optimization or the feasibility problem, as opposed to lower-bound impossibility results [31, 5]. Our trade-offs are more significant in a high accuracy regime: when $\ln \frac{1}{\epsilon} \approx d^c$, for $c > 0$ our trade-offs are polynomial, while the improvements when $\ln \frac{1}{\epsilon} = \text{poly}(\ln d)$ are only in $\ln d$ factors. A natural open direction [52] is whether there exist algorithms with polynomial trade-offs in that case. We also show that in the exponential regime $\ln \frac{1}{\epsilon} \geq \Omega(d \ln d)$, gradient descent is not Pareto-optimal. Instead, one can keep the optimal memory and decrease the dependence in $\epsilon$ of the oracle-complexity from $\frac{1}{\epsilon^2}$ to $(\ln \frac{1}{\epsilon})^d$. The question of whether the exponential dependence in $d$ is necessary is left open. Last, our algorithms rely on the consistency of the oracle, which allows re-computations. While this is a classical assumption, gradient descent and classical cutting-plane methods do not need it; removing this assumption could be an interesting research direction (potentially, this could also yield stronger lower bounds).

## Acknowledgments and Disclosure of Funding

This work was partly funded ONR grant N00014-18-1-2122, AFOSR grant FA9550-19-1-0263, and AFOSR grant FA9550-23-1-0182.

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

# A  Proof of the query complexity and memory usage of Algorithm 4

First, we give simple properties on the discretization functions. One can easily check that for any $\boldsymbol{x} \in C$,

$$\|\boldsymbol{x} - \mathsf{Discretize}_d(\boldsymbol{x}, \xi)\| \leq \xi \qquad \text{and} \qquad \|\mathsf{Discretize}_d(\boldsymbol{x}, \xi)\| \leq \|\boldsymbol{x}\|. \tag{1}$$

Further, one can easily check that to represent any output of $\mathsf{Discretize}_d(\cdot, \xi)$, one needs at most $d \ln \frac{2\sqrt{d}}{\xi} = \mathcal{O}(d \ln \frac{d}{\xi})$ bits.

We next prove Lemma 4.1.

*Proof of Lemma 4.1.* We first consider the case when the algorithm terminates because of a query $\boldsymbol{g} = O(\mathcal{P}_t)$ such that $\|\boldsymbol{g}\| \leq \delta/(2\sqrt{d})$. Then, for any $\boldsymbol{x} \in \mathcal{C}_d$, one directly has

$$\boldsymbol{g}^\top \boldsymbol{x} - b \leq \boldsymbol{g}^\top (\boldsymbol{x} - \boldsymbol{\omega}) \leq 2\sqrt{d}\|\boldsymbol{g}\| \leq \delta.$$

where $\boldsymbol{\omega}$ is the volumetric center of the resulting polyhedron. In the second inequality we used the fact that $\boldsymbol{\omega} \in \mathcal{C}_d$, otherwise the algorithm would not have terminated at that step.

We next turn to the other cases and start by showing that the output polyhedron does not contain a ball of radius $\delta$. This is immediate if the algorithm terminated because the polyhedron was empty. We then suppose this was not the case, and follow the same proof as given in [2]. Algorithm 1 and the one provided in [2] coincide when removing a constraint of the polyhedron. Hence, it suffices to consider the case when we add a constraint. We use the notation $\tilde{\boldsymbol{A}}^\top = [\boldsymbol{A}^\top, \boldsymbol{a}_{m+1}^\top]$, $\tilde{\boldsymbol{b}}^\top = [\boldsymbol{b}^\top, b_{m+1}]$ for the updated matrix $\boldsymbol{A}$ and vector $\boldsymbol{b}$ after adding the constraint. We also denote $\boldsymbol{\omega} = \mathsf{VolumetricCenter}(\boldsymbol{A}, \boldsymbol{b})$ (resp. $\tilde{\boldsymbol{\omega}} = \mathsf{VolumetricCenter}(\tilde{\boldsymbol{A}}, \tilde{\boldsymbol{b}})$) the volumetric center of the polyhedron before (resp. after) adding the constraint. Next, we consider the vector $(\boldsymbol{b}')^\top = [\boldsymbol{b}^\top, \boldsymbol{a}_{m+1}^\top \boldsymbol{\omega}]$, which would have been obtained if the cut was performed at $\boldsymbol{\omega}$ exactly. We then denote $\boldsymbol{\omega}' = \mathsf{VolumetricCenter}(\tilde{\boldsymbol{A}}, \boldsymbol{b}')$. Then proof of [2] shows that

$$V_{\tilde{\boldsymbol{A}}, \boldsymbol{b}'}(\boldsymbol{\omega}') \geq V_{\boldsymbol{A}, \boldsymbol{b}}(\boldsymbol{\omega}) + 0.0340.$$

We now observe that by construction, we have $\tilde{\boldsymbol{b}}_{m+1} \geq \boldsymbol{a}_{m+1}^\top \boldsymbol{\omega}$, so that the polyhedron associated to $(\tilde{\boldsymbol{A}}, \tilde{\boldsymbol{b}})$ is more constrained than the one associated to $(\tilde{\boldsymbol{A}}, \boldsymbol{b}')$. As a result, we have $V_{\tilde{\boldsymbol{A}}, \tilde{\boldsymbol{b}}}(\boldsymbol{x}) \geq V_{\tilde{\boldsymbol{A}}, \boldsymbol{b}'}(\boldsymbol{x})$, for any $\boldsymbol{x} \in \mathbb{R}^d$ such that $\tilde{\boldsymbol{A}}\boldsymbol{x} \geq \tilde{\boldsymbol{b}}$. Therefore,

$$V_{\tilde{\boldsymbol{A}}, \tilde{\boldsymbol{b}}}(\tilde{\boldsymbol{\omega}}) \geq V_{\tilde{\boldsymbol{A}}, \boldsymbol{b}'}(\tilde{\boldsymbol{\omega}}) \geq V_{\tilde{\boldsymbol{A}}, \boldsymbol{b}'}(\boldsymbol{\omega}') \geq V_{\boldsymbol{A}, \boldsymbol{b}}(\boldsymbol{\omega}) + 0.0340.$$

This ends the modifications in the proof of [2]. With the notations of this paper, we still have $\Delta V^+ = 0.340$ and $\Delta V^- = 0.326$, so that $\Delta V = 0.0014$. Then, because $c = \frac{1}{\Delta V}$, the same proof shows that the procedure is successful for precision $\delta$: the final polyhedron $(\boldsymbol{A}, \tilde{\boldsymbol{b}})$ returned by Algorithm 1 does not contains a ball of radius $> \delta$. As a result, whether the algorithm performed all $T_{max}$ iterations or not, $\{\boldsymbol{x} : \boldsymbol{A}\boldsymbol{x} \geq \boldsymbol{b}\}$ does not contain a ball of radius $> \delta'$, where $\boldsymbol{A}$ and $\boldsymbol{b}$ define the constraints stored in the output $\mathcal{P}$. Now letting $m$ be the objective value of the right optimization problem, there exists $\boldsymbol{x} \in \mathcal{C}_d$ such that for all $t \leq T$, $\boldsymbol{g}_t^\top (\boldsymbol{x} - \boldsymbol{c}_t) \geq m$. Therefore, for any $\boldsymbol{x}' \in B_d(\boldsymbol{x}, m)$ one has

$$\forall i \in [m], \boldsymbol{a}_i^\top \boldsymbol{x}' - b_i \geq m + \boldsymbol{a}_t^\top (\boldsymbol{x}' - \boldsymbol{x}) \geq m - \|\boldsymbol{x}' - \boldsymbol{x}\| \geq 0.$$

In the last inequality we used $\|\boldsymbol{a}_t\| \leq 1$. This implies that the polyhedron contains $B_d(\boldsymbol{x}, m)$. Hence, $m \leq \delta$.

This ends the proof of the right inequality. The left equality is a direct application of strong duality for linear programming. $\qquad \square$

We now prove that Algorithm 4 has the desired oracle-complexity and memory usage.

We first describe the recursive calls of Algorithm 3 in more detail. To do so, consider running the procedure $\mathsf{ApproxOracle}(i, j, \mathcal{P}^{(1)}, \ldots, \mathcal{P}^{(i)})$ where $i < p$, which corresponds to running Algorithm 2 for specific oracles. We say that this is a level-$i$ run. Then, the algorithm performs at most $2T(\delta, k_{i+1})$ calls to $\mathsf{ApproxOracle}(i + 1, i + 1, \mathcal{P}^{(1)}, \ldots, \mathcal{P}^{(i)}, \cdot)$, where the factor 2 comes from the fact that

Vaidya's method Algorithm 1 is effectively run twice in Algorithm 2. The solution to $(\mathcal{P}_{aux}(\mathcal{P}))$ has as many components as constraints in the last polyhedron, which is at most $\frac{k_{i+1}}{\sigma_{min}} + 1$ by Lemma 4.2. Hence, the number of calls to $\mathsf{ApproxOracle}(i+1, j, \mathcal{P}^{(1)}, \ldots, \mathcal{P}^{(i)}, \cdot)$ is at most $\frac{k_{i+1}}{\sigma_{min}} + 1$. In total, that is $\mathcal{O}(k_{i+1} \ln \frac{1}{\delta})$ calls to the level $i+1$ of the recursion.

We next aim to understand the output of running $\mathsf{ApproxOracle}(1, 1, \mathcal{P}^{(1)})$. We denote by $\boldsymbol{\lambda}(\mathcal{P}^{(1)})$ the solution $\mathcal{P}_{aux}(\mathcal{P}^{\star})$ computed at l.2 of the first call to Algorithm 2, where $\mathcal{P}^{\star}$ is the output polyhedron of the first call to Algorithm 1. Denote by $\mathcal{S}(\mathcal{P}^{(1)})$ the set of indices of coordinates from $\boldsymbol{\lambda}(\mathcal{P}^{(1)})$ for which the procedure performed a call to $\mathsf{ApproxOracle}(2, 1, \mathcal{P}^{(1)}, \cdot)$. In other words, $\mathcal{S}(\mathcal{P}^{(1)})$ contains the indices of all coordinates of $\boldsymbol{\lambda}(\mathcal{P}^{(1)})$, except those for which the corresponding query lay outside of the unit cube, or the initial constraints of the cube. For any index $l \in \mathcal{S}(\mathcal{P}^{(1)})$, let $\mathcal{P}_l^{(2)}$ denote the state of the current polyhedron ($\mathcal{P}_t$ in l.7 of Algorithm 2) when that call was performed. Up to discretization issues, the output of the complete procedure is

$$\sum_{l \in \mathcal{S}(\mathcal{P}^{(1)})} \lambda_l(\mathcal{P}^{(1)}) \mathsf{ApproxOracle}(2, 1, \mathcal{P}^{(1)}, \mathcal{P}_l^{(2)}).$$

We continue in the recursion, defining $\boldsymbol{\lambda}(\mathcal{P}^{(1)}, \mathcal{P}_l^{(2)})$ and $\mathcal{S}(\mathcal{P}^{(1)}, \mathcal{P}_l^{(2)})$ for all $l \in \mathcal{S}(\mathcal{P}^{(1)})$, until we define all vectors of the form $\boldsymbol{\lambda}(\mathcal{P}^{(1)}, \mathcal{P}_{l_2}^{(2)}, \ldots, \mathcal{P}_{l_r}^{(r)})$ and sets of the form $\mathcal{S}(\mathcal{P}^{(1)}, \mathcal{P}_{l_2}^{(2)}, \ldots, \mathcal{P}_{l_r}^{(r)})$ for $i+1 \le r \le p-1$. To simplify the notation and emphasize that all these polyhedra depend on the recursive computation path, we adopt the notation

$$\lambda^{l_2, \ldots, l_{r+1}} := \lambda_{l_{r+1}}(\mathcal{P}^{(1)}, \mathcal{P}_{l_2}^{(2)}, \ldots, \mathcal{P}_{l_r}^{(r)})$$

$$\mathcal{S}^{l_2, \ldots, l_r} := \mathcal{S}(\mathcal{P}^{(1)}, \mathcal{P}_{l_2}^{(2)}, \ldots, \mathcal{P}_{l_r}^{(r)})$$

We recall that these polyhedron are kept in memory to query their volumetric center. For ease of notation, we write $\boldsymbol{x}_1 = \mathsf{VolumetricCenter}(\mathcal{P}^{(1)})$, and we write $\boldsymbol{c}^{l_2, \ldots, l_r} = \mathsf{VolumetricCenter}(\mathcal{P}_{l_r}^{(r)})$ for $2 \le r \le p$, where $l_2, \ldots, l_{r-1}$ were the indices from the computation path leading up to $\mathcal{P}_{l_r}^{(r)}$. Last, we write $O_S = (O_{S,1}, \ldots, O_{S,p})$, where $O_{S,i} : \mathcal{C}_d \to \mathbb{R}^{k_i}$ is the "$\boldsymbol{x}_i$" component of $O_S$, for all $i \in [p]$.

With all these notations, we will show that the output of $\mathsf{ApproxOracle}(i, j, \mathcal{P}^{(1)}, \mathcal{P}_{l_2}^{(2)}, \ldots, \mathcal{P}_{l_i}^{(i)})$ is approximately equal to the vector

$$G(i, j, \boldsymbol{x}_1, \boldsymbol{c}^{l_2}, \ldots, \boldsymbol{c}^{l_2, \ldots, l_i})$$

$$:= \sum_{\substack{l_{i+1} \in \mathcal{S}, \, l_{i+2} \in \mathcal{S}^{l_{i+1}}, \\ \ldots, \, l_p \in \mathcal{S}^{l_{i+1}, \ldots, l_{p-1}}}} \lambda^{l_{i+1}} \lambda^{l_{i+1}, l_{i+2}} \cdots \lambda^{l_{i+1}, \ldots, l_p} \cdot O_{S,j}(\boldsymbol{x}_1, \boldsymbol{c}^{l_2}, \ldots, \boldsymbol{c}^{l_2, \ldots, l_p}),$$

with the convention that for $i = p$,

$$G(p, j, \boldsymbol{x}_1, \boldsymbol{c}^{l_2}, \ldots, \boldsymbol{c}^{l_2, \ldots, l_p}) := O_{S,j}(\boldsymbol{x}_1, \boldsymbol{c}^{l_2}, \ldots, \boldsymbol{c}^{l_2, \ldots, l_p}).$$

The corresponding computation tree is represented in Fig. 3. For convenience, we omitted the term $j = 1$.

We start the analysis with a simple result showing that if the oracle $O_S$ returns separation vectors of norm bounded by one, then the responses from $\mathsf{ApproxOracle}$ also lie in the unit ball.

**Lemma A.1.** *Fix $\delta, \xi \in (0, 1)$, $1 \le j \le i \le p$ and an oracle $O_S = (O_{S,1}, \ldots, O_{S,p}) : \mathcal{C}_d \to \mathbb{R}^d$. Suppose that $O_S$ takes values in the unit ball. For any $s \in [i]$ let $\mathcal{P}_{l_s}^{(s)} \in \mathcal{I}_{k_s}$ represent a bounded polyhedrons with $\mathsf{VolumetricCenter}(\mathcal{P}_{l_s}^{(s)}) \in \mathcal{C}_{k_s}$. Then, one has*

$$\|\mathsf{ApproxOracle}_{\delta, \xi, O_S}(i, j, \mathcal{P}_{l_1}^{(1)}, \ldots, \mathcal{P}_{l_i}^{(i)})\| \le 1.$$

*Proof.* We prove this by simple induction on $i$. For convenience, we define the point $\boldsymbol{x}_k = \mathsf{VolumetricCenter}(\mathcal{P}_{l_k}^{(k)})$. If $i = p$, we have

$$\|\mathsf{ApproxOracle}_{\delta, \xi, O_S}(i, j, \mathcal{P}_{l_1}^{(1)}, \ldots, \mathcal{P}_{l_i}^{(i)})\| = \|\mathsf{Discretize}_{k_j}(O_{S,j}(\boldsymbol{x}_1, \ldots, \boldsymbol{x}_p), \xi)\|$$

$$\le \|O_{S,j}(\boldsymbol{x}_1, \ldots, \boldsymbol{x}_p)\| \le 1,$$

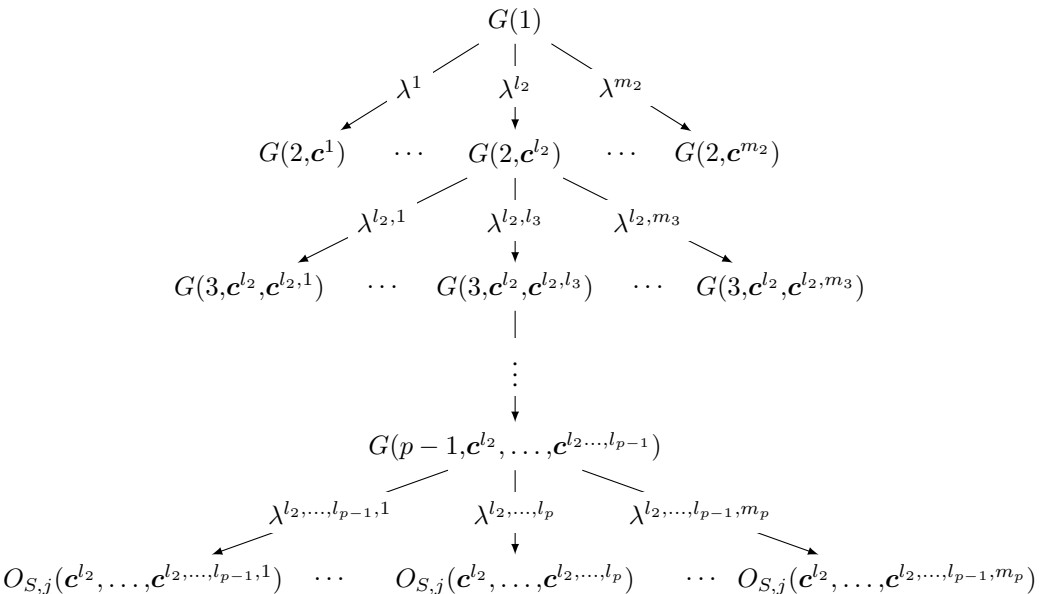

Figure 3: Computation tree representing the recursive calls to ApproxOracle starting from the calls to ApproxOracle$(1, 1, \cdot)$ from Algorithm 4

where in the first inequality we used Eq (1) and in the second inequality we used the fact that $O_S(\boldsymbol{x}_1, \ldots, \boldsymbol{x}_p)$ has norm at most one. Now suppose that the result holds for $i + 1 \leq p$. Then by construction, the output ApproxOracle$_{\delta,\xi,O_S}(i, j, \mathcal{P}_{l_1}^{(1)}, \ldots, \mathcal{P}_{l_i}^{(i)})$ is the result of iterative discretizations. Using Eq (1) and the previously defined notations, we obtain

$$\|\text{ApproxOracle}_{\delta,\xi,O_S}(i, j, \mathcal{P}_{l_1}^{(1)}, \ldots, \mathcal{P}_{l_i}^{(i)})\|$$

$$\leq \left\| \sum_{l_{i+1} \in \mathcal{S}^{l_1,\ldots,l_i}} \lambda^{l_2,\ldots,l_i} \text{ApproxOracle}_{\delta,\xi,O_S}(i+1, j, \mathcal{P}_{l_1}^{(1)}, \ldots, \mathcal{P}_{l_i}^{(i)}, \mathcal{P}_{l_{i+1}}^{(i+1)}) \right\| \leq 1.$$

In the last inequality, we used the induction hypothesis together with the fact that $\sum_{l_{i+1}} \lambda^{l_2,\ldots,l_{i+1}} \leq 1$ using Eq (1). This ends the induction and the proof. $\qquad\square$

We are now ready to compare the output of Algorithm 3 to $G(i, j, \boldsymbol{x}_1, \boldsymbol{c}^{l_2}, \ldots, \boldsymbol{c}^{l_2,\ldots,l_i})$.

**Lemma A.2.** *Fix* $\delta, \xi \in (0, 1)$, $1 \leq j \leq i \leq p$ *and an oracle* $O_S = (O_{S,1}, \ldots, O_{S,p}) : \mathcal{C}_d \to \mathbb{R}^d$. *Suppose that* $O_S$ *takes values in the unit ball. For any* $s \in [i]$ *let* $\mathcal{P}_{l_s}^{(s)} \in \mathcal{I}_{k_s}$ *represent a bounded polyhedron with* VolumetricCenter$(\mathcal{P}_{l_s}^{(s)}) \in \mathcal{C}_{k_s}$. *Denote* $\boldsymbol{x}_r = \boldsymbol{c}(\mathcal{P}_{l_r}^{(r)})$ *for* $r \in [i]$. *Then,*

$$\|\text{ApproxOracle}_{\delta,\xi,O_S}(i, j, \mathcal{P}_{l_1}^{(1)}, \ldots, \mathcal{P}_{l_i}^{(i)}) - G(i, j, \boldsymbol{x}_1, \ldots, \boldsymbol{x}_i)\| \leq \frac{4}{\sigma_{min}} d\xi.$$

*Proof.* We prove by simple induction on $i$ that

$$\|\text{ApproxOracle}_{\delta,\xi,O_S}(i, j, \mathcal{P}_{l_1}^{(1)}, \ldots, \mathcal{P}_{l_i}^{(i)}) - G(i, j, \boldsymbol{x}_1, \ldots, \boldsymbol{x}_i)\|$$

$$\leq \left(1 + \frac{2}{\sigma_{min}}(k_{i+1} + \ldots + k_p) + 2(p - i)\right)\xi.$$

First, for $i = p$, the result is immediate since the discretization is with precision $\xi$ (1.4 of Algorithm 3). Now suppose that this is the case for $i \leq p$ and any valid values of other parameters. For conciseness, we write $\boldsymbol{G} = (\mathcal{P}_{l_1}^{(1)}, \ldots, \mathcal{P}_{l_{i-1}}^{(i-1)})$. Next, recall that by Lemma 4.2, $|\mathcal{S}^{l_2,\ldots,l_{i-1}}| \leq \frac{k_i}{\sigma_{min}} + 1$. Hence,

the discretizations due to l.8 of Algorithm 2 can affect the estimate for at most that number of rounds. Then, we have

$$\left\| \mathsf{ApproxOracle}_{\delta,\xi,O_S}(i-1,j,\boldsymbol{G}) - \sum_{l_i \in \mathcal{S}^{l_2,\ldots,l_{i-1}}} \tilde{\lambda}^{l_2,\ldots,l_i} \mathsf{ApproxOracle}_{\delta,\xi,O_S}(i,j,\boldsymbol{G},\mathcal{P}_{l_i}^{(i)}) \right\|$$

$$\leq \left( \frac{k_i}{\sigma_{min}} + 1 \right) \xi,$$

where $\tilde{\lambda}^{l_2,\ldots,l_i}$ are the discretized coefficients that are used during the computation l.8 of Algorithm 2. Now using Lemma A.1, we have

$$\left\| \sum_{l_i \in \mathcal{S}^{l_2,\ldots,l_{i-1}}} (\tilde{\lambda}^{l_2,\ldots,l_i} - \lambda^{l_2,\ldots,l_i}) \mathsf{ApproxOracle}_{\delta,\xi,O_S}(i,j,\boldsymbol{G},\mathcal{P}_{l_i}^{(i)}) \right\|$$

$$\leq \|\tilde{\boldsymbol{\lambda}}^{l_{i+1},\ldots,l_{i-1}} - \boldsymbol{\lambda}^{l_{i+1},\ldots,l_{i-1}}\|_1 \leq \left( \frac{k_i}{\sigma_{min}} + 1 \right) \xi.$$

In the last inequality we used the fact that $\boldsymbol{\lambda}$ has at most $\frac{k_i}{\sigma_{min}} + 1$ non-zero coefficients. As a result, using the induction for each term of the sum, and the fact that $\sum_{l_i} \lambda^{l_2,\ldots,l_i} \leq 1$, we obtain

$$\|\mathsf{ApproxOracle}_{\delta,\xi,\mathcal{O}_f}(i-1,j,\boldsymbol{G}) - G(i-1,j,\boldsymbol{x}_1,\ldots,\boldsymbol{x}_{i-1})\|$$

$$\leq \left( 1 + \frac{2}{\sigma_{min}}(k_{i+1} + \ldots + k_p) + 2(p-i) \right) \xi + \left( \frac{2k_i}{\sigma_{min}} + 2 \right) \xi,$$

which completes the induction. Noting that $k_{i+1} + \ldots + k_p \leq k_1 + \ldots + k_p \leq d$ and $p - i \leq d - 1$ ends the proof. □

Next, we show that the outputs of Algorithm 3 provide approximate separation hyperplanes for the first $i$ coordinates $(\boldsymbol{x}_1,\ldots,\boldsymbol{x}_i)$.

**Lemma A.3.** *Fix $\delta,\xi \in (0,1)$, $1 \leq j \leq i \leq p$ and an oracle $O_S = (O_{S,1},\ldots,O_{S,p}) : \mathcal{C}_d \to \mathbb{R}^d$ for accuracy $\epsilon > 0$. Suppose that $O_S$ takes values in the unit ball $B_d(0,1)$. For any $s \in [i]$ let $\mathcal{P}_{l_s}^{(s)} \in \mathcal{I}_{k_s}$ represent a bounded polyhedron with $\mathsf{VolumetricCenter}(\mathcal{P}_{l_s}^{(s)}) \in \mathcal{C}_{k_s}$. Denote $\boldsymbol{x}_r = \boldsymbol{c}(\mathcal{P}_{l_r}^{(r)})$ for $r \in [i]$. Suppose that when running $\mathsf{ApproxOracle}_{\delta,\xi,O_S}(i,i,\mathcal{P}_{l_1}^{(1)},\ldots,\mathcal{P}_{l_i}^{(i)})$, no successful vector was queried. Then, any vector $\boldsymbol{x}^\star = (\boldsymbol{x}_1^\star,\ldots,\boldsymbol{x}_p^\star) \in \mathcal{C}_d$ such that $B_d(\boldsymbol{x}^\star,\epsilon)$ is contained in the successful set satisfies*

$$\sum_{r \in [i]} \mathsf{ApproxOracle}_{\delta,\xi,O_S}(i,r,\mathcal{P}_{l_1}^{(1)},\ldots,\mathcal{P}_{l_i}^{(i)})^\top (\boldsymbol{x}_r^\star - \boldsymbol{x}_r) \geq \epsilon - \frac{8d^{5/2}}{\sigma_{min}}\xi - d\delta.$$

*Proof.* For $i \leq r \leq p$ and $j \leq r$, we use the notation

$$\boldsymbol{g}_j^{l_{i+1},\ldots,l_r} = \mathsf{ApproxOracle}_{\delta,\xi,O_S}(r,j,\mathcal{P}_{l_1}^{(1)},\ldots,\mathcal{P}_{l_r}^{(r)}).$$

Using Lemma A.2, we always have for $j \in [r]$,

$$\|\boldsymbol{g}_j^{l_{i+1},\ldots,l_r} - G(r,j,\boldsymbol{x}_1,\ldots,\boldsymbol{x}_i,\boldsymbol{c}^{l_{i+1}},\ldots,\boldsymbol{c}^{l_{i+1},\ldots,l_r})\| \leq \frac{4d}{\sigma_{min}}\xi. \qquad (2)$$

Also, observe that by Lemma A.1 the recursive outputs of $\mathsf{ApproxOracle}$ always have norm bounded by one.

Next, let $\mathcal{T}^{l_{i+1},\ldots,l_{r-1}}$ be the set of indices corresponding to coordinates of $\boldsymbol{\lambda}^{l_{i+1},\ldots,l_{r-1}}$ for which the procedure $\mathsf{ApproxOracle}$ did not call for a level-$r$ computation. These correspond to 1. constraints from the initial cube $\mathcal{P}_0$, or 2. cases when the volumetric center was out of the unit cube (l.6-7 of Algorithm 1) and as a result, the index of the added constraint was $-1$ instead of the current iteration index $t$. Similarly as above, for any $t \in \mathcal{T}^{l_{i+1},\ldots,l_{r-1}}$, we denote by $\boldsymbol{g}_r^{l_{i+1},\ldots,l_{r-1},t}$ the corresponding

vector $\boldsymbol{a}_t$. We recall that by construction, this vector is of the form $\pm \boldsymbol{e}_j$ for some $j \in [k_r]$. Then, from Lemma 4.1, since the responses of the oracle always have norm bounded by one, for all $\boldsymbol{y}_r \in \mathcal{C}_{k_r}$,

$$\sum_{l_r \in \mathcal{S}^{l_{i+1},\ldots,l_{r-1}} \cup \mathcal{T}^{l_{i+1},\ldots,l_{r-1}}} \lambda^{l_{i+1},\ldots,l_r} (\boldsymbol{g}_r^{l_{i+1},\ldots,l_r})^\top (\boldsymbol{y}_r - \boldsymbol{c}^{l_{i+1},\ldots,l_r}) \leq \delta. \tag{3}$$

For conciseness, we use the shorthand $(\mathcal{S} \cup \mathcal{T})^{l_{i+1},\ldots,l_{r-1}} := \mathcal{S}^{l_{i+1},\ldots,l_{r-1}} \cup \mathcal{T}^{l_{i+1},\ldots,l_{r-1}}$, which contains all indices from coordinates of $\boldsymbol{\lambda}^{l_{i+1},\ldots,l_{r-1}}$. In particular,

$$\sum_{l_r \in (\mathcal{S} \cup \mathcal{T})^{l_{i+1},\ldots,l_{r-1}}} \lambda^{l_{i+1},\ldots,l_r} = 1. \tag{4}$$

We now proceed to estimate the precision of the vectors $G(i, j, \boldsymbol{x}_1, \ldots, \boldsymbol{x}_i)$ as approximate separation hyperplanes for coordinates $(\boldsymbol{x}_1, \ldots, \boldsymbol{x}_i)$. Let $\boldsymbol{x}^\star \in \mathcal{C}_d$ such that $B_d(\boldsymbol{x}^\star, \epsilon)$ is within the successful set. Then, for any choice of $l_{i+1} \in \mathcal{S}, \ldots, l_p \in \mathcal{S}^{l_{i+1},\ldots,l_{p-1}}$, since we did not query a successful vector, we have for all $\boldsymbol{z} \in B_d(\boldsymbol{x}^\star, \epsilon)$,

$$O_S(\boldsymbol{x}_1, \ldots, \boldsymbol{x}_i, \boldsymbol{c}^{l_{i+1}}, \ldots, \boldsymbol{c}^{l_{i+1},\ldots,l_p})^\top (\boldsymbol{z} - (\boldsymbol{x}_1, \ldots, \boldsymbol{x}_i, \boldsymbol{c}^{l_{i+1}}, \ldots, \boldsymbol{c}^{l_{i+1},\ldots,l_p})) \geq 0.$$

As a result, because the responses from $O_S$ have unit norm,

$$O_S(\boldsymbol{x}_1, \ldots, \boldsymbol{x}_i, \boldsymbol{c}^{l_{i+1}}, \ldots, \boldsymbol{c}^{l_{i+1},\ldots,l_p})^\top (\boldsymbol{x}^\star - (\boldsymbol{x}_1, \ldots, \boldsymbol{x}_i, \boldsymbol{c}^{l_{i+1}}, \ldots, \boldsymbol{c}^{l_{i+1},\ldots,l_p})) \geq \epsilon. \tag{5}$$

Now write $\boldsymbol{x}^\star = (\boldsymbol{x}_1^\star, \ldots, \boldsymbol{x}_p^\star)$. In addition to the previous equation, for $l_{i+1} \in \mathcal{S}, \ldots, l_{r-1} \in \mathcal{S}^{l_{i+1},\ldots,l_{r-2}}$ and any $l_r \in \mathcal{T}^{l_{i+1},\ldots,l_{r-1}}$, one has $(\boldsymbol{g}_r^{l_{i+1},\ldots,l_r})^\top \boldsymbol{x}_r^\star + 1 \geq \epsilon$, because $\boldsymbol{x}^\star$ is within the cube $\mathcal{C}_d$ and at least at distance $\epsilon$ from the constraints of the cube. Similarly as when $l_r \in \mathcal{S}^{l_{i+1},\ldots,l_{r-1}}$, for any $l_r \in \mathcal{T}^{l_{i+1},\ldots,l_{r-1}}$ we denote by $\boldsymbol{c}^{l_{i+1},\ldots,l_r}$ the volumetric center of the polyhedron $\mathcal{P}_{l_r}^{(r)}$ along the corresponding computation path, if $l_r$ corresponded to an added constraints when $\boldsymbol{c}^{l_{i+1},\ldots,l_r} \notin \mathcal{C}_{k_r}$. Otherwise, if $l_r$ corresponded to the constraint $\boldsymbol{a} = \pm \boldsymbol{e}_j$ of the initial cube, we pose $\boldsymbol{c}^{l_{i+1},\ldots,l_r} = -\boldsymbol{a}$. Now by construction, in both cases one has $(\boldsymbol{g}_r^{l_{i+1},\ldots,l_r})^\top \boldsymbol{c}^{l_{i+1},\ldots,l_r} \leq -1$ (l.7 of Algorithm 1). Thus,

$$(\boldsymbol{g}_r^{l_{i+1},\ldots,l_r})^\top (\boldsymbol{x}_r^\star - \boldsymbol{c}^{l_{i+1},\ldots,l_r}) \geq \epsilon. \tag{6}$$

Recalling Eq (4), we then sum all equations of the form Eq (5) and Eq (6) along the computation path, to obtain

$$(A) := \sum_{\substack{l_{i+1} \in \mathcal{S},\ldots, \\ l_p \in \mathcal{S}^{l_{i+1},\ldots,l_{p-1}}}} \lambda^{l_{i+1}} \ldots \lambda^{l_{i+1},\ldots,l_p}$$

$$\cdot O_S(\boldsymbol{x}_1, \ldots, \boldsymbol{x}_i, \boldsymbol{c}^{l_{i+1}}, \ldots, \boldsymbol{c}^{l_{i+1},\ldots,l_p})^\top (\boldsymbol{x}^\star - (\boldsymbol{x}_1, \ldots, \boldsymbol{x}_i, \boldsymbol{c}^{l_{i+1}}, \ldots, \boldsymbol{c}^{l_{i+1},\ldots,l_p}))$$

$$+ \sum_{i+1 \leq r \leq p} \sum_{\substack{l_{i+1} \in \mathcal{S},\ldots,l_{r-1} \in \mathcal{S}^{l_{i+1},\ldots,l_{r-2}}, \\ l_r \in \mathcal{T}^{l_{i+1},\ldots,l_{r-1}}}} \lambda^{l_{i+1}} \ldots \lambda^{l_{i+1},\ldots,l_r} \cdot (\boldsymbol{g}_r^{l_{i+1},\ldots,l_r})^\top (\boldsymbol{x}_r^\star - \boldsymbol{c}^{l_{i+1},\ldots,l_r}) \geq \epsilon.$$

Now using the convention

$$G(r, r, \boldsymbol{x}_1, \ldots, \boldsymbol{x}_i, \boldsymbol{c}^{l_{i+1}}, \ldots, \boldsymbol{c}^{l_{i+1},\ldots,l_r}) := \boldsymbol{g}_r^{l_{i+1},\ldots,l_r}, \qquad l_r \in \mathcal{T}^{l_{i+1},\ldots,l_{r-1}},$$

for any $l_{i+1} \in \mathcal{S}, \ldots, l_{r-1} \in \mathcal{S}^{l_{i+1},\ldots,l_{r-2}}$, we can write

$$(A) = \sum_{r \leq i} G(i, r, \boldsymbol{x}_1, \ldots, \boldsymbol{x}_i)^\top (\boldsymbol{x}_r^\star - \boldsymbol{x}_r) + \sum_{i+1 \leq r \leq p} \sum_{\substack{l_{i+1} \in \mathcal{S},\ldots, \\ l_{r-1} \in \mathcal{S}^{l_{i+1},\ldots,l_{r-2}}}} \lambda^{l_{i+1}} \ldots \lambda^{l_{i+1},\ldots,l_{r-1}}$$

$$\times \sum_{l_r \in (\mathcal{S} \cup \mathcal{T})^{l_{i+1},\ldots,l_{r-1}}} \lambda^{l_{i+1},\ldots,l_r} G(r, r, \boldsymbol{x}_1, \ldots, \boldsymbol{x}_i, \boldsymbol{c}^{l_{i+1}}, \ldots, \boldsymbol{c}^{l_{i+1},\ldots,l_r})^\top (\boldsymbol{x}_r^\star - \boldsymbol{c}^{l_{i+1},\ldots,l_r}).$$

We next relate the terms $G$ to the output of ApproxOracle. For simplicity, let us write $\boldsymbol{G} = (\mathcal{P}_{l_1}^{(1)}, \ldots, \mathcal{P}_{l_i}^{(i)})$, which by abuse of notation was assimilated to $(\boldsymbol{x}_1, \ldots, \boldsymbol{x}_i)$. Recall that by construction and hypothesis, all points where the oracle was queried belong to $\mathcal{C}_d$, so that for instance

$\|\boldsymbol{x}_r^\star - \boldsymbol{c}^{l_{i+1},\dots,l_r}\| \leq 2\sqrt{k_r} \leq 2\sqrt{d}$ for any $l_r \in \mathcal{S}^{l_{i+1},\dots,l_{r-1}}$. Using the above equations together with Eq (2) and Lemma A.2 gives

$$\epsilon \leq \sum_{r \leq i} \left[ \mathsf{ApproxOracle}_{\delta,\xi,\mathcal{O}_f}(i,r,\boldsymbol{G})^\top(\boldsymbol{x}_r^\star - \boldsymbol{x}_r) + \frac{8d^{3/2}}{\sigma_{min}}\xi \right] + \sum_{\substack{i+1 \leq r \leq p}} \sum_{\substack{l_{i+1} \in \mathcal{S},\dots, \\ l_{r-1} \in \mathcal{S}^{l_{i+1},\dots,l_{r-2}}}}$$

$$\lambda^{l_{i+1}} \cdots \lambda^{l_{i+1},\dots,l_{r-1}} \sum_{l_r \in (\mathcal{S} \cup \mathcal{T})^{l_{i+1},\dots,l_{r-1}}} \lambda^{l_{i+1},\dots,l_r} \left[ (\boldsymbol{g}_r^{l_{i+1},\dots,l_r})^\top(\boldsymbol{x}_r^\star - \boldsymbol{c}^{l_{i+1},\dots,l_r}) + \frac{8d^{3/2}}{\sigma_{min}}\xi \right]$$

$$\leq \frac{8pd^{3/2}}{\sigma_{min}}\xi + (p-i)\delta + \sum_{r \leq i} \mathsf{ApproxOracle}_{\delta,\xi,\mathcal{O}_f}(i,r,\boldsymbol{G})^\top(\boldsymbol{x}_r^\star - \boldsymbol{x}_r)$$

where in the second inequality, we used Eq (3). Using $p \leq d$, this ends the proof of the lemma. $\qquad\square$

We are now ready to show that Algorithm 4 is a valid algorithm for convex optimization.

**Theorem A.1.** *Let $\epsilon \in (0,1)$ and $O_S : \mathcal{C}_d \to \mathbb{R}^d$ be a separation oracle such that the successful set contains a ball of radius $\epsilon$. Pose $\delta = \frac{\epsilon}{4d}$ and $\xi = \frac{\sigma_{min}\epsilon}{32d^{5/2}}$. Next, let $p \geq 1$ and $k_1,\dots,k_p \leq \lceil \frac{d}{p} \rceil$ such that $k_1 + \dots + k_p = d$. With these parameters, Algorithm 4 finds a successful vector with $(C\frac{d}{p}\ln\frac{d}{\epsilon})^p$ queries and using memory $\mathcal{O}(\frac{d^2}{p}\ln\frac{d}{\epsilon})$, for some universal constant $C > 0$.*

*Proof.* Suppose by contradiction that Algorithm 4 never queried a successful point. Then, with the chosen parameters, Lemma A.3 shows that, for any vector $\boldsymbol{x}^\star = (\boldsymbol{x}_1^\star,\dots,\boldsymbol{x}_p^\star)$ such that $B_d(\boldsymbol{x}^\star,\epsilon)$ is within the successful set, with the same notations, one has

$$\sum_{r \leq i} \mathsf{ApproxOracle}_{\delta,\xi,O_S}(i,r,\mathcal{P}_{l_1}^{(1)},\dots,\mathcal{P}_{l_i}^{(i)})^\top(\boldsymbol{x}_r^\star - \boldsymbol{x}_r) \geq \epsilon - \frac{8d^{5/2}}{\sigma_{min}}\xi - d\delta \geq \frac{\epsilon}{2}.$$

Now denote by $(\boldsymbol{a}_t, b_t)$ the constraints that were added at any time during the run of Algorithm 1 when using the oracle ApproxOracle with $i = j = 1$. The previous equation shows that for all such constraints,

$$\boldsymbol{a}_t^\top \boldsymbol{x}_1^\star - b_t \geq \boldsymbol{a}_t^\top(\boldsymbol{x}_1^\star - \omega_t) - \xi \geq \frac{\epsilon}{2} - \xi,$$

where $\omega_t$ is the volumetric center of the polyhedron at time $t$ during Vaidya's method Algorithm 1. Now, since the algorithm terminated, by Lemma 4.1, we have that

$$\min_t(\boldsymbol{a}_t^\top \boldsymbol{x}_1^\star - b_t) \leq \delta.$$

This is absurd since $\delta + \xi < \frac{\epsilon}{2}$. This ends the proof that Algorithm 4 finds a successful vector.

We now estimate its oracle-complexity and memory usage. First, recall that a run of ApproxOracle of level $i$ makes $\mathcal{O}(k_{i+1}\ln\frac{1}{\delta})$ calls to level-$(i+1)$ runs of ApproxOracle. As a result, the oracle-complexity $Q_d(\epsilon; k_1,\dots,k_p)$ satisfies

$$Q_d(\epsilon; k_1,\dots,k_p) = \left( Ck_1\ln\frac{1}{\delta} \right) \times \dots \times \left( Ck_p\ln\frac{1}{\delta} \right) \leq \left( C'\frac{d}{p}\log\frac{d}{\epsilon} \right)^p$$

for some universal constants $C, C' \geq 2$.

We now turn to the memory of the algorithm. For each level $i \in [p]$ of runs for ApproxOracle, we keep memory placements for

1. the value $j^{(i)}$ of the corresponding call to $\mathsf{ApproxOracle}(i, j^{(i)}, \cdot)$ (for l.6-7 of Algorithm 3): $\mathcal{O}(\ln d)$ bits,

2. the iteration number $t^{(i)}$ during the run of Algorithm 1 or within Algorithm 2: $\mathcal{O}(\ln(k_i \ln\frac{1}{\delta}))$ bits

3. the polyhedron constraints contained in the state of $\mathcal{P}^{(i)}$: $\mathcal{O}(k_i \times k_i \ln\frac{1}{\xi})$ bits,

Table 1: Memory structure for Algorithm 4

| $i$ | 1 | $\ldots$ | $p$ |
|---|---|---|---|
| $j$ | $j^{(1)}$ | | $j^{(p)}$ |
| Iteration index | $t^{(1)}$ | | $t^{(p)}$ |
| Polyhedron | $\mathcal{P}^{(1)} = \begin{pmatrix} k_1, \boldsymbol{a}_1, b_1 \\ k_2, \boldsymbol{a}_2, b_2 \\ \ldots \\ k_m, \boldsymbol{a}_m, b_m \end{pmatrix}$ | | $\mathcal{P}^{(p)}$ |
| Computed dual variables | $(\boldsymbol{k}^{\star(1)}, \boldsymbol{\lambda}^{\star(1)}) = \begin{pmatrix} k_1^\star, \lambda_1^\star \\ k_2^\star, \lambda_2^\star \\ \ldots \end{pmatrix}$ | | $(\boldsymbol{k}^{\star(p)}, \boldsymbol{\lambda}^{\star(p)})$ |
| Working separation vector | $\boldsymbol{u}^{(1)}$ | | $\boldsymbol{u}^{(p)}$ |

4. potentially, already computed dual variables $\boldsymbol{\lambda}^\star$ and their corresponding vector of constraint indices $\boldsymbol{k}^\star$ (l.3 of Algorithm 2): $\mathcal{O}(k_i \times \ln \frac{1}{\xi})$ bits,

5. the working vector $\boldsymbol{u}^{(i)}$ (updated l.8 of Algorithm 2): $\mathcal{O}(k_i \ln \frac{1}{\xi})$ bits.

The memory structure is summarized in Table 1.

We can then check that this memory is sufficient to run Algorithm 4. An important point is that for any run of $\mathsf{ApproxOracle}(i, j, \cdot)$, in Algorithm 2, after running Vaidya's method Algorithm 1 and storing the dual variables $\boldsymbol{\lambda}^\star$ and corresponding indices $\boldsymbol{k}^\star$ within their placements $(\boldsymbol{k}^{\star(i)}, \boldsymbol{\lambda}^{\star(i)})$ (l.1-3 of Algorithm 2), the iteration index $t^{(i)}$ and polyhedron $\mathcal{P}^{(i)}$ memory placements are reset and can be used again for the second run of Vaidya's method (l.4-10 of Algorithm 2). During this second run, the vector $\boldsymbol{u}$ is stored in its corresponding memory placement $\boldsymbol{u}^{(i)}$ and updated along the algorithm. Once this run is finished, the output of $\mathsf{ApproxOracle}(i, j, \cdot)$ is readily available in the placement $\boldsymbol{u}^{(i)}$. For $i = p$, the algorithm does not need to wait for the output of a level-$(i+1)$ computation and can directly use the $j^{(p)}$-th component of the returned separation vector from the oracle $O_S$. As a result, the number of bits of memory used throughout the algorithm is at most

$$M = \sum_{i=1}^p \mathcal{O}\left(k_i^2 \ln \frac{1}{\xi}\right) = \mathcal{O}\left(\frac{d^2}{p} \ln \frac{d}{\epsilon}\right).$$

This ends the proof of the theorem. $\qquad \square$

We can already give the useful range for $p$ for our algorithms, which will also apply to the case with computational-memory constraints Appendix B.

*Proof of Corollary 3.1.* Suppose $\epsilon \geq \frac{1}{d^d}$. Then, for some $p_{max} = \Theta(\frac{C \ln \frac{1}{\epsilon}}{2 \ln d}) \leq d$, the algorithm from Theorem 3.2 yields a $\mathcal{O}(\frac{1}{\epsilon^2})$ oracle-complexity. On the other hand, if $\epsilon \leq \frac{1}{d^d}$, we can take $p_{max} = d$, which gives an oracle-complexity $\mathcal{O}((C \ln \frac{1}{\epsilon})^d)$. $\qquad \square$

## B  Memory-constrained feasibility problem with computations

In the last section we gave the main ideas that allow reducing the storage memory. However, Algorithm 4 does not account for memory constraints in computations as per Definition 2.2. For instance, computing the volumetric center $\mathsf{VolumetricCenter}(\mathcal{P})$ already requires infinite memory for infinite precision. More importantly, even if one discretizes the queries, the necessary precision and computational power may be prohibitive with the classical Vaidya's method Algorithm 1. Even finding a feasible point in the polyhedron (let alone the volumetric center) using only the constraints is itself computationally intensive. There has been significant work to make Vaidya's method computationally tractable [48, 1, 2]. These works address the issue of computational tractability, but the memory issue

is still present. Indeed, the precision depends among other parameters on the condition number of the matrix $\boldsymbol{H}$ in order to compute the leverage scores $\sigma_i$ for $i \in [m]$, which may not be well-conditioned. Second, to avoid memory overflow, we also need to ensure that the points queried have bounded norm, which is again not a priori guaranteed in the original version Algorithm 1.

To solve these issues and also give a computationally-efficient algorithm, the cutting-plane subroutine Algorithm 1 needs to be modified. In particular, the volumetric barrier needs to include regularization terms. Fortunately, these have already been studied in [27]. In a major breakthrough, this paper gave a cutting-plane algorithm with $\mathcal{O}(d^3 \ln^{\mathcal{O}(1)} \frac{d}{\epsilon})$ runtime complexity, improving over the seminal work from Vaidya and subsequent works which had $\mathcal{O}(d^{1+\omega} \ln^{\mathcal{O}(1)} \frac{d}{\epsilon})$ runtime complexity, where $\mathcal{O}(d^\omega)$ is the computational complexity of matrix multiplication. To achieve this result, they introduce various regularizing terms together with the logarithmic barrier. While the main motivation of [27] was computational complexity, as a side effect, these regularization terms also ensure that computations can be carried with efficient memory. We then use their method as a subroutine.

For the sake of exposition and conciseness, we describe a simplified version of their method, that is also deterministic. This comes at the expense of a suboptimal running time $\mathcal{O}(d^{1+\omega} \ln^{\mathcal{O}(1)} \frac{1}{\epsilon})$. We recall that our main concern is in memory usage rather than achieving the optimal runtime. The main technicality of this section is to show that their simplified method is numerically stable, and we emphasize that the original algorithm could also be shown to be numerically stable with similar techniques, leading to a time improvement from $\tilde{\mathcal{O}}(d^{1+\omega})$ to $\tilde{\mathcal{O}}(d^3)$. The memory usage, however, would not be improved.

## B.1 A memory-efficient Vaidya's method for computations, via [27]

Fix a polyhedron $\mathcal{P} = \{\boldsymbol{x} : \boldsymbol{A}\boldsymbol{x} \geq \boldsymbol{b}\}$. Using the same notations as for Vaidya's method in Section 4.1, we define the new leverage scores $\psi(\boldsymbol{x})_i = (\boldsymbol{A}_x(\boldsymbol{A}_x^\top \boldsymbol{A}_x + \lambda \boldsymbol{I})^{-1} \boldsymbol{A}_x^\top)_{i,i}$ and $\boldsymbol{\Psi}(\boldsymbol{x}) = diag(\psi(\boldsymbol{x}))$. Let $\mu(\boldsymbol{x}) = \min_i \psi(\boldsymbol{x})_i$. Last, let $\boldsymbol{Q}(\boldsymbol{x}) = \boldsymbol{A}_x^\top(c_e \boldsymbol{I} + \boldsymbol{\Psi}(x))\boldsymbol{A}_x + \lambda \boldsymbol{I}$, where $c_e > 0$ is a constant parameter to be defined. In [27], they consider minimizing the volumetric-analytic hybrid barrier function

$$p(\boldsymbol{x}) = -c_e \sum_{i=1}^{m} \ln s_i(\boldsymbol{x}) + \frac{1}{2} \ln \det(\boldsymbol{A}_x^\top \boldsymbol{A}_x + \lambda \boldsymbol{I}) + \frac{\lambda}{2} \|\boldsymbol{x}\|_2^2.$$

We can check [27] that

$$\nabla p(\boldsymbol{x}) = -\boldsymbol{A}_x^\top(c_e \cdot \boldsymbol{1} + \psi(x)) + \lambda \boldsymbol{x},$$

where $\boldsymbol{1}$ is the vector of ones. The following procedure gives a way to minimize this function efficiently given a good starting point.

**Input:** Initial point $\boldsymbol{x}^{(0)} \in \mathcal{P} = \{\boldsymbol{x} : \boldsymbol{A}\boldsymbol{x} \geq \boldsymbol{b}\}$
**Input:** Number of iterations $r > 0$
**Given :** $\|\nabla p(\boldsymbol{x}^{(0)})\|_{\boldsymbol{Q}(\boldsymbol{x}^{(0)})^{-1}} \leq \frac{1}{100}\sqrt{c_e + \mu(x^{(0)})} := \eta$.
1 **for** $k = 1$ *to* $r$ **do**
2     **if** $\|\nabla p(\boldsymbol{x}^{(k-1)})\|_{\boldsymbol{Q}(\boldsymbol{x}^{(0)})^{-1}} \leq 2(1 - \frac{1}{64})^r \eta$ **then Break**;
3     $\boldsymbol{x}^{(k)} = \boldsymbol{x}^{(k-1)} - \frac{1}{8}\boldsymbol{Q}(\boldsymbol{x}^{(0)})^{-1}\nabla p(\boldsymbol{x}^{(k-1)})$
4 **end**
**Output:** $\boldsymbol{x}^{(k)}$

$$\textbf{Algorithm 5: } \boldsymbol{x}^{(r)} = \mathsf{Centering}(\boldsymbol{x}^{(0)}, r)$$

We then present their simplified cutting-plane method.

In both Algorithm 5 and Algorithm 6, notice that the updates require to compute in particular the leverage scores $\psi(\boldsymbol{x})$, which can be computed in $\mathcal{O}(d^\omega)$ time using their formula. To achieve the $\mathcal{O}(d^3 \ln^{\mathcal{O}(1)} \frac{1}{\epsilon})$ computational complexity, an amortized computational cost $\mathcal{O}(d^2)$ is needed. The algorithm from [27] achieves this through various careful techniques aiming to update estimates of these leverage scores. The above cutting-plane algorithm is exactly that of [27] when these estimates are always exact (i.e. recomputed at each iteration), which yields the $d^{\omega-2}$ overhead time complexity. In particular, the original proof of convergence and correctness of [27] directly applies to this simplified algorithm.

**Input:** $\epsilon, \delta > 0$ and a separation oracle $O : \mathcal{C}_d \to \mathbb{R}^d$

**Check:** Throughout the algorithm, if $s_i(\boldsymbol{x}^{(t)}) < 2\epsilon$ for some $i$ then **return** $(\mathcal{P}_t, \boldsymbol{x}^{(t)})$

1   Initialize $\boldsymbol{x}^{(0)} = \boldsymbol{0}$ and $\mathcal{P}_0 := \{(-1, \boldsymbol{e}_i, -1), (-1, -\boldsymbol{e}_i, -1), \ i \in [d]\}$

2   **for** $t \geq 0$ **do**

3      **if** $\min_{i \in [m]} \psi(\boldsymbol{x}^{(t)})_i \leq c_d$ **then**

4         $\mathcal{P}_{t+1} = \mathcal{P}_t \setminus \{(k_j, \boldsymbol{a}_j, b_j)\}$ where $j \in \arg\min_{i \in [m]} \psi(\boldsymbol{x}^{(t)})_i$

5      **else**

6         **if** $\boldsymbol{x}^{(t)} \notin \mathcal{C}_d$ **then** $\boldsymbol{a} = -sign(x_i)\boldsymbol{e}_i$ where $i \in \arg\min_{j \in [d]} |x_j^{(t)}|$ ;

7         **else** $\boldsymbol{a} = O(\boldsymbol{x}^{(t)})$ ;

8         Let $b = \boldsymbol{a}^\top \boldsymbol{x}^{(t)} - c_a^{-1/2} \sqrt{\boldsymbol{a}^\top (\boldsymbol{A}^\top \boldsymbol{S}_{x^{(t)}}^{-2} \boldsymbol{A} + \lambda \boldsymbol{I})^{-1} \boldsymbol{a}}$

9         $\mathcal{P}_{t+1} = \mathcal{P}_t \cup \{(t, \boldsymbol{a}, b)\}$

10     $\boldsymbol{x}^{(t+1)} = \mathsf{Centering}(\boldsymbol{x}^{(t)}, 200, c_\Delta)$

11 **end**

**Algorithm 6:** An efficient cutting-plane method, simplified from [27]

It remains to check whether one can implement this algorithm with efficient memory, corresponding to checking this method's numerical stability.

**Lemma B.1.** *Suppose that each iterate of the centering Algorithm 5, $\|\nabla p(\boldsymbol{x}^{(k-1)})\|_{\boldsymbol{Q}(\boldsymbol{x}^{(0)})^{-1}}$ is computed up to precision $(1 - \frac{1}{64})^r \eta$ (l.2), and $\boldsymbol{x}^{(k)}$ is computed up to an error $\boldsymbol{\zeta}^{(k)}$ with $\|\boldsymbol{\zeta}^{(k)}\|_{\boldsymbol{Q}(\boldsymbol{x}^{(0)})} \leq \frac{1}{2^{10r}}(1 - \frac{1}{64})^r \eta$ (l.3). Then, Algorithm 5 outputs $\boldsymbol{x}^{(k)}$ such that $\|\nabla p(\boldsymbol{x}^{(k)})\|_{\boldsymbol{Q}^{-1}(\boldsymbol{x}^{(k)})} \leq 3(1 - \frac{1}{64})^r \eta$ and all iterates computed during the procedure satisfy $\|\boldsymbol{S}_{x^{(0)}}^{-1}(\boldsymbol{s}(\boldsymbol{x}^{(t)}) - \boldsymbol{s}(\boldsymbol{x}^{(0)}))\|_2 \leq \frac{1}{10}$.*

*Proof.* As mentioned above, without computation errors, the result from [27] would apply directly. Here, we simply adapt the proof to the case with computational errors to show that it still applies. Denote $\boldsymbol{Q} = \boldsymbol{Q}(\boldsymbol{x}^{(0)})$ for convenience. Let $\eta = \frac{1}{100}\sqrt{c_e + \mu(\boldsymbol{x}^{(0)})}$. We prove by induction that $\|\boldsymbol{x}^{(t)} - \boldsymbol{x}^{(0)}\|_{\boldsymbol{Q}} \leq 9\eta$, $\|\nabla p(\boldsymbol{x}^{(t)})\|_{\boldsymbol{Q}^{-1}} \leq (1 - \frac{1}{64})^t \eta$ for all $t \leq r$. For a given iteration $t$, denote $\tilde{\boldsymbol{x}}^{(t+1)} = \boldsymbol{x}^{(k-1)} - \frac{1}{8}\boldsymbol{Q}^{-1}\nabla p(\boldsymbol{x}^{(k-1)})$ the result of the exact computation. The same arguments as in the original proof give $\|\tilde{\boldsymbol{x}}^{(t+1)} - \boldsymbol{x}^{(0)}\|_{\boldsymbol{Q}} \leq 9\eta$, and

$$\|\nabla p(\tilde{\boldsymbol{x}}^{(t+1)})\|_{\boldsymbol{Q}^{-1}} \leq \left(1 - \frac{1}{32}\right)\|\nabla p(\boldsymbol{x}^{(t)})\|_{\boldsymbol{Q}^{-1}}.$$

Now because $\|\tilde{\boldsymbol{x}}^{(t+1)} - \boldsymbol{x}^{(t+1)}\|_{\boldsymbol{Q}} \leq \eta$, we have $\|\tilde{\boldsymbol{x}}^{(t+1)} - \boldsymbol{x}^{(0)}\|_{\boldsymbol{Q}}, \|\boldsymbol{x}^{(t+1)} - \boldsymbol{x}^{(0)}\|_{\boldsymbol{Q}} \leq 10\eta$, so that [27, Lemma 11] gives $\nabla^2 p(\boldsymbol{y}(u)) \preceq 8\boldsymbol{Q}(\boldsymbol{y}(u)) \preceq 16\boldsymbol{Q}$, where $\boldsymbol{y}(u) = \boldsymbol{x}^{(t+1)} + u(\tilde{\boldsymbol{x}}^{(t+1)} - \boldsymbol{x}^{(t+1)})$ for $u \in [0, 1]$. Thus,

$$\|\nabla p(\tilde{\boldsymbol{x}}^{(t+1)}) - \nabla p(\boldsymbol{x}^{(t+1)})\|_{\boldsymbol{Q}^{-1}} \leq \left\|\int_0^1 \nabla^2 p(\boldsymbol{y}(u))(\tilde{\boldsymbol{x}}^{(t+1)} - \boldsymbol{x}^{(t+1)})\right\|_{\boldsymbol{Q}^{-1}}$$

$$\leq 16\|\tilde{\boldsymbol{x}}^{(t+1)} - \boldsymbol{x}^{(t+1)}\|_{\boldsymbol{Q}}.$$

Now by construction of the procedure, if the algorithm performed iteration $t + 1$, we have $\|\nabla p(\boldsymbol{x}^{(t)})\|_{\boldsymbol{Q}^{-1}} \geq (1 - \frac{1}{64})^r \eta$. Combining this with the fact that $\|\tilde{\boldsymbol{x}}^{(t+1)} - \boldsymbol{x}^{(t+1)}\|_{\boldsymbol{Q}} \leq \frac{1}{2^{10r}}(1 - \frac{1}{64})^r \eta$, obtain

$$\|\nabla p(\boldsymbol{x}^{(t+1)})\|_{\boldsymbol{Q}^{-1}} \leq \|\nabla p(\tilde{\boldsymbol{x}}^{(t+1)}) - \nabla p(\boldsymbol{x}^{(t+1)})\|_{\boldsymbol{Q}^{-1}} + \|\nabla p(\tilde{\boldsymbol{x}}^{(t+1)})\|_{\boldsymbol{Q}^{-1}}$$

$$\leq \left(1 - \frac{1}{64}\right)\|\nabla p(\boldsymbol{x}^{(t)})\|_{\boldsymbol{Q}^{-1}}.$$

We now write

$$\|\boldsymbol{x}^{(t+1)} - \boldsymbol{x}^{(0)}\|_{\boldsymbol{Q}} \leq \sum_{k=0}^t \|\tilde{\boldsymbol{x}}^{(k+1)} - \boldsymbol{x}^{(k+1)}\|_{\boldsymbol{Q}} + \frac{1}{8}\|\boldsymbol{Q}^{-1}\nabla p(\boldsymbol{x}^{(k)})\|_{\boldsymbol{Q}}$$

$$\leq \eta + \frac{1}{8}\sum_{i=0}^\infty \left(1 - \frac{1}{64}\right)^i \eta \leq 9\eta.$$

The induction is now complete. When the algorithm stops, either the $r$ steps were performed, in which case the induction already shows that $\|\nabla p(\boldsymbol{x}^{(r)})\|_{\boldsymbol{Q}^{-1}} \leq (1 - \frac{1}{64})^r \eta$. Otherwise, if the algorithm terminates at iteration $k$, because $\|\nabla p(\boldsymbol{x}^{(k)})\|_{\boldsymbol{Q}^{-1}}$ was computed to precision $(1 - \frac{1}{64})^r \eta$, we have (see l.2 of Algorithm 5)

$$\|\nabla p(\boldsymbol{x}^{(k)})\|_{\boldsymbol{Q}^{-1}} \leq 2\left(1 - \frac{1}{64}\right)^r \eta + \left(1 - \frac{1}{64}\right)^r \eta = 3\left(1 - \frac{1}{64}\right)^r \eta.$$

The same argument as in the original proof shows that at each iteration $t$,

$$\|\boldsymbol{S}_{x^{(0)}}^{-1}(\boldsymbol{s}(\boldsymbol{x}^{(t)}) - \boldsymbol{s}(\boldsymbol{x}^{(0)}))\|_2 = \|\boldsymbol{x}^{(t)} - \boldsymbol{x}^{(0)}\|_{\boldsymbol{A}^\top \boldsymbol{S}_{x^{(0)}}^{-2} \boldsymbol{A}} \leq \frac{\|\boldsymbol{x}^{(t)} - \boldsymbol{x}^{(0)}\|_{\boldsymbol{Q}}}{\sqrt{\mu(\boldsymbol{x}^{(0)}) + c_e}} \leq \frac{1}{10}.$$

This ends the proof of the lemma. $\qquad\square$

Because of rounding errors, Lemma B.1 has an extra factor 3 compared to the original guarantee in [27, Lemma 14]. To achieve the same guarantee, it suffices to perform $70 \geq \ln(3)/\ln(1/(1 - \frac{1}{64}))$ additional centering procedures at most. hence, instead of performing 200 centering procedures during the cutting plane method, we perform 270 (l.10 of Algorithm 6). We next turn to the numerical stability of the main Algorithm 6.

**Lemma B.2.** *Suppose that throughout the algorithm, when checking the stopping criterion $\min_{i \in [m]} s_i(\boldsymbol{x}) < 2\epsilon$, the quantities $s_i(\boldsymbol{x})$ were computed with accuracy $\epsilon$. Suppose that at each iteration of Algorithm 6, the leverage scores $\psi(\boldsymbol{x}^{(t)})$ are computed up to multiplicative precision $c_\Delta/4$ (l.3), that when a constraint is added, the response of the oracle $\boldsymbol{a}$ (l.7) is stored perfectly but $b$ (l.8) is computed up to precision $\Omega(\frac{\epsilon}{\sqrt{n}})$. Further suppose that the centering Algorithm 5 is run with numerical approximations according to the assumptions in Lemma B.1. Then, all guarantees for the original algorithm in [27] hold, up to a factor 3 for $\epsilon$.*

*Proof.* We start with the termination criterion. Given the requirement on the computational accuracy, we know that the final output $\boldsymbol{x}$ satisfies $\min_{i \in [m]} s_i(\boldsymbol{x}) \leq 3\epsilon$. Further, during the algorithm, if it does not stop, then one has $\min_{i \in [m]} s_i(\boldsymbol{x}) \geq \epsilon$, which is precisely the guarantee of the original algorithm in [27].

We next turn to the computation of the leverage scores in l.4. In the original algorithm, only a $c_\Delta$-estimate is computed. Precisely, one computes a vector $\boldsymbol{w}^{(t)}$ such that for all $i \in [d]$, $\psi(\boldsymbol{x}^{(t)})_i \leq w_i \leq (1 + c_\Delta)\psi(\boldsymbol{x}^{(t)})_i$, then deletes a constraint when $\min_{i \in [m^{(t)}]} w_i^{(t)} \leq c_d$. In the adapted algorithm, let $\tilde{\psi}(\boldsymbol{x}^{(t)})_i$ denote the computed leverage scores for $i \in [d]$. By assumption, we have

$$(1 - c_\Delta/4)\psi(\boldsymbol{x}^{(t)})_i \leq \tilde{\psi}(\boldsymbol{x}^{(t)})_i \leq (1 + c_\Delta/4)\psi(\boldsymbol{x}^{(t)})_i.$$

Up to re-defining the constant $c_d$ as $(1 - c_\Delta/4)c_d$, $\tilde{\psi}(\boldsymbol{x}^{(t)})$ is precisely within the guarantee bounds of the algorithm. For the accuracy on the separation oracle response and the second-term value $b$, [27] emphasizes that the algorithm always changes constraints by a $\delta$ amount where $\delta = \Omega(\frac{\epsilon}{\sqrt{d}})$ so that an inexact separation oracle with accuracy $\Omega(\frac{\epsilon}{\sqrt{d}})$ suffices. Therefore, storing an $\Omega(\frac{\epsilon}{\sqrt{d}})$ accuracy of the second term keeps the guarantees of the algorithm. Last, we checked in Lemma B.1 that the centering procedure Algorithm 5 satisfies all the requirements needed in the original proof [27]. $\quad\square$

For our recursive method, we need an efficient cutting-plane method that also provides a proof (certificate) of convergence. This is also provided by [27] that provide a proof that the feasible region has small width in one of the directions $\boldsymbol{a}_i$ of the returned polyhedron.

**Lemma B.3.** *[27, Lemma 28] Let $(\mathcal{P}, \boldsymbol{x}, (\lambda_i)_i)$ be the output of Algorithm 7. Then, $\boldsymbol{x}$ is feasible, $\|\boldsymbol{x}\|_2 \leq 3\sqrt{d}$, $\lambda_j \geq 0$ for all $j$ and $\sum_i \lambda_i = 1$. Further,*

$$\left\|\sum_i \lambda_i \boldsymbol{a}_i\right\|_2 = \mathcal{O}\left(\epsilon\sqrt{d}\ln\frac{d}{\epsilon}\right), \quad and \quad \sum_i \lambda_i(\boldsymbol{a}_i^\top \boldsymbol{x} - b_j) \leq \mathcal{O}\left(d\epsilon\ln\frac{d}{\epsilon}\right).$$

We are now ready to show that Algorithm 6 can be implemented with efficient memory and also provides a proof of the convergence of the algorithm.

**Input:** $\epsilon > 0$ and a separation oracle $O : \mathcal{C}_d \to \mathbb{R}^d$

1 Run Algorithm 6 to obtain a polyhedron $\mathcal{P}$ and a feasible point $\boldsymbol{x}$

2 $\boldsymbol{x}^\star = \text{Centering}(\boldsymbol{x}, 64 \ln \frac{2}{\epsilon}, c_\Delta)$

3 $\lambda_i = \frac{c_e + \psi_i(\boldsymbol{x}^\star)}{s_i(\boldsymbol{x}^\star)} \left( \sum_j \frac{c_e + \psi_j(\boldsymbol{x}^\star)}{s_j(\boldsymbol{x}^\star)} \right)^{-1}$ for all $i$

**Output:** $(\mathcal{P}, \boldsymbol{x}^\star, (\lambda_i)_i)$

**Algorithm 7:** Cutting-plane algorithm with certified optimality

**Proposition B.1.** *Provided that the output of the oracle are vectors discretized to precision $\text{poly}(\frac{\epsilon}{d})$ and have norm at most $1$, Algorithm 7 can be implemented with $\mathcal{O}(d^2 \ln \frac{d}{\epsilon})$ bits of memory to output a certified optimal point according to Lemma B.3. The algorithm performs $\mathcal{O}(d \ln \frac{d}{\epsilon})$ calls to the separation oracle and runs in $\mathcal{O}(d^{1+\omega} \ln^{\mathcal{O}(1)} \frac{d}{\epsilon})$ time.*

*Proof.* We already checked the numerical stability of Algorithm 6 in Lemma B.2. It remains to check the next steps of the algorithm. The centering procedure is stable again via Lemma B.1. It also suffices to compute the coefficients $\lambda_j$ up to accuracy $\mathcal{O}(\epsilon/(\sqrt{d}) \ln(d/\epsilon))$ to keep the guarantees desired since by construction all vectors $\boldsymbol{a}_i$ have norm at most one.

It now remains to show that the algorithm can be implemented with efficient memory. We recall that at any point during the algorithm, the polyhedron $\mathcal{P}$ has at most $\mathcal{O}(d)$ constraints [27, Lemma 22]. Hence, since we assumed that each vector $\boldsymbol{a}_i$ composing a constraint is discretized to precision $\text{poly}(\frac{\epsilon}{d})$, we can store the polyhedron constraints with $\mathcal{O}(d^2 \ln \frac{d}{\epsilon})$ bits of memory. The second terms $b$ are computed up to precision $\Omega(\epsilon/\sqrt{d})$ hence only use $\mathcal{O}(d \ln \frac{d}{\epsilon})$ bits of memory. The algorithm also keeps the current iterate $x^{(t)}$ in memory. These are all bounded throughout the memory $\|x^{(t)}\|_2 = \mathcal{O}(\sqrt{d})$ [27, Lemma 23], hence only require $\mathcal{O}(d \ln \frac{d}{\epsilon})$ bits of memory for the desired accuracy.

Next, the distances to the constraints are bounded at any step of the algorithm: $s_i(\boldsymbol{x}^{(t)}) \leq \mathcal{O}(\sqrt{d})$ [27, Lemma 24], hence computing $s_i(\boldsymbol{x}^{(t)})$ to the required accuracy is memory-efficient. Recall that from the termination criterion, except for the last point, any point $\boldsymbol{x}$ during the algorithm satisfies $s_i(\boldsymbol{x}) \geq \epsilon$ for all constraints $i \in [m]$. In particular, this bounds the eigenvalues of $\boldsymbol{Q}$ since $\lambda I \preceq \boldsymbol{Q}(x) \preceq (\lambda + m(c_e + 1)/\epsilon^2)\boldsymbol{I}$. Thus, the matrix is sufficiently well-conditioned to achieve the accuracy guarantees from Lemma B.1 using $\mathcal{O}(d^2 \ln \frac{d}{\epsilon})$ memory during matrix inversions (and matrix multiplications). Similarly, for the computation of leverage scores, we use $\boldsymbol{\Psi}(x) = diag(\boldsymbol{A}_x(\boldsymbol{A}_x^\top \boldsymbol{A}_x + \lambda \boldsymbol{I})^{-1} \boldsymbol{A}_x^\top)$, where $\lambda \boldsymbol{I} \preceq \boldsymbol{A}_x^\top \boldsymbol{A}_x + \lambda \boldsymbol{I} \preceq (\lambda + m\epsilon^{-2})\boldsymbol{I}$. This same matrix inversion appears when computing the second term of an added constraint. Overall, all linear algebra operations are well conditioned and implementable with required accuracy with $\mathcal{O}(d^2 \ln \frac{d}{\epsilon})$ memory. Using fast matrix multiplication, all these operations can be performed in $\tilde{\mathcal{O}}(d^\omega)$ time per iteration of the cutting-plane algorithm since these methods are also known to be numerically stable [15]. Thus, the total time complexity is $\mathcal{O}(d^{1+\omega} \ln^{O(1)} \frac{d}{\epsilon})$. The oracle-complexity still has optimal $\mathcal{O}(d \ln \frac{d}{\epsilon})$ oracle-complexity as in the original algorithm. $\square$

Up to changing $\epsilon$ to $c \cdot \epsilon/(d \ln \frac{d}{\epsilon})$, the described algorithm finds constraints given by $\boldsymbol{a}_i$ and $b_i$, $i \in [m]$ returned by the normalized separation oracle, coefficients $\lambda_i$, $i \in [m]$, and a feasible point $\boldsymbol{x}^\star$ such that for any vector in the unit cube, $\boldsymbol{z} \in \mathcal{C}_d$, one has

$$\min_{i \in [m]} \boldsymbol{a}_i^\top \boldsymbol{z} - b_i \leq \sum_{i \in [m]} \lambda_i(\boldsymbol{a}_i^\top \boldsymbol{z} - b_i) \leq \left( \sum_{i \in [m]} \lambda \boldsymbol{a}_i \right)^\top (\boldsymbol{x}^\star - \boldsymbol{z}) + \sum_{i \in [m]} \lambda_i(\boldsymbol{a}_i^\top \boldsymbol{x}^\star - b_i) \leq \epsilon.$$

This effectively replaces Lemma 4.1.

### B.2 Merging Algorithm 7 within the recursive algorithm

Algorithms 2 to 4 from the recursive procedure need to be slightly adapted to the new format of the cutting-plane method's output. In particular, the oracles do not take as input polyhedrons (and

eventually query their volumetric center as before), but directly take as input an point (which is an approximate volumetric center).

---

**Input:** $\delta, \xi, O_x : \mathcal{C}_n \to \mathbb{R}^m$ and $O_y : \mathcal{C}_n \to \mathbb{R}^n$

1   Run Algorithm 7 with parameter $c \cdot \delta/(d \ln \frac{d}{\delta})$, $\xi$ and $O_y$ to obtain $(\mathcal{P}^\star, \boldsymbol{x}^\star, \boldsymbol{\lambda})$

2   Store $\boldsymbol{k}^\star = (k_i, i \in [m])$ where $m = |\mathcal{P}^\star|$, and $\boldsymbol{\lambda}^\star \leftarrow \mathsf{Discretize}(\boldsymbol{\lambda}^\star, \xi)$

3   Initialize $\mathcal{P}_0 := \{(-1, \boldsymbol{e}_i, -1), (-1 - \boldsymbol{e}_i, -1), \ i \in [d]\}$, $\boldsymbol{x}^{(0)} = \boldsymbol{0}$ and let $\boldsymbol{u} = \boldsymbol{0} \in \mathbb{R}^m$

4   **for** $t = 0, 1, \ldots, \max_i k_i$ **do**

5      **if** $t = k_i^\star$ *for some* $i \in [m]$ **then**

6          $\boldsymbol{g}_x = O_x(\boldsymbol{x}^{(t)})$

7          $\boldsymbol{u} \leftarrow \mathsf{Discretize}_m(\boldsymbol{u} + \lambda_i^\star \boldsymbol{g}_x, \xi)$

8      Update $\mathcal{P}_t$ to get $\mathcal{P}_{t+1}$, and $\boldsymbol{x}^{(t)}$ to get $\boldsymbol{x}^{(t+1)}$ as in Algorithm 6

9   **end**

10   **return** $\boldsymbol{u}$

**Algorithm 8:** $\mathsf{ApproxSeparationVector}_{\delta,\xi}(O_x, O_y)$

---

**Input:** $\delta, \xi, 1 \le j \le i \le p, \boldsymbol{x}^{(r)} \in \mathcal{C}_{k_r}$ for $r \in [i], O_S : \mathcal{C}_d \to \mathbb{R}^d$

1   **if** $i = p$ **then**

2      $(\boldsymbol{g}_1, \ldots, \boldsymbol{g}_p) = O_S(\boldsymbol{x}_1, \ldots, \boldsymbol{x}_p)$

3      **return** $\mathsf{Discretize}_{k_j}(\boldsymbol{g}_j, \xi)$

4   **end**

5   Define $O_x : \mathcal{C}_{k_{i+1}} \to \mathbb{R}^{k_j}$ as $\mathsf{ApproxOracle}_{\delta,\xi,\mathcal{O}_f}(i+1, j, \boldsymbol{x}^{(1)}, , \ldots, \boldsymbol{x}^{(i)}, \cdot)$

6   Define $O_y : \mathcal{C}_{k_{i+1}} \to \mathbb{R}^{k_{i+1}}$ as $\mathsf{ApproxOracle}_{\delta,\xi,\mathcal{O}_f}(i+1, i+1, \boldsymbol{x}^{(1)}, \ldots, \boldsymbol{x}^{(i)}, \cdot)$

7   **return** $\mathsf{ApproxSeparationVector}_{\delta,\xi}(O_x, O_y)$

**Algorithm 9:** $\mathsf{ApproxOracle}_{\delta,\xi,O_S}(i, j, \boldsymbol{x}^{(1)}, \ldots, \boldsymbol{x}^{(i)})$

---

**Input:** $\delta, \xi$, and $\mathcal{O}_S : \mathcal{C}_d \to \mathbb{R}^d$ a separation oracle

**Check :** Throughout the algorithm, if $O_S$ returned $\mathsf{Success}$ to a query $\boldsymbol{x}$, **return** $\boldsymbol{x}$

1   Run Algorithm 6 with parameters $\delta$ and $\xi$ and oracle $\mathsf{ApproxOracle}_{\delta,\xi,O_S}(1, 1, \cdot)$

**Algorithm 10:** Memory-constrained algorithm for convex optimization

---

The same proof as for Algorithm 4 shows that Algorithm 10 run with the parameters in Theorem A.1 also outputs a successful vector using the same oracle-complexity. We only need to analyze the memory usage in more detail.

*Proof of Theorem 3.2.* As mentioned above, we will check that Algorithm 10 with the same parameters $\delta = \frac{\epsilon}{4d}$ and $\xi = \frac{\sigma_{min}\epsilon}{32d^{5/2}}$ as in Theorem A.1 satisfies the desired requirements. We have already checked its correctness and oracle-complexity. Using the same arguments, the computational complexity is of the form $\mathcal{O}(\mathcal{O}(\mathsf{ComplexityCuttingPlanes})^p)$ where $\mathsf{ComplexityCuttingPlanes}$ is the computational complexity of the cutting-plane method used, i.e., here of Algorithm 7. Hence, the computational complexity is $\mathcal{O}((C(d/p)^{1+\omega} \ln^{\mathcal{O}(1)} \frac{d}{\epsilon})^p)$ for some universal constant $C \ge 2$. We now turn to the memory. In addition to the memory of Algorithm 4, described in Table 1, we need

1. a placement for all $i \in [p]$ for the current iterate $\boldsymbol{x}^{(i)}$: $\mathcal{O}(k_i \ln \frac{1}{\xi})$ bits,

2. a placement for computations, that is shared for all layers (used to compute leverage scores, centering procedures, etc. By Proposition B.1, since the vectors are always discretized to precision $\xi$, this requires $\mathcal{O}(\max_{i \in [p]} k_i^2 \ln \frac{d}{\epsilon})$ bits,

3. the placement $Q$ to perform queries is the concatenation of the placements $(\boldsymbol{x}^{(1)}, \ldots, \boldsymbol{x}^{(p)})$: no additional bits needed.

4. a placement $N$ to store the precision needed for the oracle responses: $\mathcal{O}(\ln \frac{1}{\xi})$ bits

5. a placement $R$ to receive the oracle responses: $\mathcal{O}(d \ln \frac{1}{\xi})$ bits.

The new memory structure is summarized in Table 2.

With the same arguments as in the original proof of Theorem A.1, this memory is sufficient to run the algorithm and perform computations, thanks to the computation placement. The total number of bits used throughout the algorithm remains the same, $\mathcal{O}(\frac{d^2}{p} \ln \frac{d}{\epsilon})$. This ends the proof of the theorem. $\qquad\square$

Table 2: Memory structure for Algorithm 10

| $i$ | 1 | $\ldots$ | $p$ | Oracle response | Precision |
|---|---|---|---|---|---|
| $j$ | $j^{(1)}$ | | $j^{(p)}$ | $R = (R_1, \ldots, R_p)$ | $N$ |
| Iteration index | $t^{(1)}$ | | $t^{(p)}$ | | |
| Polyhedron | $\mathcal{P}^{(1)} = \begin{pmatrix} k_1, \boldsymbol{a}_1, b_1 \\ k_2, \boldsymbol{a}_2, b_2 \\ \ldots \\ k_m, \boldsymbol{a}_m, b_m \end{pmatrix}$ | | $\mathcal{P}^{(p)}$ | | |
| Current iterate | $\boldsymbol{x}^{(1)}$ | | $\boldsymbol{x}^{(p)}$ | Computation memory | |
| Computed dual variables | $(\boldsymbol{k}^{\star}, \boldsymbol{\lambda}^{\star}) = \begin{pmatrix} k_1^{\star}, \lambda_1^{\star} \\ k_2^{\star}, \lambda_2^{\star} \\ \ldots \end{pmatrix}$ | | $(\boldsymbol{k}^{\star(p)}, \boldsymbol{\lambda}^{\star(p)})$ | | |
| Working separation vector | $\boldsymbol{u}^{(1)}$ | | $\boldsymbol{u}^{(p)}$ | | |

## C Improved oracle-complexity/memory lower-bound trade-offs

We recall the three oracle-complexity/memory lower-bound trade-offs known in the literature.

1. First, [31] showed that any (including randomized) algorithm for convex optimization uses $d^{1.25-\delta}$ memory or makes $\tilde{\Omega}(d^{1+4\delta/3})$ queries.

2. Then, [5] showed that any deterministic algorithm for convex optimization uses $d^{2-\delta}$ memory or makes $\tilde{\Omega}(d^{1+\delta/3})$ queries.

3. Last, [5] show that any deterministic algorithm for the feasibility problem uses $d^{2-\delta}$ memory or makes $\tilde{\Omega}(d^{1+\delta})$ queries.

Although these papers mainly focused on the regime $\epsilon = 1/\mathrm{poly}(d)$ and as a result $\ln \frac{1}{\epsilon} = \mathcal{O}(\ln d)$, neither of these lower bounds have an explicit dependence in $\epsilon$. This can lead to sub-optimal lower bounds whenever $\ln \frac{1}{\epsilon} \gg \ln d$. Furthermore, in the exponential regime $\epsilon \leq \frac{1}{2^{\mathcal{O}(d)}}$, these results do not effectively give useful lower bounds. Indeed, in this regime, one has $d^2 = \mathcal{O}(d \ln \frac{1}{\epsilon})$ and as a result, the lower bounds provided are weaker than the classical $\Omega(d \ln \frac{1}{\epsilon})$ lower bounds for oracle-complexity [34] and memory [52]. In particular, in this exponential regime, these results fail to show that there is any trade-off between oracle-complexity and memory.

In this section, we aim to explicit the dependence in $\epsilon$ of these lower-bounds. We show with simple modifications and additional arguments that one can roughly multiply these oracle-complexity and memory lower bounds by a factor $\ln \frac{1}{\epsilon}$ each. We split the proofs in two. First we give arguments to improve the memory dependence by a factor $\ln \frac{1}{\epsilon}$, which is achieved by modifying the sampling of the rows of the matrix $\boldsymbol{A}$ defining a wall term common to the functions considered in the lower bound proofs [31, 5]. Then we show how to improve the oracle-complexity dependence by an additional $\ln \frac{1}{\epsilon} / \ln d$ factor, via a standard rescaling argument.

### C.1 Improving the memory lower bound

We start with some concentration results on random vectors. [31] gave the following result for random vectors in the hypercube.

**Lemma C.1** ([31]). *Let $\boldsymbol{h} \sim \mathcal{U}(\{\pm 1\}^d)$. Then, for any $t \in (0, 1/2]$ and any matrix $\boldsymbol{Z} = [\boldsymbol{z}_1, \ldots, \boldsymbol{z}_k] \in \mathbb{R}^{d \times k}$ with orthonormal columns,*

$$\mathbb{P}(\|\boldsymbol{Z}^\top \boldsymbol{h}\|_\infty \leq t) \leq 2^{-c_H k}.$$

Instead, we will need a similar concentration result for random unit vectors in the unit sphere.

**Lemma C.2.** *Let $k \leq d$ and $\boldsymbol{x}_1, \ldots, \boldsymbol{x}_k$ be $k$ orthonormal vectors, and $\zeta \leq 1$.*

$$\mathbb{P}_{\boldsymbol{y} \sim \mathcal{U}(S^{d-1})} \left( |\boldsymbol{x}_i^\top \boldsymbol{y}| \leq \frac{\zeta}{\sqrt{d}}, \ i \in [k] \right) \leq \left( \frac{2}{\sqrt{\pi}} \zeta \right)^k \leq (\sqrt{2}\zeta)^k.$$

*Proof.* First, by isometry, we can suppose that the orthonormal vectors are simply $\boldsymbol{e}_1, \ldots, \boldsymbol{e}_k$. We now prove the result by induction on $d$. For $d = 1$, the result holds directly. Fix $d \geq 2$, and $1 \leq k < d$. Then, if $S_n$ is the surface area of $S^n$ the $n$-dimensional sphere, then

$$\mathbb{P}\left( |y_1| \leq \frac{\zeta}{\sqrt{d}} \right) \leq \frac{S_{d-2}}{S_{d-1}} \frac{2\zeta}{\sqrt{d}} = \frac{2\zeta}{\sqrt{\pi d}} \frac{\Gamma(d/2)}{\Gamma(d/2 - 1/2)} \leq \frac{2}{\sqrt{\pi}} \zeta. \tag{7}$$

Conditionally on the value of $y_1$, the vector $(y_2, \ldots, y_d)$ follows a uniform distribution on the $(d-2)$-sphere of radius $\sqrt{1 - y_1^2}$. Then,

$$\mathbb{P}\left( |y_i| \leq \frac{\zeta}{\sqrt{d}}, \ 2 \leq i \leq k \mid y_1 \right) = \mathbb{P}_{\boldsymbol{z} \sim \mathcal{U}(S^{d-2})} \left( |z_i| \leq \frac{\zeta}{\sqrt{d(1 - y_1^2)}}, \ 2 \leq i \leq k \right)$$

Now recall that since $|x_1| \leq 1/\sqrt{d}$, we have $d(1 - x_1^2) \geq d - 1$. Therefore, using the induction,

$$\mathbb{P}\left( |y_i| \leq \frac{\zeta}{\sqrt{d}}, \ 2 \leq i \leq k \mid y_1 \right) \leq \mathbb{P}_{\boldsymbol{z} \sim \mathcal{U}(S^{d-2})} \left( |z_i| \leq \frac{\zeta}{\sqrt{d-1}}, \ 2 \leq i \leq k \right) \leq \left( \frac{2\zeta}{\sqrt{\pi}} \right)^{k-1}.$$

Combining this equation with Eq (7) ends the proof. $\qquad\square$

We next use the following lemma to partition the unit sphere $S^{d-1}$.

**Lemma C.3** ([19] Lemma 21). *For any $0 < \delta < \pi/2$, the sphere $S^{d-1}$ can be partitioned into $N(\delta) = (\mathcal{O}(1)/\delta)^d$ equal volume cells, each of diameter at most $\delta$.*

Following the notation from [5], we denote by $\mathcal{V}_\delta = \{V_i(\delta), i \in [N(\delta)]\}$ the corresponding partition, and consider a set of representatives $\mathcal{D}_\delta = \{\boldsymbol{b}_i(\delta), i \in [N(\delta)]\} \subset S^{d-1}$ such that for all $i \in [N(\delta)]$, $\boldsymbol{b}_i(\delta) \in V_i(\delta)$. With these notations we can define the discretization function $\phi_\delta$ as follows

$$\phi_\delta(\boldsymbol{x}) = \boldsymbol{b}_i(\delta), \quad \boldsymbol{x} \in V_i(\delta).$$

We then denote by $\mathcal{U}_\delta$ the distribution of $\phi_\delta(\boldsymbol{z})$ where $\boldsymbol{z} \sim \mathcal{U}(S^{d-1})$ is sampled uniformly on the sphere. Note that because the cells of $\mathcal{V}_\delta$ have equal volume, $\mathcal{U}_\delta$ is simply the uniform distribution on the discretization $\mathcal{D}_\delta$.

We are now ready to give the modifications necessary to the proofs, to include a factor $\ln \frac{1}{\epsilon}$ for the necessary memory. For their lower bounds, [31] exhibit a distribution of convex functions that are hard to optimize. Building upon their work [5] construct classes of convex functions that are hard to optimize, but that also depend adaptively on the considered optimization algorithm. For both, the functions considered a barrier term of the form $\|\boldsymbol{A}\boldsymbol{x}\|_\infty$, where $\boldsymbol{A}$ is a matrix of $\approx d/2$ rows that are independently drawn as uniform on the hypercube $\mathcal{U}(\{\pm 1\}^d)$. The argument shows that memorizing $\boldsymbol{A}$ is necessary to a certain extent. As a result, the lower bounds can only apply for a memory of at most $\mathcal{O}(d^2)$ bits, which is sufficient to memorize such a binary matrix. Instead, we draw rows independently according to the distribution $\mathcal{U}_\delta$, where $\delta \approx \epsilon$. We explicit the corresponding adaptations for each known trade-off. We start with the lower bounds from [5] for ease of exposition; although these build upon those of [31], their parametrization makes the adaptation more straightforward.

### C.1.1 Lower bound of [5] for convex optimization and deterministic algorithms

For this lower bound, we use the exact same form of functions as they introduced,

$$\max \left\{ \|\boldsymbol{A}\boldsymbol{x}\|_\infty - \eta, \eta \boldsymbol{v}_0^\top \boldsymbol{x}, \eta \left( \max_{p \le p_{max}, l \le l_p} \boldsymbol{v}_{p,l}^\top \boldsymbol{x} - p\gamma_1 - l\gamma_2 \right) \right\},$$

with the difference that rows of $\boldsymbol{A}$ are take i.i.d. distributed according to $\mathcal{U}_{\delta'}$ instead of $\mathcal{U}(\{\pm 1\}^d)$. As a remark, they use $n = \lceil d/4 \rceil$ rows for $\boldsymbol{A}$. Except for $\eta$, we keep all parameters $\gamma_1, \gamma_2$, etc as in the original proof, and we will take $\delta' = \epsilon$ and $\eta = 2\sqrt{d}\epsilon$. The reason why we introduced $\delta'$ instead of $\delta$ is that the original construction also needs the discretization $\phi_\delta$. This is used during the optimization procedure which constructs adaptively this class of functions, and only needs $\delta = \text{poly}(1/d)$ instead of $\delta$ of order $\epsilon$.

**Theorem C.1.** *For $\epsilon \le 1/(2d^{4.5})$ and any $\delta \in [0,1]$, a deterministic first-order algorithm guaranteed to minimize 1-Lipschitz convex functions over the unit ball with $\epsilon$ accuracy uses at least $d^{2-\delta} \ln \frac{1}{\epsilon}$ bits of memory or makes $\tilde{\Omega}(d^{1+\delta/3})$ queries.*

With the changes defined above, we can easily check that all results from [5] which reduce convex optimization to the optimization procedure, then the optimization procedure to their Orthogonal Vector Game with Hints (OVGH) [5, Game 2], are not affected by our changes. The only modifications to perform are to the proof of query lower bound for the OVGH [5, Proposition 14]. We emphasize that the distribution of $\boldsymbol{A}$ is changed in the optimization procedure but also in OVGH as a result.

**Proposition C.2.** *Let $k \ge 20 \frac{M + 3d \log(2d) + 1}{n \log_2(\sqrt{2}(\zeta + \delta'\sqrt{d}))^{-1}}$. And let $0 < \alpha, \beta \le 1$ such that $\alpha(\sqrt{d}/\beta)^{5/4} \le \zeta/\sqrt{d}$ where $\zeta \le 1$. If the Player wins the adapted OVGH with probability at least $1/2$, then $m \ge \frac{1}{8}(1 + \frac{30 \log_2 d}{\log_2(\sqrt{2}(\zeta + \delta'\sqrt{d}))^{-1}})^{-1} d$.*

*Proof.* We use the same proof and only highlight the modifications. The proof is unchanged until the step when the concentration result Lemma C.1 is used. Instead, we use Lemma C.2. With the same notations as in the original proof, we constructed $\lceil k/5 \rceil$ orthonormal vectors $\boldsymbol{Z} = [\boldsymbol{z}_1, \ldots, \boldsymbol{z}_{\lceil k/5 \rceil}]$ such that all rows $\boldsymbol{a}$ of $\boldsymbol{A}'$ (which is $\boldsymbol{A}$ up to some observed and unimportant rows) one has

$$\|\boldsymbol{Z}^\top \boldsymbol{a}\|_\infty \le \frac{\zeta}{\sqrt{d}}.$$

Next, by Lemma C.2, we have

$$\left| \left\{ \boldsymbol{a} \in \mathcal{D}_{\delta'} : \|\boldsymbol{Z}^\top \boldsymbol{a}\|_\infty \le \frac{\zeta}{\sqrt{d}} \right\} \right| \le |\mathcal{D}_{\delta'}| \cdot \mathbb{P}_{\boldsymbol{a} \sim \mathcal{U}_{\delta'}} \left( \|\boldsymbol{Z}^\top \boldsymbol{a}\|_\infty \le \frac{\zeta}{\sqrt{d}} \right)$$

$$\le |\mathcal{D}_{\delta'}| \cdot \mathbb{P}_{\boldsymbol{z} \sim \mathcal{U}(S^{d-1})} \left( \|\boldsymbol{Z}^\top \boldsymbol{z}\|_\infty \le \frac{\zeta}{\sqrt{d}} + \delta' \right)$$

$$\le |\mathcal{D}_{\delta'}| \cdot \left( \sqrt{2}(\zeta + \delta'\sqrt{d}) \right)^{\lceil k/5 \rceil}.$$

Hence, using the same arguments as in the original proof, we obtain

$$H(\boldsymbol{A}' \mid \boldsymbol{Y}) \le (n-m) \left( \log_2 |\mathcal{D}_{\delta'}| + \mathbb{P}(\mathcal{E}) \cdot \frac{k}{5} \log_2 \left( \sqrt{2}(\zeta + \delta'\sqrt{d}) \right) \right),$$

where $\mathcal{E}$ is the event when the algorithm succeeds at the OVGH game. In the next step, we need to bound $H(\boldsymbol{A} \mid \boldsymbol{V}) - H(\boldsymbol{G}, \boldsymbol{j}, \boldsymbol{c})$ where $\boldsymbol{V}$ stores hints received throughout the game, $\boldsymbol{G}$ stores observed rows of $\boldsymbol{A}$ during the game, and $\boldsymbol{j}, \boldsymbol{c}$ are auxiliary variables. The latter can be treated as in the original proof. We obtain

$$H(\boldsymbol{A} \mid \boldsymbol{V}) - H(\boldsymbol{G}, \boldsymbol{j}, \boldsymbol{c}) \ge H(\boldsymbol{A}) - H(\boldsymbol{G}) - I(\boldsymbol{A}; \boldsymbol{V}) - 3m \log_2(2d)$$
$$\ge (n-m) \log_2 |\mathcal{D}_{\delta'}| - 3m \log_2(2d) - I(\boldsymbol{A}, \boldsymbol{V}).$$

Now the same arguments as in the original proof show that we still have $I(\boldsymbol{A}, \boldsymbol{V}) \le 3km \log_2 d + 1$, and that as a result, if $M$ is the number of bits stored in memory,

$$M \ge \frac{k}{10} \log_2 \left( \frac{1}{\sqrt{2}(\zeta + \delta'\sqrt{d})} \right) (n-m) - 3km \log_2 d - 1 - 3d \log_2(2d).$$

Then, with the same arguments as in the original proof, we can conclude. $\qquad\square$

We are now ready to prove Theorem C.1. With the parameter $k = \lceil 20 \frac{M+3d\log(2d)+1}{n\log_2(\sqrt{2}(\epsilon d^4/2+\delta'\sqrt{d}))^{-1}} \rceil$ and the same arguments, we show that an algorithm solving the convex optimization up to precision $\eta/(2\sqrt{d}) = \epsilon$ yields an algorithm solving the OVGH where the parameters $\alpha = \frac{2\eta}{\gamma_1}$ and $\beta = \frac{\gamma_2}{4}$ satisfy

$$\alpha \left( \frac{\sqrt{d}}{\beta} \right)^{5/4} \leq \frac{\eta d^3}{4} = \frac{d^{3.5}\epsilon}{2}.$$

We can then apply Proposition C.2 with $\zeta = d^4\epsilon/2$. Hence, if $Q$ is the maximum number of queries of the convex optimization algorithm, we obtain

$$\lceil Q/p_{max} \rceil + 1 \geq \frac{1}{8} \left( 1 + \frac{30\log_2 d}{\log_2 \frac{1}{d^4\epsilon} - 1/2} \right)^{-1} d \geq \frac{d}{8 \cdot 61},$$

where in the last inequality we used $\epsilon \leq 1/(2d^{4.5})$. As a result, with the same arguments, we obtain

$$Q = \Omega \left( \frac{d^{5/3}\ln^{1/3}\frac{1}{\epsilon}}{(M+\ln d)^{1/3}\ln^{2/3} d} \right).$$

This ends the proof of Theorem C.1.

### C.1.2 Lower bound of [5] for feasibility problems and deterministic algorithms

We improve the memory dependence by showing the following result.

**Theorem C.3.** *For $\epsilon = 1/(48d^3)$ and any $\delta \in [0, 1]$, a deterministic algorithm guaranteed to solve the feasibility problem over the unit ball with $\epsilon$ accuracy uses at least $d^{2-\delta}\ln\frac{1}{\epsilon}$ bits of memory or makes at least $\tilde{\Omega}(d^{1+\delta})$ queries.*

We use the exact same class of feasibility problems and only change the parameter $\eta_0$ which constrained successful points to satisfy $\|\boldsymbol{Ax}\|_\infty \leq \eta_0$, as well as the rows of $\boldsymbol{A}$ that are sampled i.i.d. from $\mathcal{U}_\delta$. The other parameter $\eta_1 = 1/(2\sqrt{d})$ is unchanged. We also take $\delta' = \epsilon$. Because the rows of $\boldsymbol{A}$ are already normalized, we can take $\eta_0 = \epsilon$ directly. Then, the same proof as in [5] shows that if an algorithm solves feasibility problems with accuracy $\epsilon$, there is an algorithm for OVGH for parameters $\alpha = \eta/\eta_1$ and $\beta = \eta_1/2$. Then, we have $\alpha(\sqrt{d}/\beta)^{5/4} \leq 12d^2\eta_0$ and we can apply Proposition C.2 with $\zeta = 12d^{2.5}\eta_0 = 12d^{2.5}\epsilon$. Similar computations as above then show that $m \geq d/(8 \cdot 61)$, with $k = \Theta(\frac{M+\ln d}{d\ln\frac{1}{\epsilon}})$, so that the query lower bound finally becomes

$$Q \geq \Omega \left( \frac{d^3\ln\frac{1}{\epsilon}}{(M+\ln d)\ln^2 d} \right).$$

**Remark C.1.** *The more careful analysis—involving the discretization $\mathcal{D}_\delta$ of the unit sphere at scale $\delta$ instead of the hypercube $\{\pm 1\}^d$—allowed to add a $\ln\frac{1}{\epsilon}$ factor to the final query lower bound but also an additional $\ln d$ factor for both convex-optimization and feasibility-problem results. Indeed, the improved Proposition C.2 shows that the OVGH with adequate parameters requires $\mathcal{O}(d)$ queries, instead of $\mathcal{O}(d/\ln d)$ in [5, Proposition 14]. At a high level, each hint queried brings information $\mathcal{O}(d\ln d)$ but memorizing a binary matrix $\boldsymbol{A} \in \{\pm 1\}^{\lceil d/4 \rceil \times d}$ only requires $d^2$ bits of memory: hence the query lower bound is limited to $\mathcal{O}(d/\ln d)$. Instead, memorizing the matrix $\boldsymbol{A}$ where each row lies in $\mathcal{D}_\delta$ requires $\Theta(d^2\ln\frac{1}{\epsilon})$ memory, hence querying $d$ hints (total information $\mathcal{O}(d^2\ln d)$) is not prohibitive for the lower bound.*

### C.1.3 Lower bound of [31] for convex optimization and randomized algorithms

We aim to improve the result to obtain the following.

**Theorem C.4.** *For $\epsilon \leq 1/d^4$ and any $\delta \in [0, 1]$, any (potentially randomized) algorithm guaranteed to minimize 1-Lipschitz convex functions over the unit ball with $\epsilon$ accuracy uses at least $d^{1.25-\delta}\ln\frac{1}{\epsilon}$ bits of memory or makes $\tilde{\Omega}(d^{1+4\delta/3})$ queries.*

The distribution considered in [31] is given by the functions

$$\frac{1}{d^6} \max\left\{ d^5 \|\boldsymbol{A}\boldsymbol{x}\|_\infty - 1, \max_{i \in [N]}(\boldsymbol{v}_i^\top \boldsymbol{x} - i\gamma) \right\},$$

where $N \leq d$ is a parameter, $\boldsymbol{A}$ has $\lfloor d/2 \rfloor$ rows drawn i.i.d. from $\mathcal{U}(\{\pm 1\}^d)$, and the vectors $\boldsymbol{v}_i$ are drawn i.i.d. from the rescaled hypercube $\boldsymbol{v}_i \sim \mathcal{U}(d^{-1/2}\{\pm 1\}^d)$. We adapt the class of functions by simply changing pre-factors as follows

$$\mu \max\left\{ \frac{1}{\mu} \|\boldsymbol{A}\boldsymbol{x}\|_\infty - 1, \max_{i \in [N]}(\boldsymbol{v}_i^\top \boldsymbol{x} - i\gamma) \right\}, \tag{8}$$

where $\boldsymbol{A}$ has the same number of rows but they are draw i.i.d. from $\mathcal{U}_\delta$, and $\delta, \mu > 0$ are parameters to specify. We use the notation $\mu$ instead of $\eta$ as in the previous sections because [31] already use a parameter $\eta$ which in our context can be interpreted as $\eta = 1/(\mu\sqrt{d})$. We choose the parameters $\mu = 16\sqrt{d}\epsilon$ and $\delta' = \epsilon$.

Again, as for the previous sections, the original proof can be directly used to show that if an algorithm is guaranteed to find a $\frac{\mu}{16\sqrt{N}} (\geq \epsilon)$-suboptimal point for the above function class, there is an algorithm that wins at their Orthogonal Vector Game (OVG) [31, Game 1], with the only difference that the parameter $d^{-4}$ (l.8 of OVG) is replaced by $\sqrt{d}\mu$. OVG requires the output to be *robustly-independent* (defined in [31]) and effectively corresponds to $\beta = 1/d^2$ in OVGH. As a result, there is a successful algorithm for the OVGH with parameters $\alpha = \sqrt{d}\mu$ and $\beta = 1/d^2$ and that even completely ignores the hints. Hence, we can now directly use Proposition C.2 with $\zeta = d^{1+25/16}\mu$ (from the assumption $\epsilon \leq d^{-4}$ we have $\zeta \leq 1/\sqrt{d}$). This shows that with the adequate choice of $k = \Theta(\frac{M+d\ln d}{d\ln\frac{1}{\epsilon}})$, the query lower bound is $\Omega(d)$.

Putting things together, a potentially randomized algorithm for convex optimization that uses $M$ memory makes at least the following number of queries

$$Q \geq \Omega\left(\frac{Nd}{k}\right) = \Omega\left(\frac{d^{4/3}}{\ln^{1/3} d}\left(\frac{d\ln\frac{1}{\epsilon}}{M+d\ln d}\right)^{4/3}\right).$$

### C.2 Proof sketch for improving the query-complexity lower bound

We now turn to improving the query-complexity lower bound by a factor $\frac{\ln\frac{1}{\epsilon}}{\ln d}$. At the high level, the idea is to replicate these constructed "difficult" class of functions at $\frac{\ln\frac{1}{\epsilon}}{\ln d}$ different scales or levels, similarly to the manner that the historical $\Omega(d\ln\frac{1}{\epsilon})$ lower bound is obtained for convex optimization [34]. This argument is relatively standard and we only give details in the context of improving the bound from [31] for randomized algorithms in convex optimization for conciseness. This result uses a simpler class of functions, which greatly eases the exposition. We first present the construction with 2 levels, then present the generalization to $p = \Theta(\frac{\ln\frac{1}{\epsilon}}{\ln d})$ levels. For convenience, we write

$$Q(\epsilon; M, d) = \Omega\left(\frac{d^{4/3}}{\ln^{1/3} d}\left(\frac{d\ln\frac{1}{\epsilon}}{M+d\ln d}\right)^{4/3}\right).$$

This is the query lower bound given in Theorem C.5 for convex optimization algorithms with memory $M$ that optimize the defined class of functions (Eq (8)) to accuracy $\epsilon$.

#### C.2.1 Construction of a bi-level class of functions $F_{\boldsymbol{A},\boldsymbol{v}_1,\boldsymbol{v}_2}$ to optimize

In the lower-bound proof, [31] introduce the point

$$\bar{\boldsymbol{x}} = -\frac{1}{2\sqrt{N}} \sum_{i \in [N]} P_{\boldsymbol{A}^\perp}(\boldsymbol{v}_i),$$

where $P_{\boldsymbol{A}^\perp}$ is the projection onto the orthogonal space to the rows of $\boldsymbol{A}$. They show that with failure probability at most $2/d$, $\bar{\boldsymbol{x}}$ has good function value

$$F_{\boldsymbol{A},\boldsymbol{v}}(\bar{\boldsymbol{x}}) := \mu \max\left\{ \frac{1}{\mu} \|\boldsymbol{A}\bar{\boldsymbol{x}}\|_\infty - 1, \max_{i \in [N]}(\boldsymbol{v}_i^\top \bar{\boldsymbol{x}} - i\gamma) \right\} \leq -\frac{\mu}{8\sqrt{N}}.$$

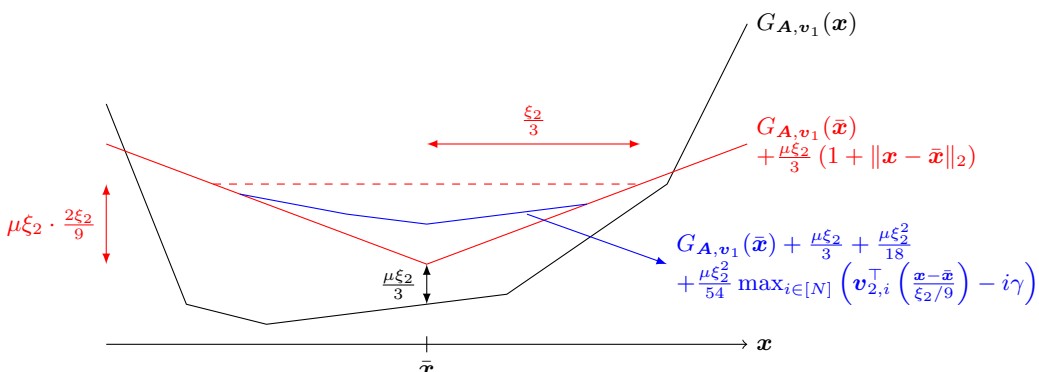

Figure 4: Representation of the procedure to rescale the optimization function.

This is shown in [31, Lemma 25]. On the other hand, from Theorem C.4, during the first

$$Q_1 = Q(\epsilon; M, d)$$

queries of any algorithm, with probability at least $1/3$, all queries are at least $\mu/(16\sqrt{N})$-suboptimal compared to $\bar{x}$ in function value [31, Theorem 28, Lemma 14 and Theorem 16]. Precisely, if $F_{A,v}$ is the sampled function to optimize, with probability at least $1/3$,

$$F_{A,v}(x_t) \geq F_{A,v}(\bar{x}) + \frac{\mu}{16\sqrt{N}} \geq F_{A,v}(\bar{x}) + \frac{\mu}{16\sqrt{d}}, \quad \forall t \leq Q_1.$$

As a result, we can replicate the term $\max_{i \in [N]}(v_i^\top x - i\gamma)$ at a smaller scale within the ball $B_d(\bar{x}, 1/(16\sqrt{d}))$. For convenience, we introduce $\xi_2 = 1/(16\sqrt{d})$ which will be the scale of the duplicate function. We separate the wall term $\|Ax\|_\infty - \mu$ for convenience. Hence, we define

$$G_{A,v_1}(x) := \mu \max_{i \in [N]}\left(v_{1,i}^\top x - i\gamma\right)$$

$$G_{A,v_1,v_2}(x) := \max\{G_{A,v^{(1)}}(x), G_{A,v_1}(\bar{x}) + \frac{\mu\xi_2}{3} \cdot$$

$$\max\left\{1 + \|x - \bar{x}\|_2, 1 + \frac{\xi_2}{6} + \frac{\xi_2}{18}\max_{i \in [N]}\left(v_{2,i}^\top\left(\frac{x - \bar{x}}{\xi_2/9}\right) - i\gamma\right)\right\}\}$$

An illustration of the construction is given in Fig. 4. The resulting optimization functions are given by adding the wall term:

$$F_{A,v_1}(x) = \max\left\{\|Ax\|_\infty - \mu, G_{A,v_1}(x)\right\}$$
$$F_{A,v_1,v_2}(x) = \max\left\{\|Ax\|_\infty - \mu, G_{A,v_1,v_2}(x)\right\}$$

We first explain the choice of parameters. First observe that since $\|A\bar{x}\| = 0$, we have $G_{A,v_1}(\bar{x}) = F_{A,v_1}(\bar{x})$. We can then check that for all $x \in B_d(0,1)$,

$$G_{A,v_1,v_2}(x) \leq \max\left\{G_{A,v_1}(x), G_{A,v_1}(\bar{x}) + \frac{2}{3}\mu\xi_2\right\}. \tag{9}$$

Further, for any $x \in B_d(\bar{x}, \xi_2/3)$, since $F_{A,v_1}$ is 1-Lipschitz, we can easily check that

$$G_{A,v_1,v_2}(x) - G_{A,v_1}(\bar{x})$$
$$= \frac{\mu\xi_2}{3}\max\left\{1 + \|x - \bar{x}\|_2, 1 + \frac{\xi_2}{6} + \frac{\xi_2}{18}\max_{i \in [N]}\left(v_{2,i}^\top\left(\frac{x - \bar{x}}{\xi_2/9}\right) - i\gamma\right)\right\} \leq \frac{2}{3}\mu\xi_2.$$

Thus, $G_{A,v_1,v_2}(x)$ does not coincide with $G_{A,v_1}(x)$ on $B_d(\bar{x}, \xi_2/3)$. Then, the $\|x - \bar{x}\|_2$ term ensures that any minimizer of $G_{A,v_1,v_2}$ is contained within the closed ball $B_d(\bar{x}, \xi_2/3)$. Also, to obtain a $\mu\xi_2/3$-suboptimal solution of $F_{A,v_1,v_2}$, the algorithm needs to find what would be a $\mu\xi_2$-suboptimal solution of $F_{A,v_1}$, while receiving the same response as when optimizing the latter.

Next, for any $\boldsymbol{x} \in B_d(\bar{\boldsymbol{x}}, \xi_2/9)$, the term $\max_{i \in [N]} \left( \boldsymbol{v}_{2,i}^\top \left( \frac{\boldsymbol{x} - \bar{\boldsymbol{x}}}{\xi_2/9} \right) - i\gamma \right)$ lies in $[-1, 1]$. Hence, we can check that for $\boldsymbol{x} \in B_d(\bar{\boldsymbol{x}}, \xi_2/9)$,

$$G_{\boldsymbol{A}, \boldsymbol{v}_1, \boldsymbol{v}_2}(\boldsymbol{x}) = G_{\boldsymbol{A}, \boldsymbol{v}_1}(\bar{\boldsymbol{x}}) + \frac{\mu\xi_2}{3} + \frac{\mu\xi_2^2}{18} + \frac{\mu\xi_2^2}{54} \max_{i \in [N]} \left( \boldsymbol{v}_{2,i}^\top \left( \frac{\boldsymbol{x} - \bar{\boldsymbol{x}}}{\xi_2/9} \right) - i\gamma \right). \qquad (10)$$

We now argue that $F_{\boldsymbol{A}, \boldsymbol{v}_1, \boldsymbol{v}_2}$ acts as a duplicate function. Until the algorithm reaches a point with function value at most $G_{\boldsymbol{A}, \boldsymbol{v}_1}(\bar{\boldsymbol{x}}) + \mu\xi_2$, the optimization algorithm only receives responses consistent with the function $F_{\boldsymbol{A}, \boldsymbol{v}_1}$ by Eq (9). Next, all minimizers of $F_{\boldsymbol{A}, \boldsymbol{v}_1, \boldsymbol{v}_2}$ are contained in $B_d(\bar{\boldsymbol{x}}, \xi_2/3)$, which was the goal of introducing the term in $\|\boldsymbol{x} - \bar{\boldsymbol{x}}\|_2$. As a result, optimizing $F_{\boldsymbol{A}, \boldsymbol{v}_1, \boldsymbol{v}_2}$ on this ball is equivalent to minimizing

$$\tilde{F}_{\boldsymbol{A}, \boldsymbol{v}_2}(\boldsymbol{y}) = \max \left\{ \|\boldsymbol{A}\boldsymbol{y}\|_\infty - \mu_2, c_2 + \nu_2 \max_{i \in [N]}(\boldsymbol{v}_{2,i}^\top \boldsymbol{y} - i\gamma), c_2' + \nu_2' \|\boldsymbol{y}\| \right\}, \quad \boldsymbol{y} \in B_d(0, 3),$$

where $\boldsymbol{y} = \frac{\boldsymbol{x} - \bar{\boldsymbol{x}}}{\xi_2/9}$. The function has been rescaled by a factor $\xi_2/9$ compared to $F_{\boldsymbol{A}, \boldsymbol{v}_1, \boldsymbol{v}_2}$ so that $\mu_2 = \frac{9\mu}{\xi_2}$, $\nu_2 = \frac{\mu\xi_2}{6}$, $\nu_2' = 6\mu$, $c_2 = \frac{9}{\xi_2} G_{\boldsymbol{A}, \boldsymbol{v}_1}(\bar{\boldsymbol{x}}) + 3\mu + \frac{\mu\xi_2}{2}$, and $c_2' = \frac{9}{\xi_2} G_{\boldsymbol{A}, \boldsymbol{v}_1}(\bar{\boldsymbol{x}}) + 3\mu$. By Eq (10), the two first terms of $\tilde{F}_{\boldsymbol{A}, \boldsymbol{v}_1}$ are preponderant for $\boldsymbol{y} \in B_d(0, 1)$.

The form of $\tilde{F}_{\boldsymbol{A}, \boldsymbol{v}_2}$ is very similar to the original form of functions

$$F_{\boldsymbol{A}, \boldsymbol{v}_2} = \max \left\{ \|\boldsymbol{A}\boldsymbol{y}\|_\infty - \mu_1', \mu_2' \max_{i \in [N]}(\boldsymbol{v}_{2,i}^\top \boldsymbol{y} - i\gamma) \right\},$$

In fact, the same proof structure for the query-complexity/memory lower-bound can be applied in this case. The main difference is that originally one had $\mu_1' = \mu_2'$; here we would instead have $\mu_1' = \mu_2 + c_2 = \Theta(\mu/\xi_2)$ and $\mu_2' = \nu_2 = \Theta(\mu\xi_2)$. Intuitively, this corresponds to increasing the accuracy to $\Theta(\epsilon\xi_2^2)$—a factor $\xi_2$ is due to the fact that $\tilde{F}_{\boldsymbol{A}, \boldsymbol{v}_2}$ was rescaled by a factor $\xi_2/9$ compared to $F_{\boldsymbol{A}, \boldsymbol{v}_1, \boldsymbol{v}_2}$, and a second factor $\xi_2$ is due to the fact that within $\tilde{F}_{\boldsymbol{A}, \boldsymbol{v}_2}$, we have $\mu_2' = \Theta(\mu\xi_2)$—while the query lower bound is similar to that obtained for $\Theta(\epsilon/\xi_2)$. As a result, during the first

$$Q_2 = Q\left( \Theta\left( \frac{\epsilon}{\xi_2} \right); M, d \right)$$

queries of any algorithm optimizing $\tilde{F}_{\boldsymbol{A}, \boldsymbol{v}_2}$, with probability at least $1/3$ on the sample of $\boldsymbol{A}$ and $\boldsymbol{v}_2$, all queries are at least $\Theta(\epsilon\xi_2)$-suboptimal compared to

$$\bar{\boldsymbol{y}} = -\frac{1}{2\sqrt{N}} \sum_{i \in [N]} P_{\boldsymbol{A}^\perp}(\boldsymbol{v}_{2,i}).$$

We are now ready to give lower bounds on the queries of an algorithm minimizing $F_{\boldsymbol{A}, \boldsymbol{v}_1, \boldsymbol{v}_2}$ to accuracy $\Theta(\epsilon\xi_2^2)$. Let $T_2$ be the index of the first query with function value at most $G_{\boldsymbol{A}, \boldsymbol{v}_1}(\bar{\boldsymbol{x}}) + \mu\xi_2$. We already checked that before that query, all responses of the oracle are consistent with minimizing $F_{\boldsymbol{A}, \boldsymbol{v}_1}$, hence on an event $\mathcal{E}_1$ of probability at least $1/3$, one has $T_2 \geq Q_1$. Next, consider the hypothetical case when at time $T_2$, the algorithm is also given the information of $\bar{\boldsymbol{x}}$ and is allowed to store this vector. Given this information, optimizing $F_{\boldsymbol{A}, \boldsymbol{v}_1, \boldsymbol{v}_2}$ reduces to optimizing $\tilde{F}_{\boldsymbol{A}, \boldsymbol{v}_2}$ since we already know that the minimum is achieved within $B_d(\bar{\boldsymbol{x}}, \xi_2/3)$. Further, any query outside of this ball either

- returns a vector $\boldsymbol{v}_{1,i}$ which does not give any useful information for the minimization ($\boldsymbol{v}_1$ and $\boldsymbol{v}_2$ are sampled independently and $\bar{\boldsymbol{x}}$ is given),

- or returns a row from $\boldsymbol{A}$, as covered by the original proof.

Hence, on an event $\mathcal{E}_2$ of probability at least $1/3$, even with the extra information of $\bar{\boldsymbol{x}}$, during the next $Q_2$ queries starting from $T_2$, the algorithm does not query a $\Theta(\mu\xi_2^3)$−suboptimal solution to $F_{\boldsymbol{A}, \boldsymbol{v}_1, \boldsymbol{v}_2}$. This holds a fortiori for the model when the algorithm is not given $\bar{\boldsymbol{x}}$ at time $T_2$.

### C.2.2 Recursive construction of a $p$-level class of functions $F_{\boldsymbol{A},\boldsymbol{v}_1,\ldots,\boldsymbol{v}_p}$

Similarly as in the last section, one can inductively construct the sequence of functions $F_{\boldsymbol{A},\boldsymbol{v}_1}$, $F_{\boldsymbol{A},\boldsymbol{v}_1,\boldsymbol{v}_2}$, $F_{\boldsymbol{A},\boldsymbol{v}_1,\boldsymbol{v}_2,\boldsymbol{v}_3}$, etc. Formally, the induction is constructed as follows: let $(\boldsymbol{v}_p)_{p\geq 1}$ be an i.i.d. sequence of $N$ i.i.d. vectors $(\boldsymbol{v}_{k,i})_{i\in[N]}$ sampled from the rescaled hypercube $d^{-1/2}\{\pm 1\}^d$. Next, we pose

$$G_{\boldsymbol{A},\boldsymbol{v}_1}(\boldsymbol{x}) = \mu^{(1)} \max_{i\in[N]} \left( \boldsymbol{v}_{1,i}^\top \left( \frac{\boldsymbol{x} - \bar{\boldsymbol{x}}^{(1)}}{s^{(1)}} \right) - i\gamma \right),$$

where $\mu^{(1)} = \mu$, $\bar{\boldsymbol{x}}^{(1)} = \boldsymbol{0}$ and $s^{(1)} = 1$. For $k \geq 1$, we pose

$$\bar{\boldsymbol{x}}^{(k+1)} = \bar{\boldsymbol{x}}^{(k)} - \frac{s^{(k)}}{2\sqrt{N}} \sum_{i\in[N]} P_{\boldsymbol{A}^\perp}(\boldsymbol{v}_{k,i}), \quad \text{and} \quad F^{(k)} := G_{\boldsymbol{A},\boldsymbol{v}_1,\ldots,\boldsymbol{v}_k}(\bar{\boldsymbol{x}}^{(k)}) + \mu^{(k)}\xi_{k+1},$$

for a certain parameter $\xi_{k+1}$ to be specified. We then define the next level as

$$G_{\boldsymbol{A},\boldsymbol{v}_1,\ldots,\boldsymbol{v}_{k+1}}(\boldsymbol{x}) := \max \left\{ G_{\boldsymbol{A},\boldsymbol{v}_1,\ldots,\boldsymbol{v}_k}(\boldsymbol{x}), G_{\boldsymbol{A},\boldsymbol{v}_1,\ldots,\boldsymbol{v}_k}(\bar{\boldsymbol{x}}^{(k+1)}) + \frac{\mu^{(k)}\xi_{k+1}}{3} \cdot \right.$$
$$\left. \max\left\{ 1 + \frac{\|\boldsymbol{x} - \bar{\boldsymbol{x}}^{(k+1)}\|_2}{s^{(k)}}, 1 + \frac{\xi_{k+1}}{6} + \frac{\xi_{k+1}}{18} \max_{i\in[N]} \left( \boldsymbol{v}_{k+1,i}^\top \left( \frac{\boldsymbol{x} - \bar{\boldsymbol{x}}^{(k+1)}}{s^{(k)}\xi_{k+1}/9} \right) - i\gamma \right) \right\} \right\}.$$

We then pose $\mu^{(k+1)} := \mu^{(k)}\xi_{k+1}^2/54$ and $s^{(k+1)} := s^{(k)}\xi_{k+1}/9$, which closes the induction. The optimization functions are defined simply as

$$F_{\boldsymbol{A},\boldsymbol{v}_1,\ldots,\boldsymbol{v}_{k+1}}(\boldsymbol{x}) = \max\left\{ \|\boldsymbol{A}\boldsymbol{x}\|_\infty - \mu, G_{\boldsymbol{A},\boldsymbol{v}_1,\ldots,\boldsymbol{v}_{k+1}}(\boldsymbol{x}) \right\}.$$

We checked before that we can use $\xi_2 = 1/(16\sqrt{d})$. For general $k \geq 0$, given that the form of the function slightly changes to incorporate the absolute term (see $\tilde{F}_{\boldsymbol{A},\boldsymbol{v}_2}$), this constant may differ slightly. In any case, one has $\xi_k = \Theta(1/\sqrt{d})$. Now fix a construction level $p \geq 1$ and for any $k \in [p]$, let $T_k$ be the first time that a point with function value at most $F^{(k)}$ is queried. For convenience let $T_0 = 0$. Using the same arguments as above recursively, we can show that on an event $\mathcal{E}_k$ with probability at least $1/3$,

$$T_k - T_{k-1} \geq Q_k = Q\left( \Theta\left( \frac{\mu}{s^{(k)}} \right); M, d \right)$$

Next note that the sequence $F^{(k)}$ is decreasing and by construction, if one finds a $\mu^{(p)}\xi_{p+1}$-suboptimal point of $F_{\boldsymbol{A},\boldsymbol{v}_1,\ldots,\boldsymbol{v}_p}$, then this point has value at most $F^{(p)}$. As a result, for an algorithm that finds a $\mu^{(p)}\xi_{p+1}$-suboptimal point, the times $T_0, \ldots, T_p$ are all well defined and non-decreasing. We recall that $\mu = \Theta(\sqrt{d}\epsilon)$. Therefore, we can still have $\mu/s^{(p)} \leq \sqrt{\epsilon}$ and $\mu^{(p)}\xi_{p+1} \geq \epsilon^2$ for $p = \Theta(\frac{\ln\frac{1}{\epsilon}}{\ln d})$. Combining these observations, we showed that when optimizing the functions $F_{\boldsymbol{A},\boldsymbol{v}_1,\ldots,\boldsymbol{v}_p}$ to accuracy $\Theta(\mu^{(p)}\xi_{p+1}) = \Omega(\epsilon^2)$, the total number of queries $Q$ satisfies

$$\mathbb{E}[Q] \geq \frac{1}{3} \sum_{k\in[p]} Q_k \geq \frac{p}{3} Q(\sqrt{\epsilon}; M, d) = \Theta\left( \frac{d^{4/3}\ln\frac{1}{\epsilon}}{\ln^{4/3} d} \left( \frac{d\ln\frac{1}{\epsilon}}{M + d\ln d} \right)^{4/3} \right).$$

Changing $\epsilon$ to $\epsilon^2$ proves the desired result.

**Theorem C.5.** *For $\epsilon \leq 1/d^8$ and any $\delta \in [0,1]$, any (potentially randomized) algorithm guaranteed to minimize 1-Lipschitz convex functions over the unit ball with $\epsilon$ accuracy uses at least $d^{1.25-\delta}\ln\frac{1}{\epsilon}$ bits of memory or makes $\tilde{\Omega}(d^{1+4\delta/3}\ln\frac{1}{\epsilon})$ queries.*

The same recursive construction can be applied to the results from Theorems C.1 and C.3 to improve their oracle-complexity lower bounds by a factor $\frac{\ln\frac{1}{\epsilon}}{\ln d}$, albeit with added technicalities due to the adaptivity of their class of functions. This yields Theorem 3.3.

**Input:** Number of iterations $T$, computation accuracy $\eta \leq 1$, target accuracy $\epsilon \leq 1$
Initialize: $\boldsymbol{x} = \boldsymbol{0}$;
**for** $t = 0, \ldots, T$ **do**
  Query the oracle at $\boldsymbol{x}$
  **if** $\boldsymbol{x}$ *successful* **then return** $x$;
  Receive a separation vector $\boldsymbol{g}$ with accuracy $\eta$
  Update $\boldsymbol{x}$ as $\boldsymbol{x} - \epsilon \boldsymbol{g}$ up to accuracy $\eta$
**end**
**return** $x$

**Algorithm 11:** Memory-constrained gradient descent

## D    Memory-constrained gradient descent for the feasibility problem

In this section, we prove a simple result showing that memory-constrained gradient descent applies to the feasibility problem. We adapt the algorithm described in [52].

We now prove that this memory-constrained gradient descent gives the desired result of Proposition 3.1.

*Proof of Proposition 3.1.* Denote by $\boldsymbol{x}_t$ the state of $\boldsymbol{x}$ at iteration $t$, and $\boldsymbol{g}_t$ (resp. $\tilde{\boldsymbol{g}}_t$) the separation oracle without rounding errors (resp. with rounding errors) at $\boldsymbol{x}_t$. By construction,

$$\|\boldsymbol{x}_{t+1} - (\boldsymbol{x}_t + \epsilon\tilde{\boldsymbol{g}}_t)\| \leq \eta \quad \text{and} \quad \|\tilde{\boldsymbol{g}}_t - \boldsymbol{g}_t\| \leq \eta. \tag{11}$$

As a result, recalling that $\|\boldsymbol{g}_t\| = 1$,

$$\|\boldsymbol{x}_{t+1} - \boldsymbol{x}^\star\|^2 \leq (\|\boldsymbol{x}_t + \epsilon\tilde{\boldsymbol{g}}_t - \boldsymbol{x}^\star\| + \eta)^2 \leq (\|\boldsymbol{x}_t + \epsilon\boldsymbol{g}_t - \boldsymbol{x}^\star\| + (1+\epsilon)\eta)^2 \leq \|\boldsymbol{x}_t + \epsilon\boldsymbol{g}_t - \boldsymbol{x}^\star\|^2 + 20\eta.$$

By assumption, $Q$ contains a ball $B_d(\boldsymbol{x}^\star, \epsilon)$ for $\boldsymbol{x}^\star \in B_d(0,1)$. Then, because $\boldsymbol{g}_t$ separates $\boldsymbol{x}_t$ from $B_d(\boldsymbol{x}^\star, \epsilon)$, one has $\boldsymbol{g}_t^\top(\boldsymbol{x}^\star - \boldsymbol{x}_t) \geq \epsilon$. Therefore,

$$\|\boldsymbol{x}_{t+1} - \boldsymbol{x}^\star\|^2 \leq \|\boldsymbol{x}_t - \boldsymbol{x}^\star\|^2 + 2\epsilon\boldsymbol{g}_t^\top(\boldsymbol{x}_t - \boldsymbol{x}^\star) + \epsilon^2\|\boldsymbol{g}_t\|^2 + 20\eta$$
$$\leq \|\boldsymbol{x}_t - \boldsymbol{x}^\star\|^2 - \epsilon^2 + 20\eta.$$

Then, take $\eta = \epsilon^2/40$ and $T = \frac{8}{\epsilon^2}$. If iteration $T$ was performed, we have using the previous equation

$$\|\boldsymbol{x}_T - \boldsymbol{x}^\star\|^2 \leq \|\boldsymbol{x}_0 - \boldsymbol{x}^\star\|^2 - \frac{\epsilon^2}{2}T \leq 4 - \frac{\epsilon^2}{2}T \leq 0.$$

Hence, $\boldsymbol{x}_T$ is an $\epsilon$-suboptimal solution.

We now turn to the memory usage of gradient descent. It only needs to store $\boldsymbol{x}$ and $\boldsymbol{g}$ up to the desired accuracy $\eta = \mathcal{O}(\epsilon^2)$. Hence, this storage and the internal computations can be done with $\mathcal{O}(d \ln \frac{d}{\epsilon})$ memory. Because we suppose that $\epsilon \leq \frac{1}{\sqrt{d}}$, this gives the desired result. $\square$

