# OpenReview forum: "Memory-Constrained Algorithms for Convex Optimization"
_NeurIPS.cc/2023/Conference — NeurIPS 2023 poster_

### Official Review · Reviewer_MixU · 2023-07-06

**Soundness:** 3 good
**Presentation:** 3 good
**Contribution:** 3 good
**Rating:** 6
**Confidence:** 4

**Summary:**

The main result of the paper is a theorem that establishes a trade-off between oracle-complexity and memory-usage for convex optimization problems. More specifically they show that given a convex optimization problem in R^d, then for any p between 1 and d there is a deterministic first-order algorithm tha solves the feasibility problem for accuracy \epsilon <=1/sqrt{d} using O(d^2/p ln 1 \epsilon) bits of memory with O((C (d/p) (ln 1/epsilon))^p) calls to the separation oracle.  In other words, by decreasing the number of bits of memory by a factor of p, one needs to call the oracle a number of times that is exponential in p. The basis of the exponential function is roughly (d/p)* ln(1/epsilon).

**Strengths:**

The papers seems to make relevant advances in the understanding of tradeoffs between memory and number of oracle calls in convex optimization algorithms.

**Weaknesses:**

The technique is heavily based on the Vaidya's cutting-plane method. It is not entirely clear what are really the advancements with respect to the original technique.

There are some minor presentation issues, such as statements of results with undefined parameters or parameters that are defined very far away.

**Questions:**

1) Please make theorem 3.2 more self contained by briefly describing two parameters before the statement of the theorem: d and omega.

It should not be necessary to read a big portion of the paper to be able to understand the meaning of the main statement (Theorem 3.2). For instance, you could add a line before the theorem stating that you are dealing with optimization problems in R^d. Same thing for omega. I only see it defined in the footnote of page 4. Although the notation is standard, it took some time to find the precise meaning of omega. So it would be good to write its meaning either in the theorem or in the text just before it.

2) What are the differences between your method and Vaidya's method? Is it the case that the only difference relies in a recursive application of Vaidya's method? Do you modify the method somehow? This should be better discussed in the introduction.

**Limitations:**

In my opinion the paper discusses the limitations of the techniques in a fair way.

---

> ### Author Rebuttal · Authors · 2023-08-07
>
> Thank you for reading our paper carefully and for the positive views! For your first point, we agree that our algorithm relies heavily on Vaidya's cutting-plane method [1] and its variant proposed in [2]. We will provide more detailed comparisons between our algorithm and (the variant of) Vaidya's methods in the introduction. Here, we would like to emphasize advancements from the following two perspectives.
>
> - We use Vaidya's methods as dimension reduction techniques. In particular, our algorithm divides the variables into blocks and applies Vaidya's methods recursively to construct approximate separation or subgradient oracles for problems with reduced dimensions. We believe that there is some novelty in this recursive dimension reduction step.
> - An important and novel technical step in the convergence analysis is to ensure that the precision of the computed approximate separation oracles is sufficient, which is primordial given the recursive nature of our algorithms (the errors accumulate at each layer of recursion). A natural approach to having a separation oracle for a father problem is to ensure that Vaidya's method of the sub-problem converges to the ``minimizer'' of the sub-problem. In fact with this approach, the precision required is multiplicative in the depth of the recursion and in the end, the required memory to store such high-precision sub-gradients (for low-level sub-problems) blows up and does not yield improvements over standard cutting-plane methods. Instead, Algorithm 2 performs Vaidya's method to high accuracy on all sub-problems (even though the sub-gradients used are of lower quality): on the sub-problem, the iterates effectively do not converge to the true minimizer of the sub-problem, but the analysis shows that the corresponding aggregated subgradient at the higher-level is still of sufficient quality (the deterioration becomes additive in the number of recursive levels). We did not emphasize this in the main body for the sake of simplicity, but this is the reason the proof requires estimating approximation errors along the complete computation path, instead of using a level-by-level recursive approach.
> - We take into account memory throughout computations. To be precise, for the original Vaidya's method in [1], only the weaker notion of memory constraint (Definition 2.1) applies. By adopting the variant of Vaidya's method in [2] (which has more penalty terms in the potential function), our algorithm is proven to satisfy the stronger notion of memory constraint (Definition 2.2), i.e. the memory is constrained not just between oracle calls but also throughout computations.
>
> Thank you for pointing out the presentation issues, we apologize for any confusion! We will clarify $d$ (the dimension of the problem) and $\omega$ (the exponent of matrix multiplication, $\omega<2.373$) in Theorem 3.2.
>
>
>
>
>
> [1] Kurt M Anstreicher. “Towards a practical volumetric cutting plane method for convex pro- gramming”. In: SIAM Journal on Optimization 9.1 (1998), pp. 190–206.
>
> [2] Yin Tat Lee, Aaron Sidford, and Sam Chiu-wai Wong. “A faster cutting plane method and its implications for combinatorial and convex optimization”. In: 2015 IEEE 56th Annual Symposium on Foundations of Computer Science. IEEE. 2015, pp. 1049–1065.

---

### Official Review · Reviewer_H4w2 · 2023-07-10

**Soundness:** 3 good
**Presentation:** 2 fair
**Contribution:** 3 good
**Rating:** 7
**Confidence:** 3

**Summary:**

The paper studies the problem of memory constrained convex optimization. In this setting, the algorithm is given oracle access to the gradients of an unknown convex function and is tasked with finding an approximate minimizer. The additional constraint is that the algorithm is constrained to use as few bits of memory as possible. Though convex programming is a classical area, understanding the memory requirements of algorithms has only recently been considered and has become an active area of research. When considering memory, two standard algorithms witness a tradeoff between memory and oracle complexity: gradient descent makes $1/\epsilon^2$ queries and uses $d \log(1/\epsilon)$ bits of memory while the center of gravity method makes $d \log(1/\epsilon) $ queries and uses $d^2
log^2(1/ \epsilon)$ bits of memory. A recent line of study has been trying to understand the tradeoff between these complexities. The paper presents a recursive implementation of a cutting plane method which uses $O( d^2 / p \log (1/ \epsilon)$ memory while making $O( (d/p \log (1/ \epsilon))^p$ queries. Here $p \in [d]$ is a parameter of the algorithm. In particular, the paper shows that for $\epsilon = d^{-d}$ the algorithm presented has improved query complexity than gradient descent while using the same amount of memory.

**Strengths:**

The question considered by the paper is very interesting as memory constraints are a very natural "simplicity" condition on algorithms. To the best of my knowledge this is the first algorithm that improves on the memory complexity over the two benchmark algorithms. Furthermore the framework of recursive partitioning seems very interesting and could lead to improvements to other optimization algorithms,

**Weaknesses:**

The regime of parameters that the algorithms improves over gradient descent is rather quant. In this regime both the algorithms make $d^O(d)$ calls to the oracle and the current algorithm improves the constant in $O(d)$ in the exponent. While this is of mathematical interest and does show the non Pareto optimality of gradient descent, justifying the interestingness of this range of parameters seems difficult.

**Questions:**

- More thorough description of the algorithm along with the intuition for why it works would be very helpful.

**Limitations:**

Yes

---

> ### Author Rebuttal · Authors · 2023-08-07
>
> Thank you for the positive view and we really appreciate your suggestions! We agree that our algorithm improves over gradient descent only in the regime $\epsilon\leq \frac{1}{d^{\mathcal O(d)}}$, which might be smaller than regimes considered in some previous works on oracle complexity of convex optimization/feasibility problem. To justify studying this regime, we would like to point out here that optimization problems in the low-dimension regimes have a growing literature [1][2]; also, in the specific literature for memory/oracle-complexity tradeoffs, some papers specifically study super-polynomial accuracy regimes, e.g. [3] considers accuracies $\epsilon\leq \exp(-\log^4 d)$ (to appear at FOCS 2023 according to the author's website). In addition, we emphasize that our algorithm shows a non-trivial memory-query trade-off in the regime $\log(\frac{1}{\epsilon})\gg\log(d)$, and outperforms the center-of-mass in the standard regime $\epsilon\leq\frac{1}{\sqrt{d}}$, which are novel results that might help understand the memory-query landscape better.
>
> Following your suggestions, we will provide more detailed description of and intuition behind the algorithm. Here in Figure 1 (in the rebuttal pdf), we use a $2$-dimensional feasibility problem to illustrate the geometry of the recursive step, where an approximate separating hyperplane to the problem ``projected'' to the $x$-axis is constructed using a separating oracle for the $2$-dimensional problem. To be precise, the target is $\boldsymbol p^* = (p_x^*,p_y^*)$, and we use two blocks (i.e. $p=2$). Suppose at a step of the algorithm, the current value of the $x$ coordinate is $c$. We then aim to find an approximate separating hyperplane between $x = p_x^*$ and $x = c$ for some $c$, Algorithm 2 first runs Algorithm 1 (i.e. the memory-constrained Vaidya) to find two separating hyperplanes (the two blue hyperplanes). Lemma 4.1 then guarantees the existence of a convex combination of the 2 blue hyperplanes -- the red hyperplane-- which is approximately parallel to the $y$-axis and thus can serve as an approximate separating hyperplane between $x = p_x^*$ and $x = c$.
>
>
>
> [1] Vavasis, Stephen A. "Black-box complexity of local minimization." SIAM Journal on Optimization 3.1 (1993): 60-80.
>
> [2] Bubeck, Sébastien, and Dan Mikulincer. "How to trap a gradient flow." Conference on Learning Theory. PMLR, 2020.
>
> [3] Chen, Xi, and Binghui Peng. "Memory-Query Tradeoffs for Randomized Convex Optimization." arXiv preprint arXiv:2306.12534 (2023).

---

> > ### Comment · Reviewer_H4w2 · 2023-08-18
> >
> > Thanks for the response. I maintain my positive score.

---

### Official Review · Reviewer_U1uo · 2023-07-10

**Soundness:** 3 good
**Presentation:** 3 good
**Contribution:** 3 good
**Rating:** 6
**Confidence:** 2

**Summary:**

This paper studies solving feasibility problems, and hence optimization problems, and focusing on the trade-offs between the number of oracle calls and use of memory. The precise problem is to find a point in a convex set inside the unit cube, given access to a separation oracle, which reports if a query point is in the convex set, or otherwise returns a hyperplane separating the input query point and the convex set.

By segmenting the variables/dimensions and working on them sequentially, and using the variant of Vaidya’s cutting-plane method by Lee, Sidford, and Wong, this paper presents a new recursive algorithm with better trade-offs between the number of oracle calls and use of memory. In particular, this algorithm uses the same optimal memory but makes less oracle queries than gradient descent for low error regime $\epsilon \le \frac1{\sqrt d}$.

In addition to algorithms, this paper slightly improves on existing lower-bounds for the trade-offs between accuracy $\epsilon$ and dimension $d$ with a more careful analysis.

**Strengths:**

This paper adapts an existing algorithm with a recursive decomposition to improve the trade-offs between number of oracle calls and memory usage, beating gradient descent for very low error regime.

**Weaknesses:**

The improvement appears incremental, and does not give new insights or understanding for solving feasibility or optimization problems.

**Questions:**

Are there geometric intuition (better than the repeated convexity argument applied to blocks of variables) to the recursive application of cutting plane method? Arguably, breaking up variables into blocks, while improving bounds, does not give new geometric understanding.

**Limitations:**

This theoretical paper does not have broader societal impacts.

---

> ### Author Rebuttal · Authors · 2023-08-07
>
> Thank you for the positive view and the questions! Regarding the weakness, although our work does not provide a full understanding of the memory-query landscape, we emphasize the following three contributions.
>
> - To the best of our knowledge, prior to this work, no positive results nor algorithms were known to improve (in the memory-query landscape) over two most foundational algorithms for optimization: gradient descent (GD) and cutting-planes (CP). Our results demonstrate that in super-polynomial accuracy regimes, it is indeed possible to improve over these algorithms (dark green region in Figure 1). This is the first result making some progress on the algorithmic side as opposed to lower bounds for which the literature is better established [1,2,3].
> - We further provide a class of algorithms giving a positive trade-off between memory and oracle complexity. This enables an optimizer to specify a desired memory usage (at the price of oracle complexity), through the parameter $p$, instead of restricting the choice to either linear or full (quadratic) memory.
> - A major question was whether one can Pareto-improve over GD and/or CP. Lower bounds [2,3] have already shown that one cannot improve over CP, but in this work, we show somewhat surprisingly that GD does not Pareto-dominates in some regime (exponential accuracy).
>
> Additionally, our algorithms work in the face of a stronger form of memory constraint than described in the literature: they are limited in memory between iterations, but also for within-iteration computations.
>
>
> For your question about the geometric intuition, we agree that the description of the algorithms and their performance is a bit abstract, and we will provide more explanations in the paper. We give in Figure 1 from the rebuttal pdf a geometric illustration in dimension 2 of the recursive step, where an approximate separating hyperplane to the problem ``projected'' to the $x$-axis is constructed using a separating oracle for the $2$-dimensional problem. To be precise, the target is $\boldsymbol p^* = (p_x^*,p_y^*)$, and we use two blocks (i.e. $p=2$). Suppose at a step of the algorithm, the current value of the $x$ coordinate is $c$. We then aim to find an approximate separating hyperplane between $x = p_x^*$ and $x = c$ for some $c$, Algorithm 2 first runs Algorithm 1 (i.e. the memory-constrained Vaidya) to find two separating hyperplanes (the two blue hyperplanes). Lemma 4.1 then guarantees the existence of a convex combination of the 2 blue hyperplanes -- the red hyperplane-- which is approximately parallel to the $y$-axis and thus can serve as an approximate separating hyperplane between $x = p_x^*$ and $x = c$.
>
> [1] Marsden, Annie, et al. "Efficient convex optimization requires superlinear memory." Conference on Learning Theory. PMLR, 2022.
>
> [2] Blanchard, Moïse, Junhui Zhang, and Patrick Jaillet. "Quadratic memory is necessary for optimal query complexity in convex optimization: Center-of-mass is Pareto-optimal." Conference on Learning Theory. PMLR, 2023.
>
> [3] Chen, Xi, and Binghui Peng. "Memory-Query Tradeoffs for Randomized Convex Optimization." arXiv preprint arXiv:2306.12534 (2023).

---

> > ### Comment · Reviewer_U1uo · 2023-08-20
> >
> > Thank you for the rebuttal, and for the geometric explanation and illustration in 2-dimension. It does help me understand better some of the intuition (though I still find it harder to geometrically understand partitioning of variables, in that the segmenting of dimensions may seem somewhat arbitrary, as a way to make the algorithm and analysis go through, but not something “forced by nature”.)
> >
> > I keep my score.

---

### Official Review · Reviewer_6ZsG · 2023-07-12

**Soundness:** 4 excellent
**Presentation:** 4 excellent
**Contribution:** 4 excellent
**Rating:** 6
**Confidence:** 2

**Summary:**

The authors propose an algorithm to solve a feasibility problem with constrained memory while minimizing the number of calls in separation oracle. The memory complexity and the calls to separation oracle is parametetrized by a parameter $p$.  They get $O(d^2/p \log 1/\epsilon)$ bits of memory with $O(d/p\log(1/\epsilon))^p$ calls to separation oracle. Their result basically improves the case where $\epsilon$ is much smaller than $1/d$. They also provide a lower bound of that shows the dependences on $\log(1/\epsilon)$ for several settings. The algorithm is based on the Vaidya's cutting plane algorithm.

**Strengths:**

Contains new contributions for the regime where $\epsilon$ is much smaller than $1/d$.  The lower bound is also an interesting and a new contribution and potentially of independent interest. Overall, the paper is well written.

**Weaknesses:**

For me, the only weakness is that as you improve the memory, you increase the number of calls exponentially, which does not sound so intuitive to me. But still, the results are interesting.

**Questions:**

-

**Limitations:**

no limitations.

---

> ### Author Rebuttal · Authors · 2023-08-07
>
> Thank you for your positive view of the paper. The number of oracle class indeed grows exponentially with the parameter $p$, due to the recursive nature of the algorithm. We emphasize however that these are the first class of algorithms providing a positive result to trade-off memory and oracle-complexity in convex optimization, while the question of understanding the memory/oracle-complexity trade-off was well-identified in the literature (in its strong form, this was formulated in a COLT 2019 open problem [1]). Our results do not fully describe the memory/oracle-complexity landscape but provide significant clarifications. We first show that it is in fact possible to improve in some regime over the two fundamental algorithms in optimization: gradient descent and cutting-planes. Second, while it was known that cutting-planes are Pareto-optimal, we show somewhat surprisingly that in some exponential regime, gradient descent is not. Our algorithms allow an optimizer to specify a memory usage (at the cost of oracle complexity) instead of being restricted to either linear or full (quadratic) memory. Prior to this work, results were only obtained for lower-bounds, i.e., impossibility results [2,3,4], and novel ideas seemed to be required to make progress on the algorithmic side. In particular, to the best of our knowledge, previous approaches for optimization in low dimensions always showed significantly stronger curse of dimensionality [5] than say our $(d/p\ln\frac{1}{\epsilon})^d$ oracle dependence. Here we propose a recursive approach to solve low-dimensional sub-problems together with a careful analysis, which we believe are novel.
>
>
> [1] Woodworth, Blake, and Nathan Srebro. "Open problem: The oracle complexity of convex optimization with limited memory." Conference on Learning Theory. PMLR, 2019.
>
> [2] Marsden, Annie, et al. "Efficient convex optimization requires superlinear memory." Conference on Learning Theory. PMLR, 2022.
>
> [3] Blanchard, Moïse, Junhui Zhang, and Patrick Jaillet. "Quadratic memory is necessary for optimal query complexity in convex optimization: Center-of-mass is Pareto-optimal." Conference on Learning Theory. PMLR, 2023.
>
> [4] Chen, Xi, and Binghui Peng. "Memory-Query Tradeoffs for Randomized Convex Optimization." arXiv preprint arXiv:2306.12534 (2023).
>
> [5] Ma, Yi-An, et al. "Sampling can be faster than optimization." Proceedings of the National Academy of Sciences 116.42 (2019): 20881-20885.

---

### Official Review · Reviewer_LsGk · 2023-07-21

**Soundness:** 4 excellent
**Presentation:** 4 excellent
**Contribution:** 3 good
**Rating:** 7
**Confidence:** 3

**Summary:**

The paper discusses a parameterized family of memory-constrained algorithms for convex optimization and compares the memory and oracle call complexity of this family with existing algorithms. The authors show that with this parameterized family, one can trade between oracle call and memory complexity, and they claim that this is the first work in this direction.

**Strengths:**

The paper is very well written and accessible even to readers slightly unfamiliar with the topic (but see also below for some comments). The clarity is particularly facilitated by the excellent setup in Section 2, and also the overview of related work in Section 2.1 is exemplary. The paper seems theoretically quite strong (although I admittedly did not check the proofs in the supplementary material) and the algorithms are easy to understand. Figure 1 excellently summarizes the significance of the paper in the sense that the known upper and lower bounds are non-trivially improved upon.

**Weaknesses:**

One weakness of the paper is the introduction, which requires (in my opinion) more prior knowledge than what may be available from a non-expert. For example, the concept of a separation oracle in line 33 did not become clear to me until line 99, and similarly for the feasibility problem in line 42. In line 36 it is said that center-of-mass methods are "quadratic" in memory, from which I deduced quadratic in $d$ (but not in $-\log\epsilon$); then the statement that the algorithm improves upon center-of-mass for $p=1$ in line 54 is confusing. This is not a critical weakness, though, as the rest of the paper is quite accessible.

Another concern is whether the content of the paper fits well the scope of the venue. This is not immediately clear to me and shall be discussed in the discussion phase.

## Minor:
- line 115: "can also be carried OUT"?
- line 122: "all known lower boundS"
- line 139: "one needs to store $O(d)$ cuts instead OF $O(...$


**Questions:**

- In Def. 2.2, you write that the "the contents of $Q$ and $N$ are $q$ and $n$, respectively" and that "$R$ must contain at least $n$ bits". Is the latter the same $n$ as the former?
- What is the meaning of the notation $\vee$ and $\wedge$ in Corollary 3.1?

**Limitations:**

I do not see any immediate negative societal impacts, hence it is fine that the authors did not mention any. The limitations of the study are clearly discussed in Section 5.

---

> ### Author Rebuttal · Authors · 2023-08-07
>
> Thank you very much for your positive view of our paper and its contributions, and for your useful remarks. We will take care to solve all minor issues in the revised manuscript. We will also make sure to clarify parts of the exposition mentioned. With respect to the conference, NeurIPS has a tradition of publishing optimization papers that provide theoretical insights. This paper contributes to the overall goal of understanding resource constraints for optimization algorithms and designing efficient algorithms for these tasks, which we believe is of interest to the NeurIPS community in general, and its optimization sub-community in particular.
>
> In Def 2.2, the two $n$ are indeed the same: roughly speaking, before making a query to the oracle, the algorithm first needs to specify a bit precision for the response. That precision $n$ is stored in the placement $N$ which is read by the oracle. The oracle then writes the subgradient rounded to a precision of $n$ bits within the response placement $R$, which should therefore contain at least $n$ bits.
>
> Thank you for noting that we did not define the notations $\land$ and $\lor$, which act as minimum and maximum operators. We will add a formal definition in the revised manuscript. For clarity, Corollary 3.1 states that using $p\leq \mathcal O(\min(d,\frac{\ln\frac{1}{\epsilon}}{\ln d}))$, this provides a tradeoff between cutting planes and having memory $\mathcal O(\max(d^2\ln d,d\ln\frac{1}{\epsilon}))$ (we recall that $d\ln\frac{1}{\epsilon}$ memory is necessary) and oracle complexity $\mathcal O(\min(\frac{1}{\epsilon^2},(C\ln \frac{1}{\epsilon})^d))$.

---

> > ### Comment · Reviewer_LsGk · 2023-08-21
> > **Thank you!**
> >
> > Dear authors,
> >
> > Thanks for your answers to my questions. Whether the paper fits the venue is anyway a decision that has to be made by the area/program chairs -- but, liking your paper, I would be happy to see that their decision is positive!

---

### Official Review · Reviewer_snNf · 2023-07-26

**Soundness:** 4 excellent
**Presentation:** 3 good
**Contribution:** 3 good
**Rating:** 6
**Confidence:** 3

**Summary:**

The authors study memory vs oracle calls trade-off for convex optimization. The examples of problems they consider are: To find a point in the $d$-dimensional unit ball up to error $\epsilon$, or to minimize $1$-Lipschitz convex functions over the unit ball up to error $\epsilon$.
They provide a family of algorithms parametrised by $p \in [d]$ that solve the problem using $O(d^2 \log (1/\epsilon) / p)$ bits of memory and make $(C d \log (1/\epsilon)/p)^p$ oracle calls, where $C$ is some absolute constant.

More precisely, the first result (the case $p=1$) is the proof that a memory-constrained Vaidya’s method has (optimal) oracle complexity $O(d\log(1/\epsilon))$ and memory complexity $O(d^2\log(1/\epsilon))$, that is a $\log(1/\epsilon)$-factor improvement over the state-of-the-art. For $p=d$, their algorithm has oracle complexity $(C\log (1/\epsilon))^d$ and (optimal) memory complexity $O(d\log(1/\epsilon))$, that improves over state-of-the art oracle complexity $O(1/\epsilon^2)$ if $\log(1/\epsilon) \gtrsim d\log d$.

**Strengths:**

The paper provides new algorithms with better memory vs oracle calls trade-off than the state-of-the art. The paper is well-written, and the contribution is clear. The algorithms are non-trivial and their analyses are sophisticated.

**Weaknesses:**

My main concern is that the results only slightly improve over the state-of-the-art.

The first result improves over the state-of-the-art only by a $\log(1/\epsilon)$-factor. It is not surprising, since there are lower bounds that (for the feasibility problem) imply that $d^{2-\Omega(1)}$ memory complexity can only be achieved with $> d^{1+\Omega(1)}$ queries, but I am not sure if this result itself is strong and/or interesting enough to justify acceptance in NeurIPS.

The importance of the second result (with $p>1$) is not very clear to me. Let memory complexity of memory-constrained Vaidya’s method be $M$. In order to achieve memory complexity $o(M)$ (even $M/\log \log M$), one needs to increase the oracle complexity by a factor that is super-polynomial in $M$. I can hardly imagine settings when it might be reasonable. For comparison, there are some problems for which information-computation trade-offs are known (e.g. planted clique), and for those problems even a minor improvement over the state-of-the-art might be important, since there are basically only two options: Either we solve the problem (and perhaps spend a lot of computational resources), or do not solve it at all. Here it is not the case: the algorithm designer can always choose to use the memory-constrained Vaidya’s method, or to slightly decrease memory complexity and significantly increase the number of oracle calls (and hence the running time), and it seems to me that the huge price of memory complexity is not adequate here.

One could argue that the trade-off that is not useful in practice can be interesting if it clarifies memory vs oracle complexity picture of the problem. But the upper bounds from the paper are very far from the currently known lower bounds, and it does not improve a high-level understanding of the complexity picture. For example, it is not clear if super-polynomial oracle complexity is necessary for sub-quadratic memory complexity.

Considering these strengths and weaknesses, I recommend borderline reject.

UPDATE: After reading the rebuttal I decided to increase the score from 4 to 6. Please see the comments below.

**Questions:**

I have a question related to non-efficient algorithms: Since to match the optimal memory complexity your algorithm requires $p=d$ (and so the running time is exponential in $d$), it makes sense to compare your algorithms with inefficient approaches. Naive brute force for the feasibility problem (that only uses a weaker oracle for an indicator of the set, not a separation oracle) seems to require $(C/\epsilon)^d$ oracle calls, which is too large, but maybe there exists some other simple brute force algorithm that uses the separation oracle and has comparable guarantees to your algorithm. Did you by chance think about such an algorithm?

**Limitations:**

The authors adequately addressed the limitations of their work.

---

> ### Author Rebuttal · Authors · 2023-08-07
>
> Thank you for your comprehensive summary and your positive view of our algorithms and analysis. We are very happy to provide some answers to the concerns you raised.
>
> 1. We believe that this work provides some important clarifications to the memory/oracle complexity landscape. We emphasize that to the best of our knowledge, prior to this work, no positive results nor algorithms were known to improve in any way over two most foundational algorithms for optimization: gradient descent (GD) and cutting-planes (CP) (including ellipsoid methods). The specific question of whether it is possible to improve these algorithms in the memory/oracle complexity picture was also present in the literature and stated in its strong form as a COLT 2019 open problem [1]. Although our work does not provide a full understanding of the picture:
>
> - We show that in super-polynomial regimes, it is indeed possible to improve over these algorithms (dark green region in Figure 1). To the best of our knowledge this is the first result making some progress on the algorithmic side as opposed to lower bounds for which the literature is better established [2,3,4]. We further provide a class of algorithms giving a positive trade-off between memory and oracle complexity. This enables an optimizer to specify a desired memory usage (at the price of oracle complexity), through the parameter $p$, instead of restricting the choice to either linear or full (quadratic) memory.
> - A major question was whether one can Pareto-improve over GD and/or CP. Lower bounds [3,4] have already shown that one cannot improve over CP, but in this work, we show somewhat surprisingly that GD does not Pareto-dominates in some regime (exponential accuracy).
>
> Additionally, our algorithms work in the face of a stronger form of memory constraint than described in the literature: they are limited in memory between iterations, but also for within-iteration computations.
>
> 2. To answer your question, we are not aware of any brute-force (or other) algorithms that have comparable guarantees to our algorithms (say for $p=d$) for non-smooth convex optimization. Closest to brute-force methods are some works that consider smooth optimization in low-dimensional settings.
>
> - Vavasis [5] showed that thanks to smoothness, combining brute-force search with gradient approaches one can achieve $1/\epsilon^{\frac{2d}{d+2}}$ oracle complexity. This slightly improves over what gradient descent would achieve in our setting, only if $\log \frac{1}{\epsilon} \gg d$. However, in that regime $\log \frac{1}{\epsilon} \gtrsim d\ln d$, the oracle complexity $(C\log\frac{1}{\epsilon})^d$ from our method with $p=d$ is much lower than a polynomial in $\frac{1}{\epsilon}$, and does not require smoothness.
> - The polynomial exponent for the oracle complexity from [5] was then improved in dimensions 2 and 3 by Bubeck and Mikulincer [6] with clever ideas to side-step brute-force approaches. In high-dimensions, their algorithms achieve $(\frac{\log 1/\epsilon}{\epsilon})^{\frac{d-1}{2}}$, which exhibits a strong curse of dimensionality (at least $\epsilon^{-d/4}$).
>
> These two works considered harder non-convex problems but heavily used the smoothness assumption. It is worth noting that using smoothness, Monte-Carlo approaches may outperform optimization in terms of oracle complexity (e.g [7], but the suboptimality measure considered is somewhat different), which always shows some form of dimensionality curse [7].
>
> The above-mentioned approaches do not seem to give successful results for our non-smooth convex optimization setup and other ideas seemed to be required compared to the literature to possibly improve over gradient descent. Instead, we use different techniques based on a recursive reduction to lower-dimensional subproblems, which we believe are novel.
>
> 3. Last, the study of super-polynomial accuracies is not uncommon in optimization. Following the seminal paper [5], there has been a growing literature that considered optimization in low-dimensional (or even constant as in [6]) settings---for instance, in dimension $d$, the improvement from [5] in smooth-nonconvex optimization over $\frac{1}{\epsilon^2}$ is significant when $\log\frac{1}{\epsilon}\gg d$. In that asymptotic in $\epsilon$ perspective, our bounds in the exponential accuracy regime are (a high-degree) polynomial in $\log\frac{1}{\epsilon}$ instead of $\frac{1}{\epsilon^2}$. Also, in the specific literature for memory/oracle-complexity tradeoffs, some papers specifically study super-polynomial accuracy regimes, e.g. [4] which consider accuracies $\epsilon\leq \exp(-\log^4 d)$ (to appear at FOCS 2023 according to the author's website).
>
>
> [1] Woodworth, Blake, and Nathan Srebro. "Open problem: The oracle complexity of convex optimization with limited memory." Conference on Learning Theory. PMLR, 2019.
>
> [2] Marsden, Annie, et al. "Efficient convex optimization requires superlinear memory." Conference on Learning Theory. PMLR, 2022.
>
> [3] Blanchard, Moïse, Junhui Zhang, and Patrick Jaillet. "Quadratic memory is necessary for optimal query complexity in convex optimization: Center-of-mass is Pareto-optimal." Conference on Learning Theory. PMLR, 2023.
>
> [4] Chen, Xi, and Binghui Peng. "Memory-Query Tradeoffs for Randomized Convex Optimization." arXiv preprint arXiv:2306.12534 (2023).
>
> [5] Vavasis, Stephen A. "Black-box complexity of local minimization." SIAM Journal on Optimization 3.1 (1993): 60-80.
>
> [6] Bubeck, Sébastien, and Dan Mikulincer. "How to trap a gradient flow." Conference on Learning Theory. PMLR, 2020.
>
> [7] Ma, Yi-An, et al. "Sampling can be faster than optimization." Proceedings of the National Academy of Sciences 116.42 (2019): 20881-20885.

---

> > ### Comment · Reviewer_snNf · 2023-08-20
> >
> > Dear Authors,
> >
> > I apologize for such a late reply. I would like to thank you for your detailed response, it was very interesting to read.
> > After reading your reply and having another look at the paper, I increase my score to 6 (weak accept), and the contribution to 3 (good).
> >
> > Before I did not find your result with optimal memory complexity very interesting since it is better than the gradient descent only in the regime $\log(1/\varepsilon)\gtrsim d\log d$, and I didn’t find this regime natural since it seemed to me that prior works only focused on the regime $\varepsilon = poly(1/d)$. However, since the recent FOCS paper [4] you mentioned focused on the super-polynomial accuracy, it somehow changes the story: Super-polynomial accuracy is for some reason needed in [4] to show a lower bound against (randomized) algorithms that use linear number of oracle calls. You work with super-polynomial accuracy to show a new upper bound among algorithms that use linear memory complexity, which indeed sounds nice in the context of the result of [4].
> >
> > Actually, the accuracy used in [4] (in their theorem formulation) is $\varepsilon = \exp(-\log^5 d)$, while you wrote $\varepsilon \le \exp(-\log^4 d)$. I don’t care so much about 4 or 5, but an important thing is the inequality you wrote: if they need only an upper bound for $\varepsilon$, that sounds good, and I believe you here that they actually have $\varepsilon \le \ldots$ and not only $\varepsilon = \ldots$, but please confirm this if you have time (I understand that my reply is very late, and you might not be able to response…). And of course please add a reference to [4] to the final version of your paper and perhaps even some discussion on why do they need super-polynomial accuracy (I believe they didn't just chose to work in this regime, there is perhaps the reason why they don't work with $\varepsilon = poly(1/d)$). That might make the motivation of studying the regime you work with more clear.
> >
> > Another thing is the paper [5] you mentioned. I think it would be nice if you also cite it in your paper. As I understood (I didn’t have a look at that paper), they have a minor, but asymptotic $1/poly(d)$ improvement over the gradient descent in the regime  $\log(1/\varepsilon)=\Theta(d\log d)$. Still, if we denote the oracle complexity of the gradient descent by $T$, prior to your work no algorithms with oracle complexity $T^{1-\Omega(1)}$ were known, and given lower bounds [3,4] one could have expected that they might not exist. You show that it is not the case, and I think it is good to explicitly write it (otherwise your current text might be wrongly interpreted that the gradient descent was asymptotically the best in terms of oracle calls).

---

> > > ### Author Response · Authors · 2023-08-20
> > >
> > > Dear reviewer,
> > >
> > > Thank you very much for your thoughtful comments and positive view of our contributions!
> > >
> > > Indeed, the super-polynomial accuracy seems to be important in [4] in order to get lower bounds for a communication game that they then encode into a convex optimization problem. As suggested, we will make sure to add a discussion on [4] in the revised manuscript and more details about why they also need super-polynomial accuracy.
> > >
> > > The rebuttal indeed had a typo. Their accuracy is indeed $\exp(-\log^5 d)$. As you had anticipated, we wrote an inequality $\epsilon\leq ...$ to emphasize that their arguments hold for that whole regime (not just the equality): taking their same class of hard functions, an algorithm solving the problem to accuracy $\epsilon\leq \exp(-\log^5 d)$ solves it in particular to accuracy $\exp(-\log^5 d)$.
> > >
> > > As you kindly suggested, we will also of course add a discussion [5] which indeed gave some improvements over gradient descent in the regime $\log1/\epsilon = \Theta(d\log d)$. We would like to point out, however, that this result relies heavily on smoothness and as such would not give improvements in our non-smooth convex optimization setting (the idea is to run gradient descent starting from all initial points in a grid: smoothness ensures that in the neighborhood of the initial point closest to the optimum, the Lipschitz constant of the gradients is also reduced, which allows to adapt gradient descent parameters by taking longer more aggressive steps). To the best of our knowledge, it was unknown in prior works whether using the same $\Theta(d\log 1/\epsilon)$ memory (or in fact anything strictly less than quadratic memory, say memory $d^{2-\Omega(1)}\log 1/\epsilon$), gradient descent was asymptotically best in that non-smooth case.
> > >
> > > Thank you again for your time and efforts in reviewing the paper!

---

### Author Rebuttal · Authors · 2023-08-07

In this document, we provide in Figure 1, a geometric illustration of the recursive step of our algorithm, where an approximate separating hyperplane to the father problem (in the recursive hierarchy) is computed (Reviewers U1uo and H4w2).

---

### Decision · Program_Chairs · 2023-09-21

**Decision:**

Accept (poster)

**Comment:**

This paper proposes a parametrized class of algorithms for convex optimization that allow to trade-off memory and oracle calls via the choice of the parameter (p). For a certain range of parameters (depending also on the target accuracy epsilon) the method attains an improved memory/oracle call trade-off than previously known schemes. The paper also shows that the attained trade-off cannot significantly be improved in the high accuracy (low epsilon) regime.

The question on the optimal memory/oracle call trade-off was originally raised in (Woodworth and Srebro, 2019, "Open problem: The oracle complexity of convex optimization with limited memory"). This paper makes advances to understand this trade-off better (in the deterministic case), and complements other works on this open problem that focused don lower-bound impossibility results.

Please add the comments/discussion regarding the super-polynomial accuracy regime (as below) to the final version of the paper.